# Evaluation of the U.S. Peanut Germplasm Mini-Core Collection in the Virginia-Carolina Region Using Traditional and New High-Throughput Methods

Sayantan Sarkar [1], Joseph Oakes [2], Alexandre-Brice Cazenave [3], Mark D. Burow [4,5], Rebecca S. Bennett [6], Kelly D. Chamberlin [6], Ning Wang [7], Melanie White [8], Paxton Payton [8], James Mahan [8], Jennifer Chagoya [4], Cheng-Jung Sung [5], David S. McCall [9], Wade E. Thomason [9] and Maria Balota [10,*]

1   Blackland Research and Extension Center, Texas A&M Agrilife Research, Temple, TX 76502, USA
2   Eastern Virginia AREC, Virginia Tech, Warsaw, VA 22572, USA
3   Bayer Crop Science, Stanton, MN 55018, USA
4   Texas A&M AgriLife Research, Lubbock, TX 79403, USA
5   Department of Plant and Soil Science, Texas Tech University, Lubbock, TX 79409, USA
6   U.S. Department of Agriculture-Agricultural Research Service, Stillwater, OK 74075, USA
7   Biosystems & Agricultural Engineering, Oklahoma State University, Stillwater, OK 74078, USA
8   U.S. Department of Agriculture-Agricultural Research Service, Lubbock, TX 79415, USA
9   School of Plant and Environmental Sciences, Virginia Tech, Blacksburg, VA 24060, USA
10  Tidewater AREC, Virginia Tech, Suffolk, VA 23437, USA
*   Correspondence: mbalota@vt.edu

**Abstract:** Peanut (*Arachis hypogaea* L.) is an important food crop for the U.S. and the world. The Virginia-Carolina (VC) region (Virginia, North Carolina, and South Carolina) is an important peanut-growing region of the U.S and is affected by numerous biotic and abiotic stresses. Identification of stress-resistant germplasm, along with improved phenotyping methods, are important steps toward developing improved cultivars. Our objective in 2017 and 2018 was to assess the U.S. mini-core collection for desirable traits, a valuable source for resistant germplasm under limited water conditions. Accessions were evaluated using traditional and high-throughput phenotyping (HTP) techniques, and the suitability of HTP methods as indirect selection tools was assessed. Traditional phenotyping methods included stand count, plant height, lateral branch growth, normalized difference vegetation index (NDVI), canopy temperature depression (CTD), leaf wilting, fungal and viral disease, thrips rating, post-digging in-shell sprouting, and pod yield. The HTP method included 48 aerial vegetation indices (VIs), which were derived using red, blue, green, and near-infrared reflectance; color space indices were collected using an octocopter drone at the same time, with traditional phenotyping. Both phenotypings were done 10 times between 4 and 16 weeks after planting. Accessions had yields comparable to high yielding checks. Correlation coefficients up to 0.8 were identified for several Vis, with yield indicating their suitability for indirect phenotyping. Broad-sense heritability ($H^2$) was further calculated to assess the suitability of particular VIs to enable genetic gains. VIs could be used successfully as surrogates for the physiological and agronomic trait selection in peanuts. Further, this study indicates that UAV-based sensors have potential for measuring physiologic and agronomic characteristics measured for peanut breeding, variable rate input application, real time decision making, and precision agriculture applications.

**Keywords:** peanut; U.S. mini-core collection; Virginia-Carolina region; vegetation indices; high-throughput phenotyping; color space indices; crop physiology

## 1. Introduction

Peanut (*Arachis hypogaea* L.) is an important oil and food crop, with an acreage of 42 million worldwide. It is one of the major oilseed crops, and China, India, Nigeria, and the U.S. contribute to about 70% of its global production [1]. In 2019, in the U.S., over

5.5 billion pounds were produced from 567 thousand hectares across 11 states, which are limited to three geographical regions: Southeast region (Alabama, Florida, Georgia, and Mississippi), Southwest region (Oklahoma, New Mexico, and Texas), and VC region (North Carolina, South Carolina, and Virginia) [2]. Because each growing region differs in climate and disease pressures, breeding programs develop peanut varieties specifically adapted to each growing region [3,4]. The VC region primarily produces the large seeded Virginia peanut market type, and has an annual production of around $170 million [2]. The climate of the VC region is different from the other regions. Virginia and the Carolinas have a humid, subtropical climate with a 35-year multi-annual cumulative precipitation at 590 mm, average minimum, and maximum temperatures of 4 °C and 36 °C, respectively, and 78% relative humidity during the peanut growing season (May to September) (Figure 1). Because of the warm and humid climate, the peanut is prone to numerous diseases and pathogens including southern stem rot (SSR) [caused by *athelia rolfsii* (Curzi) C.C. Tu and Kimbr.], sclerotinia blight (SB, caused by *sclerotinia minor*, Jagger), cylindrocladium black rot (CBR, caused by *calonectria ilicicola*, Boedijn and Reitsma) and tomato spotted wilt virus (TSWV, genus *Tospovirus*, family *Bunyaviridae*) [5]. As soils are shallow and sandy, and summer temperatures are high, peanut crops can experience sudden drought in the VC region [6]. Water deficit during pegging or pod formation stages severely effect peanut yield [7–10]. Low-moisture stress may also reduce nitrogen (N) fixation and biomass growth, and increase aflatoxin contamination [11–14]. Future predictions have also shown that peanuts would be one of the worst affected crops, as a result of global warming and associated climate change by 2050 [15].

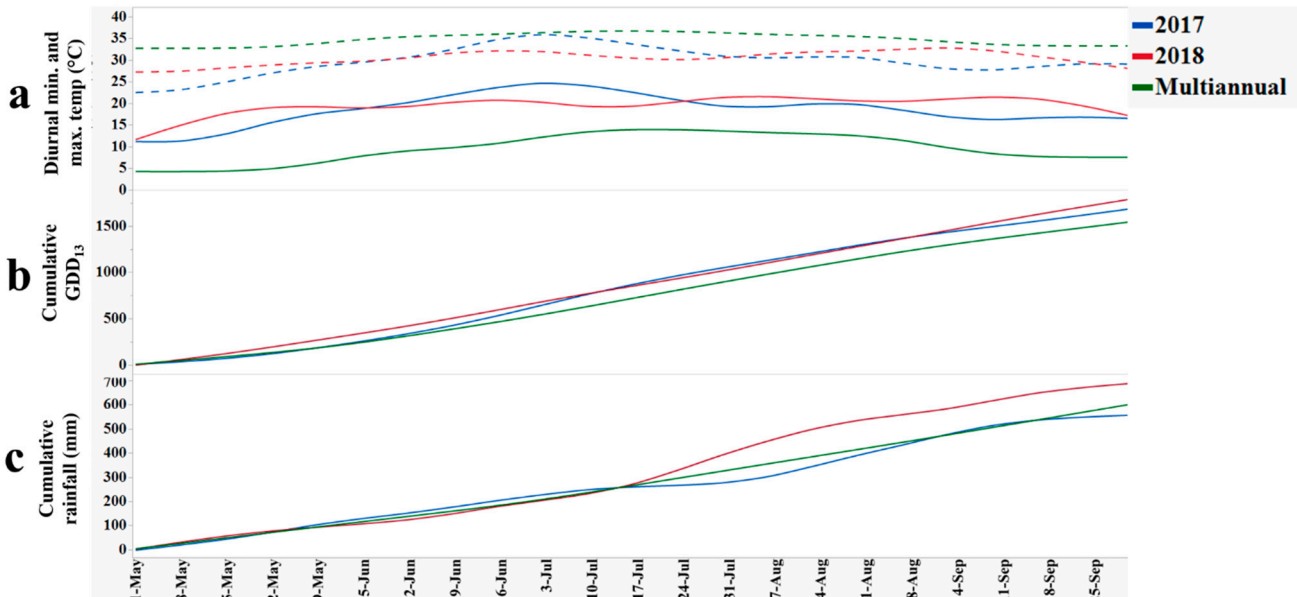

**Figure 1.** Weather data at Suffolk, VA, including: (**a**) diurnal minimum (solid line) and maximum (dashed line) temperatures (°C); (**b**) cumulative growing degree days calculated from daily min and max temperatures with 13 °C as base temperature (GDD$_{13}$); and (**c**) cumulative rainfall (mm) for 2017, 2018, and multiannual (1984–2019) average.

Identification of sources with resistance to biotic and abiotic stressors is needed to further improve peanut production in the VC region, but its success heavily relies on the phenotyping methods [16,17]. Aerially derived vegetation indices (VIs) from red-green-blue (RGB) and near-infrared (NIR) imagery were recently used for phenotyping morphological, physiological, and agronomic characteristics of multiple crops: peanut [18–21]; winter wheat (*Triticum aestivum* L.) [22,23]; sorghum (*Sorghum bicolor* L. Moench) [24–26]; cotton (*Gossypium hirsutum* L.) [27]; tall fescue (*Festuca arundinacea* Schreb) [28]; corn (*Zea mays* L.) [29]; soybean (*Glycine max* L.) [30–32]; and other crops [33]. However, to effectively use

the VIs in breeding, they need to be heritable [34], i.e., a high proportion of the VIs' variation is attributable to genetic factors [35–37]. Therefore, information on the VIs accuracy to predict crop characteristics, along with their heritability, is needed to develop successful high-throughput (HTP) methods for breeding selection.

The U.S. peanut germplasm collection currently has 7432 accessions, and is a potential resource for resistance to biotic and abiotic stresses. In the late 1990s, approximately 10% of these accessions, representing the geographical and morophological variation of the entire collection, were selected as the U.S. peanut core collection [38]. Because evaluating all 831 core accessions in the field is difficult, a 112-accession representative subset of the core collection, the mini-core collection, was selected to identify genes of interest in peanut breeding [39]. The mini-core collection has been evaluated in the Southeast and Southwest for multiple traits, including resistance to the peanut root-knot nematode (caused by *meloidogyne arenaria* [Neal] Chitwood), early leaf spot (caused by *passalora arachidicola* [Hori] U. Braun), late leaf spot (caused by *nothopassalora personata* [Berk. and M.A. Curtis] U. Braun, C. Nakash., Videira and Crous), TSWV, SSR, and SB; post-harvest quality traits including oil, fatty acid, flavonoid, and resveratrol content; and seed dormancy [39–47]. To our knowledge, the U.S. peanut mini-core has neither been evaluated in the VC region, nor has it been evaluated for morphological, physiological, and agronomic characteristics anywhere else. The objectives of this study were: (1) to assess the mini-core accessions for morphological, physiological, and agronomic characteristics relevant to the VC environment, using traditional and new phenotyping techniques such as RGB and NIR aerial imagery; and (2) to demonstrate the suitability of the new techniques for high throughput phenotyping.

## 2. Materials and Methods

### 2.1. Germplasm Information

The mini-core collection includes accessions with varieties *hirsuta*, *hypogaea*, *fastigiata*, *vulgaris*, and *peruviana*, and four peanut market types, Runner, Virginia, Spanish, and Valencia [38,48,49]. Varieties *hypogaea* and *hirsuta* belong to peanut subspecies *hypogaea*, which includes Runner and Virginia market types. Varieties *vulgaris*, *peruviana*, and *fastigiata*, belong to subspecies *fastigiata*, which includes Spanish and Valencia market types [50]. Details on the botanical and market types of the mini-core accession are available within the Germplasm Resource Information Network (GRIN) plant germplasm database (https://npgsweb.ars-grin.gov/gringlobal/search, accessed on 14 June 2020) and several publication; however, for some accessions, the information differs depending on the source. A compiled list of 112 accessions of the U.S. mini-core peanut germplasm collection, with information regarding taxonomy, market and variety, pod weight, 100 seed weight, and seed kernel color properties from all sources, is available to date and presented in Table 1 (GRIN database, https://npgsweb.ars-grin.gov/gringlobal/search, accessed on 14 June 2020) [39,41,48]. As some sources presented different market and varieties with the same PI numbers, separate columns for each source were included within Table 1. The kernel color information in Table 1, i.e., hue, lightness (L), a*, and b* color properties, were derived by pictures available on the GRIN database. The pictures were checked for uniformity to ensure each picture had similar resolution, background, and margins, before color properties of the kernels were extracted using BreedPix tool from the CIMMYT maize scanner 1.16 plugin (http://github.com/george-haddad/CIMMYT, accessed on 18 September, 2020; Copyright 2015 Shawn Carlisle Kefauver, University of Barcelona), produced as a part of the Image J/Fiji (open source software; http://fiji.sc/Fiji, accessed on 18 September 2020) [51,52]. In Table 1, the PI accessions used in this study are underlined.

**Table 1.** List of the 112 accessions of the U.S. mini-core peanut germplasm collection, including PI numbers from the GRIN database. The table includes market type, variety, pod type, pod shape, and 100-seed weight, compiled from different sources (in the footnote). Kernel color (CIE Lab) is also included, derived from seed pictures available on the GRIN database. The underlined PI and CC numbers were planted for this study.

| S. No. | PI Number | CC | Market Type [1] | Market Type [2] | Variety [3] | Pod Type [4] | Pod Shape [4] | 100 Seed wt. (g) [4] | Kernel Color [5] | | | |
|---|---|---|---|---|---|---|---|---|---|---|---|---|
| | | | | | | | | | Hue | L | a | b |
| 1 | PI 152146 | 406 | Spanish | Spanish | . | Spanish | hypogaea | 48.5 | 28.4 | 67.2 | 3.96 | 26.0 |
| 2 | PI 155107 | 384 | . | Valencia | vulgaris | Spanish | vulgaris | 38.3 | 28.9 | 67.3 | 4.18 | 27.1 |
| 3 | PI 157542 | 553 | . | Runner | vulgaris | Virginia | hypogaea | 60.8 | 26.6 | 64.2 | 3.79 | 23.1 |
| 4 | PI 158854 | 559 | Valencia | Valencia | fastigiata | . | vulgaris | 60.8 | 10.6 | 57.0 | 11.73 | 18.6 |
| 5 | PI 159786 | 334 | Virginia | Virginia | hypogaea | Virginia | . | 34.8 | 27.2 | 64.2 | 4.03 | 24.1 |
| 6 | PI 162655 | 388 | Spanish | Spanish | hypogaea | Virginia | . | 39.5 | 28.1 | 67.3 | 4.73 | 27.1 |
| 7 | PI 162857 | 731 | Virginia | Virginia | hypogaea | Virginia | hypogaea | 87.2 | 24.4 | 61.3 | 7.25 | 25.7 |
| 8 | PI 196622 | 802 | Virginia | Virginia | hypogaea | Virginia | . | 50.9 | 26.1 | 61.8 | 5.28 | 24.6 |
| 9 | PI 196635 | 270 | Runner | Runner | hypogaea | Virginia | . | 32.8 | 26.9 | 64.4 | 4.57 | 24.9 |
| 10 | PI 200441 | 266 | Spanish | Spanish | fastigiata | Spanish | vulgaris | 42.4 | 28.7 | 69.0 | 3.18 | 25.3 |
| 11 | PI 240560 | 725 | . | Runner | hypogaea | Spanish | . | 38.2 | 29.4 | 67.5 | 4.00 | 27.4 |
| 12 | PI 259617 | 508 | . | Mixed | fastigiata | Valencia | fastigiata | . | 11.4 | 57.2 | 9.60 | 17.6 |
| 13 | PI 259658 | 506 | Runner | Runner | hypogaea | Virginia | . | | 24.7 | 64.8 | 6.77 | 26.0 |
| 14 | PI 259836 | 546 | Spanish | Spanish | fastigiata | Valencia | . | 30.8 | 13.0 | 58.8 | 8.58 | 17.9 |
| 15 | PI 259851 | 277 | Virginia | Virginia | hypogaea | Virginia | hypogaea | 59.8 | 26.2 | 65.4 | 5.14 | 25.2 |
| 16 | PI 262038 | 408 | Valencia | Valencia | fastigiata | Valencia | . | 34.3 | 10.3 | 57.4 | 10.00 | 17.3 |
| 17 | PI 268586 | 580 | Valencia | Valencia | hypogaea | Virginia | . | 48.1 | 18.6 | 57.3 | 0.99 | 13.2 |
| 18 | PI 268696 | 338 | Spanish | Spanish | hypogaea | Spanish | . | 38.5 | 30.1 | 69.0 | 3.04 | 26.7 |
| 19 | PI 268755 | 481 | . | Runner | hypogaea | Spanish | . | 49.5 | 28.6 | 66.8 | 3.22 | 24.8 |
| 20 | PI 268806 | 477 | Spanish | Spanish | hypogaea | Spanish | . | 49.0 | 29.7 | 67.7 | 3.27 | 26.5 |
| 21 | PI 268868 | 367 | Virginia | Virginia | hypogaea | Virginia | fastigiata | 46.2 | 29.3 | 67.9 | 1.91 | 23.2 |
| 22 | PI 268996 | 458 | . | Runner | hypogaea | Virginia | . | 38.7 | 26.7 | 64.0 | 4.75 | 24.8 |
| 23 | PI 270786 | 485 | . | Mixed | hypogaea | Spanish | . | 38.5 | 25.9 | 51.0 | −0.89 | 11.9 |
| 24 | PI 270905 | 446 | . | Mixed | hypogaea | Virginia | . | 48.8 | 25.3 | 64.0 | 5.17 | 24.0 |
| 25 | PI 270907 | 433 | . | Mixed | hypogaea | Virginia | hypogaea | 47.1 | 27.1 | 65.4 | 4.06 | 24.4 |
| 26 | PI 270998 | 468 | . | Mixed | vulgaris | Spanish | . | . | 16.6 | 61.2 | 5.68 | 17.7 |
| 27 | PI 271019 | 579 | . | Mixed | vulgaris | Spanish | . | 35.0 | 28.2 | 68.0 | 4.55 | 27.1 |
| 28 | PI 274193 | 208 | Virginia | Virginia | hypogaea | Spanish | vulgaris | 52.3 | 6.6 | 54.0 | 9.50 | 14.7 |
| 29 | PI 288146 | 516 | Virginia | Virginia | vulgaris | Spanish | . | 36.8 | 28.7 | 66.8 | 3.87 | 26.2 |
| 30 | PI 288210 | 526 | . | Runner | vulgaris | Virginia | hypogaea | 31.7 | 31.1 | 65.6 | −0.58 | 18.7 |
| 31 | PI 290536 | 233 | Virginia | . | hypogaea | Virginia | hypogaea | 40.1 | 26.4 | 65.2 | 4.42 | 24.2 |
| 32 | PI 290560 | 221 | . | Spanish | vulgaris | Spanish | vulgaris | 36.0 | 31.1 | 68.4 | 2.49 | 26.8 |
| 33 | PI 290566 | 227 | Runner | Runner | fastigiata | Valencia | fastigiata | 43.2 | 26.6 | 64.0 | 3.78 | 23.0 |
| 34 | PI 290594 | 230 | Runner | Runner | hypogaea | Valencia | fastigiata | 48.8 | 26.4 | 65.9 | 5.83 | 26.6 |
| 35 | PI 290620 | 223 | Virginia | Virginia | fastigiata | Spanish | vulgaris | 44.8 | 28.8 | 63.5 | 2.34 | 22.6 |
| 36 | PI 292950 | 728 | Runner | Mixed | hypogaea | Virginia | hypogaea | 67.8 | 45.6 | 71.8 | −4.63 | 27.1 |
| 37 | PI 295250 | 540 | Virginia | Virginia | hypogaea | Virginia | hypogaea | 44.8 | 9.8 | 58.6 | 11.37 | 18.1 |
| 38 | PI 295309 | 541 | . | Mixed | hypogaea | Virginia | hypogaea | 56.9 | 26.0 | 64.0 | 4.30 | 23.3 |
| 39 | PI 295730 | 8 | Virginia | Virginia | fastigiata | Valencia | vulgaris | 41.2 | 27.2 | 65.4 | 4.16 | 24.7 |
| 40 | PI 296550 | 534 | . | Runner | hypogaea | Virginia | hypogaea | 78.7 | 29.3 | 67.2 | 2.99 | 25.2 |
| 41 | PI 296558 | 535 | . | Runner | hypogaea | Virginia | hypogaea | 56.2 | 27.1 | 67.7 | 4.23 | 25.2 |
| 42 | PI 298854 | 342 | . | Runner | hypogaea | Virginia | hypogaea | 80.9 | 24.0 | 62.5 | 6.55 | 24.5 |
| 43 | PI 313129 | 381 | . | Mixed | fastigiata | Valencia | . | 46.7 | 31.1 | 67.9 | 2.61 | 26.8 |
| 44 | PI 319768 | 529 | Virginia | Virginia | hypogaea | Virginia | hypogaea | 45.2 | 28.4 | 61.5 | 4.99 | 26.7 |
| 45 | PI 319770 | . | . | . | . | . | vulgaris | 44.2 | 34.0 | 66.1 | 0.82 | 26.0 |
| 46 | PI 323268 | 812 | Virginia | Virginia | hypogaea | Virginia | . | 72.0 | 25.6 | 63.1 | 5.01 | 24.0 |
| 47 | PI 325943 | 548 | Valencia | Valencia | hypogaea | Valencia | fastigiata | 42.9 | 8.9 | 56.9 | 11.09 | 17.2 |
| 48 | PI 331297 | 202 | . | Mixed | hypogaea | Virginia | hypogaea | . | 10.4 | 53.9 | 9.74 | 16.6 |
| 49 | PI 331314 | 187 | . | Mixed | hypogaea | Valencia | vulgaris | 33.4 | 17.2 | 51.6 | 9.80 | 20.6 |
| 50 | PI 337293 | 431 | Valencia | . | hypogaea | Spanish | . | 44.0 | 25.0 | 57.5 | 2.04 | 17.6 |
| 51 | PI 337399 | 808 | Spanish | . | hypogaea | Spanish | . | 40.8 | 27.3 | 66.9 | 4.57 | 25.8 |
| 52 | PI 337406 | 310 | Runner | Runner | fastigiata | Spanish | vulgaris | 39.9 | 30.7 | 51.5 | −1.75 | 12.0 |
| 53 | PI 338338 | 552 | . | Valencia | peruviana | . | . | 34.1 | 14.5 | 54.7 | 1.70 | 12.2 |
| 54 | PI 339960 | 189 | Valencia | Valencia | fastigiata | Valencia | fastigiata | 54.3 | 9.3 | 58.1 | 11.75 | 18.0 |
| 55 | PI 343384 | 249 | Intermediate | Mixed | hypogaea | Virginia | hypogaea | 57.3 | 13.3 | 56.2 | 9.59 | 18.5 |
| 56 | PI 343398 | 246 | . | Virginia | fastigiata | Virginia | . | 63.0 | 25.9 | 62.9 | 5.14 | 24.4 |
| 57 | PI 355268 | 805 | Virginia | . | hypogaea | Virginia | . | 45.4 | 25.4 | 62.8 | 5.45 | 24.3 |
| 58 | PI 355271 | 287 | . | Runner | hypogaea | Virginia | . | 58.1 | 24.4 | 65.8 | 5.25 | 23.6 |
| 59 | PI 356004 | 488 | . | Mixed | fastigiata | Valencia | . | 38.2 | 11.1 | 57.3 | 9.86 | 17.6 |
| 60 | PI 370331 | 542 | Virginia | Virginia | hypogaea | Virginia | hypogaea | . | 9.6 | 54.9 | 9.00 | 15.9 |
| 61 | PI 371521 | 255 | . | . | hypogaea | Virginia | hypogaea | . | 25.8 | 65.6 | 4.81 | 24.3 |
| 62 | PI 372271 | 294 | Virginia | Virginia | hypogaea | Valencia | fastigiata | 45.8 | 25.7 | 64.4 | 4.94 | 24.1 |
| 63 | PI 372305 | 698 | Virginia | Virginia | hypogaea | Virginia | . | 43.7 | 28.7 | 66.4 | 3.10 | 24.5 |
| 64 | PI 399581 | 296 | Virginia | Virginia | hypogaea | Virginia | hypogaea | 52.6 | 28.0 | 61.9 | 4.82 | 26.0 |
| 65 | PI 403813 | 588 | Spanish | Valencia | vulgaris | Valencia | vulgaris | 36.0 | 10.2 | 55.5 | 10.73 | 17.5 |
| 66 | PI 407667 | 740 | Spanish | Spanish | vulgaris | Spanish | . | 59.0 | 26.2 | 66.2 | 4.13 | 23.7 |
| 67 | PI 408743 | 631 | Intermediate | Mixed | . | Spanish | vulgaris | . | 46.3 | 71.5 | −4.93 | 23.7 |
| 68 | PI 429420 | 787 | Valencia | Valencia | fastigiata | Valencia | . | 44.3 | 9.0 | 55.4 | 10.89 | 16.9 |
| 69 | PI 433347 | 643 | Spanish | . | . | Virginia | hypogaea | . | 33.0 | 62.3 | 0.54 | 22.8 |
| 70 | PI 442768 | 763 | Virginia | Virginia | hypogaea | Virginia | hypogaea | 43.0 | 29.0 | 64.2 | 4.01 | 26.2 |

**Table 1.** *Cont.*

| S. No. | PI Number | CC | Market Type [1] | Market Type [2] | Variety [3] | Pod Type [4] | Pod Shape [4] | 100 Seed wt. (g) [4] | Kernel Color [5] | | | |
|---|---|---|---|---|---|---|---|---|---|---|---|---|
| | | | | | | | | | Hue | L | a | b |
| 71 | PI 461427 | 647 | . | . | *hypogaea* | Valencia | *fastigiata* | 47.3 | 11.3 | 57.3 | 10.02 | 17.8 |
| 72 | PI 461434 | 798 | . | Runner | *hypogaea* | runner | *vulgaris* | 47.3 | 29.8 | 69.1 | 3.24 | 26.8 |
| 73 | PI 468271 | . | . | . | . | Virginia | *hypogaea* | . | 47.2 | 59.9 | −4.40 | 16.4 |
| 74 | PI 471952 | 760 | Spanish | Spanish | *hypogaea* | Virginia | *hypogaea* | 73.5 | 24.1 | 63.1 | 5.82 | 23.7 |
| 75 | PI 471954 | 781 | Valencia | Valencia | *fastigiata* | Valencia | *fastigiata* | 41.0 | 28.4 | 64.2 | 5.16 | 27.6 |
| 76 | PI 475863 | 87 | Valencia | Valencia | *fastigiata* | Valencia | *fastigiata* | 38.5 | 18.5 | 61.5 | 8.26 | 21.6 |
| 77 | PI 475918 | 605 | . | . | *fastigiata* | Valencia | . | 37.3 | 12.0 | 58.9 | 9.71 | 18.2 |
| 78 | PI 475931 | 610 | Virginia | Virginia | *fastigiata* | Valencia | *fastigiata* | . | 29.7 | 57.6 | −1.14 | 14.5 |
| 79 | PI 476025 | 711 | . | . | *fastigiata* | Valencia | . | 56.7 | 20.3 | 54.6 | 0.23 | 12.4 |
| 80 | PI 476432 | 703 | Intermediate | Mixed | *hypogaea* | Spanish | . | . | 7.2 | 58.1 | 10.68 | 16.2 |
| 81 | PI 476596 | . | . | Runner | . | . | . | . | . | . | . | . |
| 82 | PI 476636 | 678 | Virginia | Virginia | *hypogaea* | Virginia | . | 50.5 | 29.6 | 62.9 | 2.77 | 24.2 |
| 83 | PI 478819 | 650 | Valencia | Valencia | *vulgaris* | Virginia | . | 55.4 | 28.1 | 65.3 | 3.47 | 24.4 |
| 84 | PI 478850 | 747 | . | Valencia | *fastigiata* | Valencia | *peruviana* | 33.4 | 8.2 | 56.5 | 12.64 | 17.8 |
| 85 | PI 481795 | 673 | Spanish | Spanish | *hypogaea* | Spanish | . | 34.8 | 27.2 | 66.5 | 4.43 | 25.4 |
| 86 | PI 482120 | 775 | . | Spanish | *hypogaea* | . | . | 36.6 | 27.0 | 67.3 | 5.44 | 27.0 |
| 87 | PI 482189 | 755 | Spanish | Spanish | *fastigiata* | Valencia | *fastigiata* | 41.0 | 28.6 | 67.0 | 3.82 | 26.0 |
| 88 | PI 493329 | 12 | Valencia | Valencia | *fastigiata* | Valencia | *fastigiata* | 40.9 | 26.0 | 61.3 | 3.68 | 21.8 |
| 89 | PI 493356 | 16 | Virginia | Virginia | *fastigiata* | Valencia | *fastigiata* | 34.2 | 9.7 | 57.6 | 10.57 | 17.4 |
| 90 | PI 493547 | 33 | Valencia | Valencia | *fastigiata* | . | . | 37.6 | 11.1 | 55.8 | 9.92 | 17.4 |
| 91 | PI 493581 | 38 | Valencia | Valencia | *fastigiata* | Valencia | *fastigiata* | 39.7 | 12.2 | 58.0 | 9.27 | 17.8 |
| 92 | PI 493631 | 41 | Valencia | Valencia | *fastigiata* | . | . | 40.6 | 9.2 | 53.6 | 12.25 | 17.7 |
| 93 | PI 493693 | 47 | Virginia | Virginia | *fastigiata* | Valencia | *fastigiata* | 53.5 | 28.1 | 67.8 | 4.34 | 26.6 |
| 94 | PI 493717 | 50 | Valencia | Valencia | *fastigiata* | . | . | 52.0 | 26.9 | 66.7 | 5.85 | 27.4 |
| 95 | PI 493729 | 53 | . | . | *fastigiata* | . | . | 40.1 | 29.4 | 68.1 | 3.30 | 26.2 |
| 96 | PI 493880 | 68 | Valencia | Valencia | *fastigiata* | . | . | 50.7 | 5.1 | 54.5 | 15.37 | 17.4 |
| 97 | PI 493938 | 75 | . | . | *fastigiata* | . | . | 33.5 | 27.1 | 55.7 | −1.34 | 12.5 |
| 98 | PI 494018 | 80 | . | . | *vulgaris* | . | . | 35.2 | 29.4 | 50.8 | −1.73 | 11.4 |
| 99 | PI 494034 | 82 | Spanish | Spanish | *vulgaris* | . | . | 33.0 | 29.1 | 54.7 | −1.39 | 12.9 |
| 100 | PI 494795 | 166 | Runner | Runner | *hypogaea* | . | . | . | 31.2 | 67.1 | −0.85 | 18.5 |
| 101 | PI 496401 | 115 | Virginia | Virginia | *hypogaea* | . | . | 45.7 | 27.0 | 62.3 | 4.49 | 24.3 |
| 102 | PI 496448 | 119 | Virginia | Virginia | *hypogaea* | . | . | 47.5 | 26.2 | 63.3 | 5.67 | 25.6 |
| 103 | PI 497318 | 92 | . | . | *hypogaea* | Valencia | . | 42.9 | 22.4 | 59.3 | 3.03 | 17.7 |
| 104 | PI 497395 | 97 | Virginia | Virginia | *hypogaea* | . | . | . | 6.3 | 53.5 | 9.09 | 14.3 |
| 105 | PI 497517 | 112 | Valencia | Valencia | *fastigiata* | . | . | 37.5 | 9.8 | 56.6 | 11.11 | 17.7 |
| 106 | PI 497639 | 132 | Valencia | Valencia | *fastigiata* | Valencia | *fastigiata* | . | 32.3 | 69.2 | 2.46 | 28.7 |
| 107 | PI 497668 | . | . | . | . | Valencia | *peruviana* | . | . | . | . | . |
| 108 | PI 502037 | . | . | . | . | Valencia | *peruviana* | . | 35.2 | 65.4 | −1.11 | 21.3 |
| 109 | PI 502040 | 149 | Spanish | Spanish | *fastigiata* | . | . | 24.9 | 31.6 | 68.0 | 2.17 | 26.7 |
| 110 | PI 502111 | 155 | . | Valencia | *peruviana* | Valencia | *peruviana* | . | 23.5 | 55.4 | 5.05 | 20.5 |
| 111 | PI 502120 | 157 | . | Virginia | *peruviana* | . | . | 53.8 | 29.1 | 66.1 | 3.76 | 26.2 |
| 112 | PI 504614 | 125 | . | Mixed | *hypogaea* | Virginia | *hypogaea* | 53.7 | 28.5 | 67.3 | 3.79 | 25.8 |

1–[43]; 2–[41]; 3–[48]; 4–Germplasm Resource Information Network (GRIN); 5–Kernel colors were derived as a part of this study using BreedPix tool of the CIMMYT maize scanner using kernel pictures in GRIN database. The dot in Table 1 means missing values or information not found.

### 2.2. Experiment Information

The experiment was conducted at Virginia Tech's Tidewater Agricultural Research and Extension Center (TAREC) in Suffolk, VA (latitude 36.66 N, longitude 76.73 W). Based on seed availability, 93 mini-core accessions and 11 check cultivars were planted in 2017, and 81 accessions and 7 check cultivars were planted in 2018. The checks were: 'Wynne' [53], 'Walton' [54], 'TAMVal OL14' (TVOL14) [55], 'Tamspan 90' (TS90) [56], 'Tamrun OL11' (TROL11) [57], 'New Mexico Valencia' (NMVal) [58], 'C76-16', 'Southwest Runner' (SWR) [59], 'Sullivan' [53], 'OLé' [60], and 'Georgia-09B' (GA09B) [61]. In 2017, mini-core peanut accessions included 48 *hypogaea*, 23 *fastigiata*, 31 *vulgaris*, and two *peruviana* varieties; in 2018, 41 *hypogaea*, 21 *fastigiata*, 24 *vulgaris*, and two *peruviana* accessions were planted [48]. Seeds were planted in two-row plots of 3.05 m long × 0.9 m wide, in a randomized complete block design (RCBD) with three replications. The size of each block was 37.3 m long by 13.7 m wide. Genotypes were planted on 15 May 2017 and 13 May 2018, on uniformly raised beds of 15 cm height, with one seed planted every 11 cm in the center of the bed. Approximately 55 seeds were planted per plot; however, a few accessions had limited seeds and, for these, fewer than 20 seeds per plot were planted. In 2017, 95% of the accessions and checks were provided by USDA-ARS in Stillwater, OK; 5% of the accessions were from Texas. In 2018, seed for all accessions were from Oklahoma. Cultural practices were performed following extension recommendations [62]. Plots were not irrigated.

Weather data were recorded using on-site weather station (WatchDog 2000 Series Weather Station). Rainfall, air temperature, and relative humidity (RH) were recorded daily starting from May 1 until September 30 (Figure 1). Daily growing degree days (GDD$_{13}$) were calculated from min and max daily temperatures, using a base temperature of 13 °C. Only positive values were used, and negative values were recorded as 0. Similarly, the temperatures above 35 °C were taken as 35 °C. Cumulative GDD$_{13}$ from 1 May to 30 September were computed from the daily GDDs for both growing seasons.

### 2.3. Traditional, Ground-Based Phenotyping

Stand counts were collected at 2 weeks after planting (WAP) by counting the total number of peanut plants in both rows of every plot (Table 2).

**Table 2.** Peanut crop growth stages, with respect to weeks after planting (WAP), when the measurements were taken.

| WAP | Crop Growth Stages | Measurements Taken | |
|---|---|---|---|
| | | **2017** | **2018** |
| 0 | Planting (15 May 2017 and 13 May 2018) | | |
| 2 | Emergence | Stand count | Stand count |
| 3 | | | Thrips damage |
| 4 | Vegetative stage | Plant height, lateral growth, aerial measurements | Plant height, lateral growth, NDVI, aerial measurements |
| 5 | | Plant height, lateral growth, CTD, wilting | Plant height, lateral growth, NDVI, CTD, wilting |
| 6 | Beginning bloom | Plant height, lateral growth, NDVI, aerial measurements | Plant height, lateral growth, NDVI, aerial measurements |
| 7 | Beginning peg | Wilting, CTD | Wilting, CTD |
| 8 | Beginning pod | Aerial measurements | NDVI, aerial measurements |
| 9 | Pod development | Plant height, NDVI | Plant height, NDVI |
| 10 | Full pod | NDVI, CTD, wilting, disease rating, aerial measurements | NDVI, CTD, wilting, disease rating, aerial measurements |
| 11 | Beginning seed | | |
| 12 | | NDVI, CTD, disease rating, wilting, aerial measurements | NDVI, CTD, disease rating, wilting, aerial measurements |
| 13 | Full seed | | |
| 14 | Beginning maturity | Aerial measurements | Aerial measurements |
| 15 | | | |
| 16 | Digging (15 September 2017 and 17 September 2018) | | |
| 17 | Post-digging | Pod yield measurements, post-harvest sprouting | Pod yield measurements, post-harvest sprouting |

Growth stages were evaluated according to [63].

Plant height was collected weekly between 4 WAP and 9 WAP, randomly from every row, and the length of the main stem from the ground to the tip of the newest leaf of one randomly selected plant per row was recorded. Plant height values from two rows were averaged to obtain the plant height of each plot. Similar to plant height, lateral branch growth was measured from one randomly selected plant within each row, and the two rows of a plot were averaged for the lateral growth of the plot. Plants from end of plots were avoided.

The normalized difference vegetation index (NDVI) of each row was measured using a GreenSeeker Handheld Crop Sensor (Trimble Ag., Sunnyvale, CA, USA). The GreenSeeker was scanned over the foliage of the entire row at a height of 50 cm and NDVI from both rows of each plot were averaged. NDVI was measured every two weeks from 4 WAP to 12 WAP (full seed stage), for a total of four assessments in 2017, and seven in 2018.

The canopy temperature depression (CTD) of each row was measured using an AGRI-THERM II™ (Model 100 L) Infrared Thermometer. The "diff" option was selected and the CTD value was calculated by subtracting the canopy temperature from the ambient air temperature. CTD was measured over a random spot on each row, and values from two rows were averaged for plot CTD. As CTD is sensitive to wind and intermittent cloud

covers, data were collected on sunny days with minimal wind. CTD was measured from 5 to 12 WAP, for a total of four assessments each year.

Leaf wilting was visually assessed using the following 0–5 rating scale: 0, healthy plant with no visible wilting or leaves drooping; 1, some terminal and newer leaves wilted but overall the plant looked healthy; 2, almost all upper leaves with visible signs of wilting, and lower and older leaves started to fold; 3, all leaves wilting and drooping, drought effect on older leaves was prominent, and bare ground starting to become visible; 4, all leaves wilted and some leaves started to change color due to chlorophyll degradation, bare ground prominently visible, some leaves dried and crisped; 5, all leaves were severely wilted and light green to yellow in color, bare ground fully visible, more than 50% of leaves desiccated, and the plant almost physiologically dead [64]. Leaf wilting was measured from 5 to 12 WAP, for a total of four assessments per year (Table 2).

Disease incidence, the percentage of diseased plants exhibiting symptoms of TSWV, SSR, SB, and CBR, in each plot, was rated at 10 and 12 WAP each year. The percentage was calculated as a fraction of the number of diseased plants observed, to the number of plants in each plot. Thrips (*scirtothrips dorsalis*) damage was rated using a scale from 0 to 10, with 0 being a plant not damaged by thrips and 10 being all leaves damaged [65].

At the physiological maturity (16 WAP), peanut pods were dug (15 September 2017, and 17 September 2018) using a Sweere C200 peanut digger, windrow dried for 7 days and combined using an Amadas 2110 two row peanut combine. For each plot, pod yield was calculated at 7% seed moisture. Peanut sprouting was evaluated 7 days after digging by counting the number of germinated seeds on the ground.

*2.4. Aerial Data Collection*

Aerial images were taken every 2 weeks starting at 4 WAP to 14 WAP for estimation of leaf reflectance and color space indices (Table 3). An AscTec® Falcon 8 octocopter UAV platform (Ascending Technologies, Krailling, Germany), equipped with an RGB digital camera [Sony® α6000, 24.3-megapixel, (6000 × 4000)] and a near infra-red (NIR) camera [Tetracam® ADC micro, 3.2-megapixel, (2048 × 1536)] was used. The flight campaign was in waypoint navigation, auto pilot, and at 20 m altitude with an image overlap of 75% forward and 90% sideways. Flight campaigns were created in AscTec®Navigator 3.4.5 software (Ascending Technologies, Krailling, Germany). The UAV used its built-in GPS (accuracy within 20 cm) to navigate, acquire nadir images, and coordinate recordings of individual images. Image orthomosaic was processed using Pix4Dmapper Version 4.2.26 software (Prilly, Switzerland) to create a RGB field map. The 'reflectance map' option in 'index calculator' under 'DSM, orthomosaic, and index' step of Pix4D processing was used to create individual red, green, and blue reflectance maps. The same settings were used for NIR orthomosaic to create an NIR reflectance map, and an additional 'reflectance map' option was used to generate the NDVI orthomosaic using red and NIR from the NIR images.

**Table 3.** Spectral reflectance (red, green, blue, and near-infrared) derived using aerial images, and vegetation indices derived using reflectance (S. No. 1–24); and red-green-blue (RGB) color space indices derived from the same images using Breedpix software and indices derived using arithmetic combinations of color indices (S. no. 24–48).

| S. No. | Indices | Full Name | Formula | Reference |
|--------|---------|-----------|---------|-----------|
| 1 | | Red | | |
| 2 | | Green | Aerial leaf reflectance | |
| 3 | | Blue | | |
| 4 | | Near-Infrared (NIR) | | |
| 5 | BGI | Blue green pigment index | $\frac{Blue}{Green}$ | [66] |
| 6 | RGR | Red-Green ratio | $\frac{Red}{Green}$ | [67] |
| 7 | NPPR | Normalized Plant Pigment ratio | $\frac{Green}{Red+Blue}$ | [68] |
| 8 | NGRDI | Normalized Green Red Difference Index | $\frac{Green-Red}{Green+Red}$ | [69] |
| 9 | PPR | Plant Pigment Ratio | $\frac{Green-Blue}{Green+Blue}$ | [70] |
| 10 | NCPI | Normalized Pigment Chlorophyll Index | $\frac{Red-Blue}{Red+Blue}$ | [71] |
| 11 | NDVI | Normalized difference vegetation index | $\frac{NIR-Red}{NIR+Red}$ | [72] |
| 12 | SRI | Simple ratio index | $\frac{NIR}{Red}$ | [73] |
| 13 | GRVI | Green Ratio Vegetation Index | $\frac{NIR}{Green}$ | [74] |
| 14 | IO | Simple Ratio Red/Blue Iron Oxide | $\frac{Red}{Blue}$ | [75] |
| 15 | GNDVI | Green Normalized difference vegetation index | $\frac{NIR-Green}{NIR+Green}$ | [76] |
| 16 | BNDVI | Blue Normalized difference vegetation index | $\frac{NIR-Blue}{NIR+Blue}$ | [77] |
| 17 | CIG | Chlorophyll index green | $\frac{NIR}{Green}-1$ | [78] |
| 18 | CVI | Coloration index | $\frac{Red-Blue}{Red}$ | [79] |
| 19 | GLI | Green leaf index | $\frac{2\times Green-(Red+Blue)}{2\times Green+(Red+Blue)}$ | [80] |
| 20 | GBNDVI | Green-Blue NDVI | $\frac{NIR-(Green+Blue)}{NIR+(Green+Blue)}$ | [81] |
| 21 | GRNDVI | Green-Red NDVI | $\frac{NIR-(Green+Red)}{NIR+(Green+Red)}$ | [81] |
| 22 | RBNDVI | Red-Blue NDVI | $\frac{NIR-(Green-(Blue-Red))}{NIR-(Green+(Blue-Red))}$ | [81] |
| 23 | mSR | Modified Simple Ratio | $\frac{NIR-Blue}{Red+Blue}$ | [82] |
| 24 | GARI | Green atmospherically resistant vegetation index | $\frac{NIR-(Blue+Red)}{NIR+(Blue+Red)}$ | [76] |
| 25 | | Intensity | Measures greyness in 0 (black) to 1 (white) scale in HSI color space | [83] |
| 26 | | Hue | Color judgement (in °) based on position in HSI color space | [83] |
| 27 | | Saturation | Measures dilution of pure color (hue) with white light within 0 to 1 | [83] |
| 28 | | Lightness | Light reflected by a non-luminous body [0 (black) to 100 (white)] | [84] |
| 29 | | a* | color shift from green (−a) to red (+a) in CIE-Lab color space | [84] |
| 30 | | b* | color shift from blue (−b) to yellow (+b) in CIE-Lab color space | [84] |
| 31 | | u* | color shift from green (−a) to red (+a) in CIE-Luv color space | [84] |
| 32 | | v* | color shift from blue (−b) to yellow (+b) in CIE-Luv color space | [84] |

**Table 3.** *Cont.*

| S. No. | Indices | Full Name | Formula | Reference |
|--------|---------|-----------|---------|-----------|
| 33 | GA | Green area | Percentage of pixels in 60°–120° hue angle in CIE-Lab | [85] |
| 34 | GGA | Greener area | Percentage of pixels in 80°–120° hue angle in CIE-Lab | [85] |
| 35 | CSI | Crop senescence index | $\frac{100 \times (GA - GGA)}{GA}$ | [86] |
| 36 | ab | | $a^* \times b^*$ | [28] |
| 37 | uv | | $u^* \times u^*$ | [28] |
| 38 | abI | ab Index | $\frac{a^*}{b^*}$ | |
| 39 | uvI | uv Index | $\frac{u^*}{v^*}$ | |
| 40 | auI | au Index | $\frac{a^*}{u^*}$ | |
| 41 | bvI | bv Index | $\frac{b^*}{v^*}$ | |
| 42 | NDabI | Normalized difference ab Index | $\frac{b^* - a^*}{b^* + a^*}$ | |
| 43 | NDuvI | Normalized difference uv Index | $\frac{v^* - u^*}{v^* + u^*}$ | |
| 44 | NDLab | Normalized difference CIELab Index | $\frac{1 - a^* - b^*}{1 - a^* + b^*}$ | [87] |
| 45 | NDLuv | Normalized difference CIELuv Index | $\frac{1 - u^* - v^*}{1 - u^* + v^*}$ | [87] |
| 46 | GI | Greenness Index | $\frac{GA}{GGA}$ | |
| 47 | GPI | Greenness product index | $GA \times GGA$ | |
| 48 | NDGI | Normalized difference greenness Index | $\frac{GA - GGA}{GA + GGA}$ | |

Red, green, blue, NIR, and NDVI orthomosaics were exported to ArcMap (version 10.6) tool of ArcGIS (ESRI, Redlands, CA, USA). Polygons bordering every row were drawn on the orthomosaic. Each polygon had the same dimension as that of each row (3.05 m long × 0.9 m wide) and was numbered. Polygons were shifted to overlap the respective plot rows and collated into a single shapefile to create a fishnet. Fishnets were common for all images from every flight campaign with georeferencing. Georeferencing was done using GPS coordinates of pre-installed ground control points (GCPs) on the study field. The zonal statistics option was used to extract the digital numbers (DNs) of each row. This process averaged the raster information of every pixel within each polygon to give the DN of red, green, blue, and NIR rasters.

Calibration was performed using a reflectance panel with eight different shades, from white to black. The DNs of the eight shades were recorded for red, green, blue, and NIR rasters from each orthomosaic. On the day of every flight, the reflectance from each of the eight shades of the panel were measured using an ASD HH2 Hand-held VNIR Spectroradiometer (Malvern Panalytical, Malvern, UK). The DNs and reflectance from the panel were fitted in exponential regression models. The models trained to derive reflectance values from DNs were:

$$\text{Equation A} - red = 0.1263 \times 1.0091^{DNr}$$
$$\text{Equation B} - green = 0.1263 \times 1.0087^{DNg}$$
$$\text{Equation C} - blue = 0.1144 \times 1.0087^{DNb}$$
$$\text{Equation D} - NIR = 0.0563 \times 1.0147^{DNn}$$

where *red*, *green*, *blue*, and *NIR* are the reflectance from the respective rasters; and $DN_r$, $DN_g$, $DN_b$, and $DN_n$, are the digital numbers from red, green, blue, and *NIR* rasters, respectively. The reflectance values of both rows of each plot were averaged to obtain the average reflectance value of the plot.

### 2.5. Calculation of the VIs

A total of 48 VIs were extracted, or calculated, and their definitions are presented in Table 3. Twenty-four VIs were calculated using the four bands (red, green, blue, near-infrared) of leaf reflectance. The VIs were selected based on their power to discriminate among healthy, stressed, and dead vegetation (https://www.indexdatabase.de, accessed on 3 March 2020) [66,70,71].

An additional 24 VIs were extracted, or computed, from the color space indices extracted from the RGB orthomosaic using ArcMap. Individual rows (two rows per plot) were used to extract eight RGB color space indices using BreedPix tool. The extracted indices were intensity, hue, saturation, lightness, a*, b*, u*, and v*. The green area (GA), greener area (GGA), crop senescence index (CSI), along with 13 other indices, were computed from the extracted indices, as shown in Table 3 [88–92].

*2.6. Data Analyses*

Statistical analyses were performed in Statistical Analysis Software (SAS) 9.4 (SAS Institute Inc., Cary, NC, USA). PROC GLM was used for analysis of variance (ANOVA). Measurements collected multiple times (plant height, lateral growth, NDVI, CTD, leaf wilting, and disease ratings) were analyzed as repeated measures ANOVA using "nouni" command and repeated option in PROC GLM. Fisher's protected least significant difference (LSD) was used for mean separation, when appropriate, based on the number of levels in a particular factor. When designs were unbalanced, least square means (LSmeans) mean separation procedure adjusted for Student's *t*-test was used. PROC CORR was used for Pearson's correlation analysis. All image derived VIs and color space indices were correlated to ground based traits. Pearson's correlation was performed separately for image and ground traits of each botanical variety. Since only two genotypes of variety *peruviana* were included, it was pooled along with *vulgaris*, owing to their morphological similarities (Stalker, 2017). PROC CORR was further used to create Pearson's correlation matrix heatmap. Graphs and figures were built using JMP® Pro 15.0.0 (SAS Institute Inc., Cary, NC, USA).

*Calculation of the broad sense heritability*: Broad sense heritability (H$^2$) was calculated as the ratio of genotypic variance ($\sigma_G^2$) by phenotypic variance ($\sigma_P^2$). Variance was calculated as the ratio of the total sum of squares (TSS) to population size (n). H$^2$ was calculated for all ground based and aerially derived traits.

$$H^2 = \frac{\sigma_G^2}{\sigma_G^2 + \frac{\sigma_G^2\sigma_E^2}{E} + \frac{\sigma_E^2}{ER}}$$

here: $\sigma_P^2 = \sigma_G^2 + \frac{\sigma_G^2\sigma_E^2}{E} + \frac{\sigma_E^2}{ER}$

$\sigma_E^2$ = environmental variance;
$E$ = number of environments;
$R$ = number of replications

**3. Results**

*3.1. ANOVA of Genotype and Variety*

In both years, repeated measures ANOVA showed significant WAP × genotype, and WAP × variety interaction, for plant height, lateral growth, leaf wilting, ground NDVI, and CTD. Therefore, with factorial ANOVA reported in Table 4, and trait means reported in Tables 5 and 6, averages of all WAP were used only for the traits for which these interactions were not significant, i.e., TSW, SSR, SB, and CBR. For plant height, lateral branching, and NDVI, data at 6 WAP were used. Based on our visual observations, 6 WAP coincided with the maximum point of the rapid growth phase before growth rate slowed and, even though continued through 10 WAP, steadied; 6 WAP also marked the end of vegetative and the beginning of the generative growth stage. For wilting and CTD, average values from 10 and 12 WAP in 2017, and 5 and 7 WAP in 2018, were used. During these times, sudden droughts were encountered, and plants experienced low moisture stress; the interactions of these WAP with genotype and variety were not significant. In this way, ANOVA showed that year, genotype, and their interactions, had significant effects on morphological, physiological, and agronomic characteristics, measured on the ground (Table 4). Stand count, plant height, lateral branching, leaf wilting, TSW, SSR, post-harvest sprouting, and pod yield, varied

significantly ($p < 0.05$) among genotypes ($p < 0.05$ to $p < 0.0001$) (Table 4). Ground NDVI was significant at $p < 0.1$ ($p = 0.074$). Among the varieties, only plant height in 2017, wilting in both years, and sprouting in 2018 were significantly different at $p < 0.05$; pod yield, along with stand count and SB, were significant at $p < 0.1$ (Table 4).

**Table 4.** a: Analysis of variance for the effect of genotype, year, and their interaction on morphological, physiological, and agronomic characteristics, measured in 2017 and 2018 on the U.S. peanut germplasm mini-core collection. b: Analysis of variance for the effect of variety, year, and their interaction on morphological, physiological, and agronomic characteristics, measured in 2017 and 2018 on the U.S. peanut germplasm mini-core collection.

| Source of Variation | | Stand Count | Plant Height | Lateral Growth | NDVI | CTD | Wilting | TSW | SSR | SB | CBR | Thrips Damage | Pod Yield | Sprouting |
|---|---|---|---|---|---|---|---|---|---|---|---|---|---|---|
| | DF | | | | | | | *p*-Value | | | | | | |
| year | 1 | <0.0001 | <0.0001 | <0.0001 | <0.0001 | <0.0001 | <0.0001 | <0.0001 | <0.0001 | <0.0001 | <0.0001 | <0.0001 | <0.0001 | <0.0001 |
| block | 2 | 0.001 | 0.110 | <0.0001 | 0.703 | <0.0001 | <0.0001 | 0.0003 | <0.0001 | 0.619 | 0.009 | 0.248 | 0.211 | <0.0001 |
| genotype | 102 | <0.0001 | <0.0001 | <0.0001 | 0.074 | 0.114 | <0.0001 | <0.0001 | 0.039 | 0.232 | 0.329 | 0.678 | <0.0001 | 0.005 |
| year*genotype | 87 | <0.0001 | <0.0001 | 0.0041 | 0.461 | 0.003 | 0.366 | 0.409 | 0.011 | 0.049 | 0.658 | 0.303 | <0.0001 | 0.766 |
| Error | 381 | | | | | | | | | | | | | |
| year | 1 | 0.164 | 0.061 | <0.0001 | <0.0001 | <0.0001 | <0.0001 | <0.0001 | <0.0001 | <0.0001 | <0.0001 | <0.0001 | <0.0001 | 0.005 |
| block | 2 | 0.193 | 0.236 | <0.0001 | 0.701 | <0.0001 | 0.0016 | 0.001 | <0.0001 | 0.619 | 0.01 | 0.249 | 0.425 | <0.0001 |
| variety | 3 | 0.090 | 0.0003 | 0.925 | 0.2159 | 0.527 | <0.0001 | 0.406 | 0.454 | 0.092 | 0.839 | 0.512 | 0.096 | 0.11 |
| year*variety | 3 | 0.092 | 0.013 | 0.129 | 0.6538 | 0.315 | 0.9781 | 0.262 | 0.427 | 0.097 | 0.668 | 0.512 | 0.283 | 0.976 |
| Error | 566 | | | | | | | | | | | | | |

TSW–tomato spotted wilt; SSR–southern stem rot; SB–sclerotinia blight; CBR-cylindrocladium black rot.

### 3.2. Mean Separation of Genotypes

Even though fresh seed was produced for each planting year, several genotypes had insufficient seed and this resulted in significantly poor stand in 2017 for some accessions; for this reason, these accessions were removed from the test in 2018 (Table 5). Plant height and lateral growth at 6 WAP was significantly different ($p < 0.0001$) within genotypes (Table 5). In 2017, genotypes CC760 and CC605 were the tallest (42.5 and 40.6 cm, respectively) and had the most lateral growth (65.4 cm and 64.1 cm, respectively); whereas CC115B and CC631 were the shortest (12.3 cm and 16.1 cm tall, and 23.5 cm and 27.7 cm lateral growth, respectively) (Table 5). Genotypes CC588 and CC760 showed severe leaf wilting (around three) in both years, whereas CC208, CC223, CC296, CC342, CC381, CC458, CC535, CC548, CC559, CC698, CC703B, CC812, and Wynne, were least wilted (<2) in both years. Incidence of TSWV and SSR differed significantly among the genotypes. In both years, CC053 and CC781 had one of the highest incidences of TSWV, whereas Wynne and Walton had the lowest. For SSR, there were no differences among genotypes in 2017; however, in 2018, CC781 showed the highest and CC787 the lowest incidences of SSR (Table 6). There were no differences for SB, CBR, and thrips damage incidence, among genotypes in 2017 and 2018. Pod yield between years were significantly different ($p < 0.0001$) within genotypes with Wynne (8253 kg ha$^{-1}$ in 2017 and 6276 kg ha$^{-1}$ in 2018), Walton (8459 kg ha$^{-1}$ in 2017 and 6915 kg ha$^{-1}$ in 2018), and C76-16 (8407 kg ha$^{-1}$ in 2017 and 4753 kg ha$^{-1}$ in 2018) performing among the best in both years (Table 5). Average pod yield was significantly lower in 2018 (2886 kg ha$^{-1}$), as compared to 2017 (5334 kg ha$^{-1}$). Post-harvest sprouting showed no significant difference among genotypes in 2017; however, in 2018, CC038 had the highest post-harvest sprouting (51.7 seeds per plot [5.5 m$^2$]) (Table 5).

### 3.3. Mean Separation of Varieties

Among the varieties, *vulgaris* was the tallest and had the most robust lateral growth (Table 7). Variety *peruviana* had the highest wilting score, followed by *vulgaris* and *fastigiata*; variety *hypogaea* had the least wilting score in both years. There were no differences among botanical varieties for disease incidence, or thrips damage. Although variety *hypogaea* had significantly higher yield in 2017 than other varieties, no yield differences were observed among varieties in 2018. Post-harvest sprouting was higher for *fastigiata* and *vulgaris* in 2018, than in *hypogaea* and *peruviana* (Table 7).

**Table 5.** Plant growth and yield parameters (stand count, plant height, lateral growth, normalized ifference vegetation index (NDVI), canopy temperature depression (CTD), leaf wilting, pod yield, and post-harvest sprouting) of 104 mini-core genotypes of peanut in 2017, and 88 in 2018. The plant height, lateral growth, and NDVI, are measured at maximum vegetative growth [6 weeks after planting (WAP). Leaf wilting and canopy temperature depression (CTD) are the average of two dates (10 and 12 WAP in 2017 and 5 to 7 WAP in 2018), with highest values corresponding to sudden droughts. The values followed by the same letters are not significantly different using Fisher's protected LSD at $\alpha$ = 0.05.

| Genotypes | Stand Count (Plants/Plot) 2017 | 2018 | Plant Height (cm) 2017 | 2018 | Lateral Growth (cm) 2017 | 2018 | NDVI (0–1) 2017 | 2018 | CTD (°C) 2017 | 2018 | Leaf Wilting (0–5) 2017 | 2018 | Pod Yield (kg ha$^{-1}$) 2017 | 2018 | Sprouting (#/Plot) 2017 | 2018 |
|---|---|---|---|---|---|---|---|---|---|---|---|---|---|---|---|---|
| Wynne | 33 k-s | 16 j | 24 h-t | 26 a | 49 a-k | 67 a | 0.90 a-d | 0.69 a | −0.8 e-u | 3.2 a-d | 1.2 f-j | 1.5 h | 8253 a-d | 6276 a | 0 a | 0 h |
| Walton | 40 c-q | 36 c-i | 24 f-t | 28 a | 53 a-j | 68 a | 0.89 a-d | 0.74 a | −0.4 b-q | 1.8 d-r | 1.2 f-j | 2.0 b-h | 8459 a | 6915 a | 2 a | 0 h |
| TVOL14 | 47 a-m | 50 a-g | 32 a-p | 31 a | 51 a-k | 64 a | 0.89 a-d | 0.69 a | −0.1 a-k | 1.0 qr | 1.6 c-j | 2.7 a-h | 5915 a-j | 2424 f-s | 3 a | 4 gh |
| TS90 | 49 a-j | 52 a-e | 30 a-r | 29 a | 51 a-k | 72 a | 0.88 a-d | 0.75 a | −0.6 b-t | 2.5 b-o | 2.1 a-i | 3.0 a-h | 7056 a-h | 2702 c-r | 6 a | 5 gh |
| TROL11 | 32 l-s | 43 a-i | 22 j-t | 23 a | 46 a-l | 58 a | 0.85 cd | 0.67 a | 0.5 a-c | 3.6 a-c | 1.1 g-j | 2.9 a-h | 5999 a-j | 3260 c-m | 5 a | 8 e-h |
| NMVAL | 53 a-e | 52 a-c | 34 a-l | 28 a | 52 a-j | 59 a | 0.87 a-d | 0.75 a | −0.4 b-q | 1.4 k-r | 2.1 a-j | 3.3 a-e | 4667 a-j | 1627 rs | 1 a | 28 b-d |
| CC812 | 47 a-m | 39 b-i | 28 b-s | 27 a | 61 a-e | 58 a | 0.88 a-d | 0.76 a | 0.2 a-f | 2.2 c-q | 1.1 g-j | 1.7 f-h | 6214 a-j | 3766 b-g | 1 a | 0 h |
| CC808 | 53 a-e | 53 a-c | 33 a-m | 28 a | 51 a-k | 62 a | 0.87 a-d | 0.78 a | 0.8 a | 1.8 f-r | 2.4 a-e | 2.8 a-h | 4829 a-j | 2874 c-r | 3 a | 13 c-h |
| CC805 | 49 a-j | 37 c-i | 24 h-t | 31 a | 52 a-j | 64 a | 0.88 a-d | 0.77 a | −0.8 f-u | 2.6 b-n | 1.1 g-j | 2.1 b-h | 5481 a-j | 2860 c-r | 1 a | 0 h |
| CC802 | 52 a-g | 51 a-f | 28 b-s | 26 a | 48 a-k | 63 a | 0.89 a-d | 0.79 a | −0.8 e-u | 2.4 c-p | 1.3 e-j | 2.7 a-h | 4768 a-j | 2702 c-r | 3 a | 7 f-h |
| CC798 | 51 a-h | 50 a-f | 31 a-r | 28 a | 50 a-k | 55 a | 0.89 a-d | 0.80 a | −1.7 tu | 2.8 a-k | 1.5 c-j | 2.8 a-h | 4911 a-j | 4004 bc | 3 a | 9 e-h |
| CC787 | 47 a-m | 43 a-i | 38 a-g | 28 a | 59 a-f | 64 a | 0.87 a-d | 0.75 a | −0.2 a-n | 1.7 h-r | 2.0 a-i | 3.0 a-h | 4640 b-j | 2311 i-s | 4 a | 2 gh |
| CC781 | 50 a-i | 53 a-c | 37 a-h | 28 a | 61 a-d | 64 a | 0.88 a-d | 0.71 a | −0.3 a-q | 2.5 b-o | 2.4 a-e | 3.3 a-d | 5047 a-j | 2602 d-s | 1 a | 28 b-d |
| CC775 | 37 f-r | 38 c-i | 32 a-o | 28 a | 52 a-j | 66 a | 0.88 a-d | 0.71 a | −0.4 b-q | 3.8 ab | 1.5 c-j | 2.2 a-h | 4476 d-j | 1747 p-s | 2 a | 4 gh |
| CC760 | 49 a-j | 42 a-i | 43 a | 27 a | 65 a | 71 a | 0.86 a-d | 0.66 a | −0.2 a-m | 1.1 p-r | 2.7 a-c | 3.6 a | 5902 a-j | 3147 c-o | 5 a | 30 bc |
| CC755 | 52 a-g | 47 a-i | 36 a-i | 25 a | 60 a-e | 62 a | 0.86 b-d | 0.75 a | 0.6 ab | 2.8 a-k | 2.3 a-g | 3.3 a-g | 4550 c-j | 2425 f-s | 1 a | 4 gh |
| CC740 | 50 a-h | 43 a-i | 27 d-s | 25 a | 42 a-l | 58 a | 0.88 a-d | 0.71 a | −1.4 p-u | 1.4 l-r | 2.0 a-j | 2.4 a-h | 6423 a-i | 2964 c-r | 6 a | 27 b-e |
| CC725 | 50 a-h | 51 a-e | 38 a-f | 26 a | 49 a-k | 62 a | 0.88 a-d | 0.74 a | −0.8 f-u | 2.3 c-q | 1.3 e-j | 2.6 a-h | 5176 a-j | 2776 c-r | 8 a | 28 b-d |
| CC711 | 38 e-r | 32 h-j | 27 b-s | 28 a | 46 a-l | 68 a | 0.88 a-d | 0.74 a | −0.2 a-n | 2.9 a-j | 1.7 b-j | 2.3 a-h | 4094 f-j | 2284 i-s | 2 a | 0 h |
| CC703B | 39 d-q | 50 a-i | 23 i-t | 26 a | 52 a-j | 57 a | 0.90 a-d | 0.77 a | −1.2 k-u | 1.7 i-r | 1.0 h-j | 1.8 e-h | 6593 a-i | 3711 b-h | 1 a | 0 h |
| CC703A | 48 a-j | 50 a-g | 28 b-s | 27 a | 40 b-l | 59 a | 0.87 a-d | 0.77 a | −1.4 n-u | 1.7 g-r | 2.1 a-i | 2.7 a-h | 5210 a-j | 3386 b-m | 1 a | 0 h |
| CC698 | 23 r-u | 40 a-i | 19 n-t | 28 a | 31 j-l | 57 a | 0.89 a-d | 0.73 a | −0.7 e-u | 2.7 a-m | 0.8 j | 1.8 e-h | 3522 g-j | 3360 c-m | 1 a | 0 h |
| CC678 | 15 t-v | 41 a-i | 18 o-t | 28 a | 36 f-l | 55 a | 0.89 a-d | 0.76 a | −1.0 g-u | 2.4 b-p | 1.0 h-j | 2.0 b-h | 5327 a-j | 3475 b-k | 5 a | 0 h |
| CC673 | 50 a-i | 51 a-e | 32 a-o | 24 a | 48 a-k | 55 a | 0.88 a-d | 0.71 a | −0.7 c-t | 2.7 a-l | 1.4 d-j | 2.3 a-h | 4465 d-j | 3141 c-o | 4 a | 8 e-h |
| CC650 | 48 a-k | 42 a-i | 29 b-s | 29 a | 50 a-k | 56 a | 0.88 a-d | 0.73 a | −1.1 i-u | 2.3 c-q | 1.3 e-j | 2.3 a-h | 8351 a-c | 1772 n-s | 1 a | 5 gh |
| CC643 | 28 p-t | 43 a-i | 28 b-s | 27 a | 52 a-j | 66 a | 0.89 a-d | 0.69 a | −0.3 a-q | 2.2 d-r | 2.1 a-i | 3.3 a-e | 5373 a-j | 2382 g-s | 3 a | 2 gh |
| CC605 | 56 ab | 52 a-d | 41 a-c | 27 a | 64 ab | 59 a | 0.89 a-d | 0.71 a | −0.7 e-u | 2.2 c-q | 2.8 ab | 3.0 a-h | 5650 a-j | 2824 c-r | 3 a | 13 c-h |
| CC588 | 58 a | 47 a-i | 40 a-d | 31 a | 54 a-j | 63 a | 0.88 a-d | 0.66 a | −0.3 a-q | 2.2 c-q | 3.0 a | 3.0 a-h | 4247 f-j | 2327 h-s | 6 a | 30 bc |
| CC580 | 52 a-f | 53 a-c | 35 a-k | 25 a | 53 a-j | 67 a | 0.86 a-d | 0.72 a | −0.6 b-s | 2.9 a-j | 1.8 a-j | 2.9 a-h | 4925 a-j | 2918 c-r | 1 a | 6 gh |
| CC579 | 49 a-j | 49 a-h | 32 a-p | 29 a | 43 a-l | 66 a | 0.87 a-d | 0.67 a | −0.5 b-q | 1.9 d-r | 1.5 c-j | 2.2 a-h | 4815 a-j | 2702 c-r | 4 a | 7 f-h |
| CC559 | 48 a-j | 36 c-i | 24 h-t | 27 a | 37 e-l | 55 a | 0.90 a-d | 0.68 a | −0.6 b-r | 1.8 f-r | 1.3 e-j | 1.8 d-h | 3743 g-j | 2101 k-s | 2 a | 0 h |
| CC553 | 53 a-d | 50 a-g | 33 a-m | 26 a | 47 a-l | 61 a | 0.88 a-d | 0.74 a | −1.7 s-u | 1.8 e-r | 1.1 g-j | 2.1 b-h | 6291 a-j | 2425 f-s | 1 a | 10 d-h |

**Table 5.** *Cont.*

| Genotypes | Stand Count (Plants/Plot) 2017 | 2018 | Plant Height (cm) 2017 | 2018 | Lateral Growth (cm) 2017 | 2018 | NDVI (0–1) 2017 | 2018 | CTD (°C) 2017 | 2018 | Leaf Wilting (0–5) 2017 | 2018 | Pod Yield (kg ha$^{-1}$) 2017 | 2018 | Sprouting (#/Plot) 2017 | 2018 |
|---|---|---|---|---|---|---|---|---|---|---|---|---|---|---|---|---|
| CC548 | 36 h-s | 44 a-i | 26 d-s | 28 a | 56 a-i | 61 a | 0.88 a-d | 0.71 a | −0.3 a-q | 2.3 c-q | 1.0 h-j | 1.5 h | 6446 a-i | 3523 b-j | 0 a | 0 h |
| CC546 | 52 a-e | 51 a-e | 35 a-l | 27 a | 49 a-k | 59 a | 0.87 a-d | 0.71 a | −1.9 u | 2.6 b-n | 2.2 a-h | 2.7 a-h | 4700 a-j | 2354 h-s | 2 a | 9 e-h |
| CC535 | 51 a-h | 33 f-j | 32 a-q | 25 a | 45 a-l | 61 a | 0.90 a-c | 0.70 a | −0.7 d-t | 3.1 a-g | 1.0 h-j | 1.5 h | 6792 a-i | 3161 c-n | 0 a | 0 h |
| CC529 | 47 a-l | 47 a-i | 24 g-t | 26 a | 38 d-l | 63 a | 0.91 a | 0.66 a | −0.2 a-n | 2.7 a-m | 1.3 e-j | 2.1 b-h | 5032 a-j | 2977 c-r | 1 a | 4 gh |
| CC526 | 31 n-s | 56 ab | 26 f-t | 30 a | 44 a-l | 65 a | 0.88 a-d | 0.77 a | 0.2 a-g | 0.8 r | 1.9 a-j | 3.1 a-g | 5480 a-j | 3186 c-m | 1 a | 17 b-h |
| CC508 | 52 a-e | 50 a-g | 38 a-g | 24 a | 54 a-j | 65 a | 0.87 a-d | 0.71 a | −0.7 e-u | 2.4 c-p | 1.8 b-j | 2.8 a-h | 5653 a-j | 2989 c-r | 0 a | 2 gh |
| CC488 | 55 ab | 52 a-d | 40 a-e | 24 a | 53 a-j | 65 a | 0.89 a-d | 0.79 a | −0.2 a-o | 2.3 c-q | 2.3 a-f | 3.3 a-e | 5507 a-j | 2523 e-s | 3 a | 20 b-g |
| CC485 | 39 d-q | 44 a-i | 31 a-r | 29 a | 49 a-k | 58 a | 0.88 a-d | 0.70 a | 0.4 a-e | 1.7 i-r | 2.3 a-g | 2.6 a-h | 4152 f-j | 1628 rs | 2 a | 8 e-h |
| CC481 | 46 a-n | 47 a-i | 31 a-r | 27 a | 51 a-k | 69 a | 0.89 a-d | 0.72 a | 0.1 a-h | 1.4 k-r | 1.6 c-j | 2.7 a-h | 5050 a-j | 3414 b-l | 7 a | 18 b-h |
| CC477 | 50 a-i | 52 a-d | 30 a-r | 29 a | 45 a-l | 65 a | 0.86 a-d | 0.73 a | 0.0 a-h | 2.1 d-r | 1.4 d-j | 2.5 a-h | 4427 e-j | 3857 b-e | 8 a | 17 b-h |
| CC458 | 35 i-s | 47 a-i | 18 p-t | 24 a | 43 a-l | 58 a | 0.90 a-c | 0.74 a | −0.2 a-n | 2.0 d-r | 1.0 h-j | 1.5 h | 4340 e-j | 2931 c-r | 3 a | 0 h |
| CC446 | 50 a-i | 53 a-c | 34 a-l | 25 a | 42 a-l | 61 a | 0.85 d | 0.72 a | −1.2 k-u | 2.5 b-p | 1.8 a-j | 2.3 a-h | 4139 f-j | 2079 l-s | 6 a | 11 d-h |
| CC431 | 50 a-i | 46 a-i | 30 a-s | 27 a | 46 a-l | 58 a | 0.88 a-d | 0.70 a | −0.6 c-t | 3.1 a-i | 1.8 a-j | 2.8 a-h | 4982 a-j | 1758 o-s | 0 a | 14 c-h |
| CC408 | 49 a-j | 49 a-h | 36 a-i | 27 a | 56 a-h | 59 a | 0.88 a-d | 0.73 a | 0.0 a-j | 1.3 m-r | 2.6 a-d | 3.2 a-f | 5881 a-j | 2496 e-s | 5 a | 30 bc |
| CC406 | 51 a-h | 48 a-i | 30 a-s | 28 a | 42 a-l | 66 a | 0.89 a-d | 0.74 a | −0.4 b-q | 2.1 d-r | 1.4 d-j | 2.5 a-h | 5806 a-j | 3600 b-i | 3 a | 3 gh |
| CC388 | 49 a-j | 44 a-i | 29 b-s | 25 a | 48 a-k | 63 a | 0.87 a-d | 0.93 a | 0.3 a-f | 2.7 b-m | 1.9 a-j | 2.8 a-h | 5263 a-j | 2867 c-r | 1 a | 30 bc |
| CC384 | 54 a-c | 48 a-i | 34 a-l | 24 a | 46 a-l | 64 a | 0.88 a-d | 0.60 a | 0.2 a-g | 2.5 b-o | 1.3 e-j | 2.0 b-h | 4341 e-j | 3022 c-q | 5 a | 5 gh |
| CC381 | 46 a-n | 34 e-j | 24 h-t | 29 a | 49 a-k | 60 a | 0.88 a-d | 0.64 a | −1.3 l-u | 3.1 a-h | 1.0 h-j | 1.8 e-h | 6070 a-j | 2563 d-s | 0 a | 0 h |
| CC342 | 46 a-n | 34 e-j | 26 e-t | 31 a | 63 a-c | 70 a | 0.89 a-d | 0.75 a | −0.2 a-o | 2.4 c-p | 1.1 g-j | 1.6 gh | 7030 a-j | 2456 f-s | 1 a | 0 h |
| CC338 | 52 a-f | 46 a-i | 33 a-m | 28 a | 42 b-l | 61 a | 0.88 a-d | 0.76 a | 0.2 a-g | 2.3 c-q | 1.9 a-j | 2.3 a-h | 4770 a-j | 3121 c-p | 1 a | 25 b-f |
| CC310 | 57 a | 49 a-h | 30 a-r | 29 a | 50 a-k | 63 a | 0.87 a-d | 0.72 a | 0.2 a-g | 1.2 o-r | 2.2 a-h | 2.8 a-h | 5483 a-j | 3170 c-m | 2 a | 35 ab |
| CC296 | 47 a-k | 43 a-i | 29 a-s | 26 a | 50 a-k | 60 a | 0.89 a-d | 0.75 a | −0.2 a-o | 2.2 d-r | 2.1 a-i | 2.6 a-h | 6463 a-i | 2425 f-s | 0 a | 4 gh |
| CC287 | 49 a-j | 40 a-i | 28 b-s | 29 a | 53 a-j | 64 a | 0.88 a-d | 0.75 a | −0.3 a-p | 3.2 a-e | 1.1 g-j | 1.9 c-h | 5697 a-j | 2441 f-s | 1 a | 3 gh |
| CC277 | 49 a-j | 33 e-j | 30 a-r | 30 a | 47 a-l | 63 a | 0.90 ab | 0.68 a | −1.4 o-u | 2.3 c-q | 1.2 f-j | 1.8 e-h | 5439 a-j | 2707 c-r | 1 a | 0 h |
| CC266 | 29 o-t | 44 a-i | 25 f-t | 33 a | 42 a-l | 74 a | 0.87 a-d | 0.65 a | −0.3 a-q | 1.8 f-r | 1.8 b-j | 2.2 a-h | 4076 f-j | 3323 c-m | 1 a | 15 c-h |
| CC249 | 48 a-k | 44 a-i | 34 a-l | 28 a | 47 a-l | 72 a | 0.90 a-c | 0.78 a | −0.6 c-t | 1.9 d-r | 1.1 g-j | 1.9 c-h | 6295 a-j | 3154 c-n | 1 a | 1 h |
| CC246 | 51 a-g | 36 c-i | 25 f-t | 27 a | 40 c-l | 69 a | 0.88 a-d | 0.78 a | −1.7 r-u | 2.4 b-p | 1.4 d-j | 2.1 b-h | 8095 a | 2538 e-s | 3 a | 0 h |
| CC233 | 47 a-l | 45 a-i | 21 k-t | 29 a | 46 a-l | 61 a | 0.89 a-d | 0.73 a | −0.5 b-q | 1.3 n-r | 1.0 h-j | 1.8 e-h | 6400 a-j | 3198 c-m | 0 a | 0 h |
| CC230 | 49 a-j | 49 a-h | 27 b-s | 25 a | 58 a-h | 65 a | 0.90 a-c | 0.74 a | −1.3 k-u | 2.6 b-o | 1.0 h-j | 2.4 a-h | 6660 a-i | 3324 c-m | 0 a | 1 h |
| CC227 | 47 a-l | 38 b-i | 25 f-t | 32 a | 48 a-k | 70 a | 0.89 a-d | 0.69 a | −1.2 j-u | 2.5 b-p | 1.3 e-j | 2.5 a-h | 6504 a-i | 2626 c-s | 1 a | 18 b-h |
| CC223 | 49 a-j | 33 e-j | 27 c-s | 30 a | 58 a-g | 64 a | 0.90 a-d | 0.74 a | −0.3 a-q | 3.2 a-f | 1.4 d-j | 2.5 a-h | 7222 a-g | 2087 k-s | 0 a | 2 gh |
| CC221 | 46 a-n | 43 a-i | 32 a-p | 28 a | 42 a-l | 58 a | 0.87 a-d | 0.76 a | −1.5 q-u | 1.5 k-r | 1.8 a-j | 2.3 a-h | 4317 e-j | 2001 m-s | 3 a | 28 b-d |
| CC208 | 52 a-f | 30 j | 24 h-t | 24 a | 48 a-k | 60 a | 0.89 a-d | 0.73 a | −1.0 h-u | 2.3 c-q | 1.8 b-j | 1.8 d-h | 4455 d-j | 2111 k-s | 0 a | 1 h |
| CC202 | 51 a-h | 47 a-i | 35 a-k | 29 a | 56 a-i | 60 a | 0.88 a-d | 0.68 a | 0.5 a-d | 2.6 b-o | 2.2 a-h | 2.8 a-h | 5751 a-j | 2303 i-s | 2 a | 13 c-h |
| CC189 | 48 a-j | 50 a-g | 36 a-j | 23 a | 56 a-h | 64 a | 0.87 a-d | 0.70 a | 0.3 a-g | 2.6 b-n | 2.1 a-i | 2.9 a-h | 5345 a-j | 2146 j-s | 2 a | 9 e-h |
| CC187 | 41 b-q | 44 a-i | 27 b-s | 30 a | 40 b-l | 61 a | 0.87 a-d | 0.72 a | 0.2 a-g | 1.2 n-r | 1.3 e-j | 2.4 a-h | 4603 c-j | 3056 c-q | 2 a | 27 b-e |
| CC157 | 43 a-p | 36 c-i | 31 a-r | 23 a | 50 a-k | 60 a | 0.88 a-d | 0.80 a | −0.1 a-k | 2.6 b-n | 1.8 a-j | 2.6 a-h | 4884 a-j | 2890 c-r | 1 a | 0 h |
| CC155 | 47 a-l | 32 ij | 26 e-t | 27 a | 47 a-k | 67 a | 0.88 a-d | 0.73 a | −0.7 e-u | 2.6 b-n | 1.9 a-j | 2.8 a-h | 5482 a-j | 2434 f-s | 2 a | 2 gh |
| CC149 | 51 a-g | 43 a-i | 28 b-s | 25 a | 42 a-l | 63 a | 0.88 a-d | 0.70 a | −0.8 e-u | 2.3 c-q | 2.0 a-j | 2.8 a-h | 3835 g-j | 2566 d-s | 2 a | 2 gh |
| CC125 | 45 a-n | 43 a-i | 34 a-l | 29 a | 54 a-j | 65 a | 0.89 a-d | 0.76 a | −1.0 h-u | 2.6 b-n | 1.7 b-j | 2.3 a-h | 7035 a-h | 3406 b-l | 1 a | 4 gh |

**Table 5.** *Cont.*

| Genotypes | Stand Count (Plants/Plot) 2017 | Stand Count 2018 | Plant Height (cm) 2017 | Plant Height 2018 | Lateral Growth (cm) 2017 | Lateral Growth 2018 | NDVI (0–1) 2017 | NDVI 2018 | CTD (°C) 2017 | CTD 2018 | Leaf Wilting (0–5) 2017 | Leaf Wilting 2018 | Pod Yield (kg ha⁻¹) 2017 | Pod Yield 2018 | Sprouting (#/Plot) 2017 | Sprouting 2018 |
|---|---|---|---|---|---|---|---|---|---|---|---|---|---|---|---|---|
| CC119 | 50 a-h | 46 a-i | 29 b-s | 25 a | 48 a-k | 63 a | 0.87 a-d | 0.74 a | −0.5 b-r | 1.6 j-r | 2.1 a-i | 2.8 a-h | 6081 a-j | 3175 c-m | 2 a | 3 gh |
| CC115A | 10 uv | 46 a-i | 20 m-t | 27 a | 32 i-l | 70 a | 0.80 e | 0.71 a | −0.9 g-u | 3.2 a-e | 1.0 h-j | 1.6 gh | 4873 a-j | 3808 b-f | 1 a | 2 gh |
| CC112 | 50 a-h | 52 a-d | 38 a-g | 28 a | 53 a-j | 57 a | 0.88 a-d | 0.72 a | −1.1 i-u | 1.6 j-r | 2.1 a-i | 3.5 ab | 5644 a-j | 3552 b-i | 2 a | 18 b-h |
| CC087 | 50 a-i | 47 a-i | 34 a-l | 27 a | 54 a-j | 63 a | 0.87 a-d | 0.58 a | −0.4 b-q | 2.6 b-o | 1.9 a-j | 2.6 a-h | 4640 b-j | 3815 b-f | 2 a | 2 gh |
| CC082 | 46 a-n | 48 a-h | 31 a-r | 28 a | 48 a-k | 67 a | 0.87 a-d | 0.72 a | −1.4 m-u | 2.8 a-k | 1.9 a-j | 2.5 a-h | 4157 f-j | 2403 g-s | 9 a | 30 bc |
| CC080 | 48 a-j | 55 a-c | 33 a-m | 26 a | 54 a-j | 60 a | 0.87 a-d | 0.79 a | −0.4 b-q | 2.2 d-r | 1.8 a-j | 2.9 a-h | 4165 f-j | 2148 j-s | 1 a | 25 b-f |
| CC075 | 46 a-n | 44 a-i | 31 a-r | 29 a | 51 a-k | 64 a | 0.87 a-d | 0.75 a | −0.5 b-q | 2.4 c-q | 2.1 a-i | 2.9 a-h | 3852 f-j | 2588 d-s | 2 a | 3 gh |
| CC068 | 52 a-g | 52 a-d | 40 a-d | 32 a | 57 a-h | 74 a | 0.87 a-d | 0.70 a | 0.1 a-h | 1.8 d-r | 2.1 a-i | 2.6 a-h | 4942 a-j | 3935 b-d | 1 a | 2 gh |
| CC053 | 51 a-h | 46 a-i | 41 ab | 29 a | 60 a-e | 59 a | 0.88 a-d | 0.75 a | −0.5 b-q | 2.6 b-n | 2.3 a-g | 3.2 a-f | 4748 a-j | 2832 c-r | 0 a | 2 gh |
| CC047 | 49 a-j | 44 a-i | 33 a-m | 28 a | 53 a-j | 65 a | 0.86 a-d | 0.74 a | −0.1 a-k | 1.6 j-r | 2.6 a-d | 3.3 a-d | 6132 a-j | 3560 b-i | 2 a | 2 gh |
| CC041 | 49 a-j | 44 a-i | 30 a-r | 30 a | 42 a-l | 62 a | 0.88 a-d | 0.74 a | −0.1 a-k | 2.0 d-r | 1.8 a-j | 3.3 a-d | 5101 a-j | 1699 q-s | 1 a | 20 b-g |
| CC038 | 45 a-n | 35 d-i | 36 a-j | 29 a | 52 a-j | 64 a | 0.88 a-d | 0.72 a | −0.4 b-q | 4.1 a | 2.4 a-e | 3.4 a-c | 5915 a-j | 1290 s | 0 a | 52 a |
| CC033 | 46 a-n | 37 c-i | 32 a-p | 25 a | 50 a-k | 58 a | 0.89 a-d | 0.77 a | −1.0 g-u | 2.4 c-q | 2.1 a-i | 2.8 a-h | 5543 a-j | 2395 g-s | 4 a | 17 b-h |
| CC016 | 48 a-k | 51 a-e | 40 a-e | 28 a | 57 a-h | 61 a | 0.88 a-d | 0.78 a | −0.3 a-q | 3.4 a-c | 2.2 a-h | 3.4 a-c | 4233 f-j | 2870 c-r | 5 a | 17 b-h |
| CC012 | 50 a-h | 42 a-i | 32 a-n | 32 a | 46 a-l | 62 a | 0.88 a-d | 0.71 a | −0.7 e-u | 1.7 h-r | 2.3 a-g | 2.9 a-h | 4966 a-j | 2574 d-s | 1 a | 9 e-h |
| C76-16 | 44 a-o | 33 g-j | 25 f-t | 24 a | 54 a-j | 58 a | 0.90 a-c | 0.73 a | 0.2 a-g | 3.1 a-i | 1.2 f-j | 1.8 d-h | 8407 ab | 4753 b | 1 a | 0 h |
| SWR | 44 a-o | . | 27 c-s | . | 59 a-f | . | 0.86 a-d | . | −0.3 a-q | . | 1.8 a-j | . | 5713 a-j | . | 1 a | . |
| Sullivan | 37 g-s | . | 22 i-t | . | 39 c-l | . | 0.89 a-d | . | −0.7 d-t | . | 1.0 h-j | . | 7651 a-f | . | 0 a | . |
| OLE | 52 a-f | . | 34 a-l | . | 52 a-j | . | 0.88 a-d | . | −0.8 f-u | . | 1.9 a-j | . | 6973 a-h | . | 0 a | . |
| GA09B | 40 c-q | . | 18 q-t | . | 47 a-l | . | 0.89 a-d | . | −1.0 h-u | . | 0.9 ij | . | 8105 a-e | . | 1 a | . |
| CC763 | 22 s-u | . | 21 l-t | . | 34 h-l | . | 0.87 a-d | . | −1.3 k-u | . | 1.3 e-j | . | 2606 j | . | 1 a | . |
| CC631 | 9 uv | . | 16 st | . | 28 kl | . | 0.85 b-d | . | −0.5 b-r | . | 1.3 e-j | . | 3502 g-j | . | 4 a | . |
| CC610 | 6 v | . | 18 r-t | . | 34 g-l | . | 0.71 f | . | −0.3 a-q | . | 1.7 b-j | . | 3971 f-j | . | 1 a | . |
| CC552B | 30 o-t | . | 26 f-t | . | 39 c-l | . | 0.88 a-d | . | −0.7 e-u | . | 2.0 a-j | . | 3337 h-j | . | 1 a | . |
| CC552A | 27 q-t | . | 28 b-s | . | 44 a-l | . | 0.87 a-d | . | −0.9 f-u | . | 2.3 a-f | . | 3535 g-j | . | 2 a | . |
| CC516B | 38 d-q | . | 25 f-t | . | 45 a-l | . | 0.88 a-d | . | −0.2 a-m | . | 1.7 b-j | . | 5233 a-j | . | 3 a | . |
| CC516A | 54 a-c | . | 36 a-i | . | 59 a-f | . | 0.86 b-d | . | 0.2 a-g | . | 2.2 a-h | . | 5644 a-j | . | 2 a | . |
| CC433A | 32 m-s | . | 24 f-t | . | 43 a-l | . | 0.88 a-d | . | 0.1 a-i | . | 2.0 a-j | . | 4433 e-j | . | 1 a | . |
| CC132 | 22 s-u | . | 27 d-s | . | 45 a-l | . | 0.86 a-d | . | 0.1 a-h | . | 1.8 a-j | . | 3335 h-j | . | 1 a | . |
| CC115B | 15 t-v | . | 12 t | . | 24 l | . | 0.78 e | . | −0.4 b-q | . | 1.1 g-j | . | 3535 g-j | . | 0 a | . |
| CC050 | 34 j-s | . | 22 i-t | . | 36 f-l | . | 0.88 a-d | . | −0.2 a-l | . | 2.1 a-i | . | 3167 ij | . | 3 a | . |
| Mean | 44 | 44 | 29 | 27 | 48 | 63 | 0.9 | 0.73 | −0.5 | 2.3 | 1.7 | 2.5 | 5334 | 2886 | 2 | 10 |
| *p*-value | <0.0001 | <0.0001 | <0.0001 | 0.998 | <0.0001 | 0.436 | <0.0001 | 0.421 | 0.0006 | 0.004 | <0.0001 | <0.0001 | <0.0001 | <0.0001 | 0.098 | <0.0001 |

**Table 6.** Disease and insect damage parameters (spotted wilt (TSWV), southern stem rot (SSR), sclerotinia blight (SB), cylindrocladium black rot (CBR), and thrips damage) of 104 mini-core genotypes of peanut in 2017, and 88 in 2018. Values for TSWV, SSR, SB, and CBR, are averages over both measurement dates (10 and 12 WAP). The values followed by the same letters are not significantly different using Fisher's protected LSD at α = 0.05.

| Genotypes | TSWV (%) [†] | | SSR (%) | | SB (%) | | CBR (%) | | Thrips Damage (0–10) [‡] | |
|---|---|---|---|---|---|---|---|---|---|---|
| | 2017 | 2018 | 2017 | 2018 | 2017 | 2018 | 2017 | 2018 | 2017 | 2018 |
| Wynne | 4.4 f-g | 17.3 b | 0.0 a | 3 k-n | 0.5 a | 3 a | 0 a | 4.9 a | 0 a | 3.8 a |
| Walton | 5.4 e-g | 17.1 b | 0.5 a | 2.9 l-n | 1.5 a | 2.9 a | 0 a | 2.9 a | 0 a | 2.8 a |
| TVOL14 | 16.7 b-g | 48.6 ab | 0.0 a | 8.4 e-n | 2 a | 14 a | 0 a | 4.6 a | 0 a | 3.8 a |
| TS90 | 18.6 b-g | 38.7 b | 0.0 a | 14.1 a-j | 2.5 a | 3.8 a | 1.5 a | 8.5 a | 0 a | 2.5 a |
| TROL11 | 10.8 b-g | 40.1 b | 0.0 a | 5.3 h-n | 1.5 a | 4.3 a | 0 a | 4.3 a | 0 a | 3.3 a |
| NMVAL | 23.2 a-g | 38.3 b | 0.7 a | 10.7 c-n | 4 a | 18.3 a | 2.2 a | 10.6 a | 0 a | 3.3 a |
| CC812 | 16.7 b-g | 34.3 b | 1.0 a | 10 c-n | 1.5 a | 4.4 a | 0 a | 3.5 a | 0 a | 3.1 a |
| CC808 | 34.8 a-d | 43.2 b | 0.5 a | 12 b-n | 2.9 a | 14.8 a | 2 a | 8.2 a | 0 a | 4.0 a |
| CC805 | 24.0 a-g | 40.8 b | 1.5 a | 8.4 e-n | 2.5 a | 7.4 a | 0 a | 4.6 a | 0 a | 2.7 a |
| CC802 | 16.7 b-g | 39.4 b | 1.5 a | 13.7 a-l | 4.9 a | 2.4 a | 0.5 a | 6.2 a | 0 a | 2.8 a |
| CC798 | 21.1 a-g | 38.3 b | 0.0 a | 12.8 a-n | 2 a | 5.2 a | 0 a | 7.2 a | 0 a | 2.2 a |
| CC787 | 25.5 a-g | 37.8 b | 0.5 a | 2.3 n | 2.9 a | 10.7 a | 0 a | 5.1 a | 0 a | 3.3 a |
| CC781 | 49.5 a | 48.8 ab | 1.0 a | 23.2 a | 1.5 a | 6.2 a | 0 a | 4.3 a | 0 a | 3.2 a |
| CC775 | 32.4 a-f | 38.3 b | 0.0 a | 19.8 a-d | 4.4 a | 5.4 a | 0.5 a | 8.3 a | 0 a | 2.3 a |
| CC760 | 33.8 a-e | 37.2 b | 0.5 a | 5.8 g-n | 2.9 a | 4.9 a | 0 a | 3.9 a | 0 a | 4.2 a |
| CC755 | 25.5 a-g | 32.6 b | 0.0 a | 15.8 a-h | 4.9 a | 1.8 a | 0 a | 8.3 a | 0 a | 2.4 a |
| CC740 | 24.0 a-g | 38.2 b | 0.0 a | 9.1 d-n | 0.5 a | 9.2 a | 1 a | 5.2 a | 0 a | 3.2 a |
| CC725 | 24.0 a-g | 40.6 b | 0.0 a | 15.2 a-i | 2 a | 4.8 a | 0.5 a | 6.7 a | 0 a | 2.8 a |
| CC711 | 27.9 a-g | 53.0 ab | 0.0 a | 7.2 f-n | 3.4 a | 5.3 a | 0 a | 4.3 a | 0 a | 3.2 a |
| CC703B | 7.4 c-g | 42.8 b | 0.0 a | 6.5 g-n | 0 a | 2.7 a | 0 a | 2.7 a | 0 a | 3.3 a |
| CC703A | 16.7 b-g | 42.0 b | 1.5 a | 17.6 a-f | 4.4 a | 10 a | 0 a | 3.4 a | 0 a | 2.8 a |
| CC698 | 9.3 b-g | 38.7 b | 0.0 a | 5.9 g-n | 3.4 a | 3.1 a | 0 a | 5 a | 0 a | 2.8 a |
| CC678 | 8.8 b-g | 39.1 b | 0.0 a | 3.5 j-n | 0 a | 5.4 a | 0 a | 4.4 a | 0 a | 2.6 a |
| CC673 | 22.6 a-g | 29.9 b | 2.5 a | 7.4 f-n | 2 a | 6.5 a | 0 a | 11.1 a | 0 a | 2.8 a |
| CC650 | 12.3 b-g | 30.5 b | 0.5 a | 12.6 a-n | 4.9 a | 6.1 a | 0 a | 6.1 a | 0 a | 3.0 a |
| CC643 | 24.0 a-g | 36.5 b | 0.0 a | 13.9 a-l | 3.9 a | 4.5 a | 0 a | 8.3 a | 0 a | 3.5 a |
| CC605 | 32.8 a-f | 42.0 b | 0.0 a | 6.2 g-n | 4.9 a | 13.8 a | 0 a | 3.2 a | 0 a | 4.1 a |
| CC588 | 18.1 b-g | 47.9 ab | 0.0 a | 7.2 f-n | 8.3 a | 8.1 a | 0.5 a | 10.1 a | 0 a | 2.6 a |
| CC580 | 35.8 a-c | 41.0 b | 2.0 a | 9.9 c-n | 2.5 a | 9 a | 1 a | 3.4 a | 0 a | 2.4 a |
| CC579 | 20.1 b-g | 36.8 b | 1.0 a | 13.2 a-n | 3.4 a | 4.9 a | 0 a | 4.9 a | 0 a | 3.1 a |
| CC559 | 22.1 a-g | 52.7 ab | 0.5 a | 4 j-n | 3.9 a | 6.8 a | 0 a | 8.8 a | 0 a | 3.2 a |
| CC553 | 22.1 a-g | 36.5 b | 2.0 a | 11 b-n | 1.5 a | 9.1 a | 1 a | 8.2 a | 0 a | 2.8 a |

**Table 6.** *Cont.*

| Genotypes | TSWV (%) [†] | | | | SSR (%) | | | | SB (%) | | | | CBR (%) | | | | Thrips Damage (0–10) [‡] | | | |
|---|---|---|---|---|---|---|---|---|---|---|---|---|---|---|---|---|---|---|---|---|
| | 2017 | | 2018 | | 2017 | | 2018 | | 2017 | | 2018 | | 2017 | | 2018 | | 2017 | | 2018 | |
| CC548 | 11.3 | b-g | 42.2 | b | 0.0 | a | 3.4 | j-n | 1 | a | 2.5 | a | 0 | a | 5.3 | a | 0 | a | 2.9 | a |
| CC546 | 26.0 | a-g | 48.1 | ab | 1.5 | a | 8.5 | e-n | 1.5 | a | 7.6 | a | 1 | a | 6.6 | a | 0 | a | 3.2 | a |
| CC535 | 25.0 | a-g | 36.7 | b | 2.0 | a | 4.7 | h-n | 2.9 | a | 2.9 | a | 0 | a | 7.6 | a | 0 | a | 2.5 | a |
| CC529 | 8.8 | b-g | 37.0 | b | 0.0 | a | 10.8 | b-n | 2.5 | a | 5 | a | 0 | a | 5 | a | 0 | a | 2.9 | a |
| CC526 | 14.7 | b-g | 40.0 | b | 0.0 | a | 15.2 | a-i | 0 | a | 9.4 | a | 0 | a | 5.5 | a | 0 | a | 2.9 | a |
| CC508 | 31.4 | a-f | 52.3 | ab | 2.0 | a | 10.5 | c-n | 3.9 | a | 2.8 | a | 0 | a | 14.1 | a | 0 | a | 3.6 | a |
| CC488 | 18.1 | b-g | 38.1 | b | 0.0 | a | 6.6 | f-n | 0 | a | 9.4 | a | 0 | a | 12.4 | a | 0 | a | 3.8 | a |
| CC485 | 22.1 | a-g | 44.1 | ab | 2.0 | a | 2.5 | mn | 6.4 | a | 10.1 | a | 0 | a | 6.3 | a | 0 | a | 3.4 | a |
| CC481 | 29.9 | a-g | 39.6 | b | 2.5 | a | 4.7 | h-n | 2 | a | 6.7 | a | 0 | a | 2.9 | a | 0 | a | 3.0 | a |
| CC477 | 18.6 | b-g | 33.5 | b | 1.5 | a | 9.1 | d-n | 1 | a | 10.9 | a | 0 | a | 10.9 | a | 0 | a | 3.9 | a |
| CC458 | 13.2 | b-g | 44.4 | ab | 0.5 | a | 3.4 | j-n | 0.5 | a | 7.2 | a | 0 | a | 3.4 | a | 0 | a | 3.5 | a |
| CC446 | 23.0 | a-g | 44.5 | ab | 0.0 | a | 2.4 | mn | 5.9 | a | 8 | a | 0 | a | 5.3 | a | 0 | a | 3.1 | a |
| CC431 | 22.6 | a-g | 45.3 | ab | 0.0 | a | 16.7 | a-g | 1.5 | a | 5.2 | a | 0 | a | 6.1 | a | 0 | a | 3.3 | a |
| CC408 | 22.1 | a-g | 43.0 | b | 0.5 | a | 12.7 | a-n | 0 | a | 11.7 | a | 0 | a | 6.9 | a | 0 | a | 3.7 | a |
| CC406 | 21.6 | a-g | 35.5 | b | 0.0 | a | 9.3 | d-n | 2.5 | a | 2.7 | a | 0 | a | 6.5 | a | 0 | a | 2.6 | a |
| CC388 | 19.1 | b-g | 40.9 | b | 0.0 | a | 15.2 | a-i | 0.5 | a | 6.4 | a | 1 | a | 13.1 | a | 0 | a | 3.1 | a |
| CC384 | 33.3 | a-f | 31.8 | b | 0.0 | a | 8.1 | e-n | 4.9 | a | 2.5 | a | 1.5 | a | 5.4 | a | 0 | a | 3.3 | a |
| CC381 | 15.7 | b-g | 33.2 | b | 0.5 | a | 5.9 | g-n | 1 | a | 6.9 | a | 0 | a | 2.2 | a | 0 | a | 2.5 | a |
| CC342 | 19.6 | b-g | 43.3 | ab | 0.5 | a | 5.6 | h-n | 1.5 | a | 4.6 | a | 0 | a | 5.6 | a | 0 | a | 3.1 | a |
| CC338 | 22.1 | a-g | 37.8 | b | 1.0 | a | 9.7 | c-n | 2 | a | 6.9 | a | 3.4 | a | 12.4 | a | 0 | a | 2.8 | a |
| CC310 | 29.9 | a-g | 46.0 | ab | 0.0 | a | 18.5 | a-e | 1.5 | a | 3.2 | a | 1.5 | a | 7.9 | a | 0 | a | 3.2 | a |
| CC296 | 24.5 | a-g | 46.2 | ab | 0.5 | a | 5 | h-n | 2 | a | 9.8 | a | 1.5 | a | 11.8 | a | 0 | a | 2.8 | a |
| CC287 | 11.3 | b-g | 44.7 | ab | 0.0 | a | 7.4 | f-n | 0 | a | 2.6 | a | 1.5 | a | 2.6 | a | 0 | a | 3.6 | a |
| CC277 | 14.7 | b-g | 29.0 | b | 0.0 | a | 4.4 | i-n | 2.5 | a | 2.5 | a | 0 | a | 3.5 | a | 0 | a | 2.9 | a |
| CC266 | 29.9 | a-g | 42.8 | b | 1.0 | a | 13.4 | a-m | 1.5 | a | 8.6 | a | 0 | a | 4.8 | a | 0 | a | 3.5 | a |
| CC249 | 22.1 | a-g | 48.9 | ab | 0.0 | a | 12.8 | a-n | 3.9 | a | 7.2 | a | 2.5 | a | 5.2 | a | 0 | a | 3.5 | a |
| CC246 | 6.4 | d-g | 29.8 | b | 0.0 | a | 4.4 | i-n | 0 | a | 6.2 | a | 0 | a | 8 | a | 0 | a | 3.4 | a |
| CC233 | 8.3 | b-g | 40.6 | b | 0.0 | a | 8.9 | d-n | 0 | a | 3.2 | a | 0 | a | 11.8 | a | 0 | a | 2.9 | a |
| CC230 | 6.9 | c-g | 39.0 | b | 0.0 | a | 12.8 | a-n | 0 | a | 4.4 | a | 0 | a | 5.4 | a | 0 | a | 2.8 | a |
| CC227 | 10.3 | b-g | 45.5 | ab | 0.0 | a | 3.5 | j-n | 0.5 | a | 11.1 | a | 0 | a | 11.1 | a | 0 | a | 2.5 | a |
| CC223 | 7.8 | c-g | 53.6 | ab | 5.4 | a | 14.2 | a-j | 0 | a | 13.3 | a | 0.5 | a | 5.8 | a | 0 | a | 2.4 | a |

Table 6. *Cont.*

| Genotypes | TSWV (%) [†] 2017 | | 2018 | | SSR (%) 2017 | | 2018 | | SB (%) 2017 | | 2018 | | CBR (%) 2017 | | 2018 | | Thrips Damage (0–10) [‡] 2017 | | 2018 | |
|---|---|---|---|---|---|---|---|---|---|---|---|---|---|---|---|---|---|---|---|---|
| CC221 | 20.1 | b-g | 39.6 | b | 1.0 | a | 20.7 | a-c | 2.5 | a | 2.9 | a | 0 | a | 6.6 | a | 0 | a | 3.6 | a |
| CC208 | 23.5 | a-g | 45.1 | ab | 1.5 | a | 8.1 | e-n | 0.5 | a | 2.5 | a | 0 | a | 6.3 | a | 0 | a | 3.7 | a |
| CC202 | 21.6 | a-g | 37.9 | b | 2.0 | a | 9.4 | d-n | 2.9 | a | 12.4 | a | 3.4 | a | 8.5 | a | 0 | a | 3.3 | a |
| CC189 | 26.5 | a-g | 38.9 | b | 1.0 | a | 21.9 | ab | 0 | a | 14.4 | a | 0 | a | 3.9 | a | 0 | a | 2.5 | a |
| CC187 | 21.1 | a-g | 39.2 | b | 0.5 | a | 10.9 | b-n | 0 | a | 6.2 | a | 0.5 | a | 10.9 | a | 0 | a | 3.3 | a |
| CC157 | 31.4 | a-f | 45.3 | ab | 2.5 | a | 10.9 | b-n | 2 | a | 5 | a | 0 | a | 3.1 | a | 0 | a | 3.3 | a |
| CC155 | 15.2 | b-g | 41.3 | b | 0.0 | a | 7 | f-n | 1.5 | a | 6.1 | a | 0 | a | 8 | a | 0 | a | 3.3 | a |
| CC149 | 27.0 | a-g | 45.6 | ab | 2.0 | a | 14 | a-k | 2.5 | a | 8.3 | a | 0 | a | 7.2 | a | 0 | a | 3.3 | a |
| CC125 | 29.4 | a-g | 40.3 | b | 0.5 | a | 11.1 | b-n | 0.5 | a | 7.2 | a | 0 | a | 6.3 | a | 0 | a | 3.5 | a |
| CC119 | 24.0 | a-g | 39.2 | b | 0.0 | a | 7.3 | f-n | 2 | a | 4.5 | a | 1.5 | a | 8.2 | a | 0 | a | 2.3 | a |
| CC115A | 9.3 | b-g | 38.0 | b | 0.0 | a | 15.2 | a-i | 0.5 | a | 2.9 | a | 0 | a | 6.7 | a | 0 | a | 3.3 | a |
| CC112 | 16.2 | b-g | 39.6 | b | 0.0 | a | 11.6 | b-n | 0 | a | 10.8 | a | 0 | a | 3.1 | a | 0 | a | 4.5 | a |
| CC087 | 25.0 | a-g | 42.1 | b | 0.0 | a | 8.9 | d-n | 2 | a | 14.7 | a | 0 | a | 7.1 | a | 0 | a | 3.7 | a |
| CC082 | 17.6 | b-g | 36.3 | b | 0.0 | a | 13.8 | a-l | 2.9 | a | 4.1 | a | 0 | a | 6 | a | 0 | a | 3.8 | a |
| CC080 | 27.5 | a-g | 33.9 | b | 0.0 | a | 6.6 | f-n | 1 | a | 9.5 | a | 0 | a | 5.7 | a | 0 | a | 3.1 | a |
| CC075 | 23.5 | a-g | 35.3 | b | 0.5 | a | 5.5 | h-n | 1.5 | a | 6.5 | a | 0 | a | 11.3 | a | 0 | a | 2.8 | a |
| CC068 | 19.6 | b-g | 33.9 | b | 4.4 | a | 7.6 | e-n | 3.4 | a | 10.4 | a | 0 | a | 3.8 | a | 0 | a | 3.8 | a |
| CC053 | 28.4 | a-g | 95.7 | a | 0.0 | a | 5 | h-n | 0 | a | 10.7 | a | 1 | a | 3.1 | a | 0 | a | 3.3 | a |
| CC047 | 19.1 | b-g | 36.4 | b | 0.0 | a | 3.6 | j-n | 2.5 | a | 3.6 | a | 1 | a | 5.5 | a | 0 | a | 2.7 | a |
| CC041 | 27.0 | a-g | 44.6 | ab | 0.0 | a | 5.6 | h-n | 0.5 | a | 16.1 | a | 0 | a | 5.7 | ab | 0 | a | 3.6 | a |
| CC038 | 29.4 | a-g | 46.9 | ab | 0.0 | a | 10.1 | c-n | 1 | a | 13.7 | a | 1 | a | 4.4 | a | 0 | a | 3.3 | a |
| CC033 | 21.6 | a-g | 35.6 | b | 1.0 | a | 2.6 | mn | 3.9 | a | 4.5 | a | 1.5 | a | 9.2 | a | 0 | a | 2.3 | a |
| CC016 | 18.1 | b-g | 44.0 | ab | 0.5 | a | 12.7 | a-n | 2.9 | a | 12.8 | a | 1 | a | 5.2 | a | 0 | a | 4.0 | a |
| CC012 | 18.1 | b-g | 32.0 | b | 0.0 | a | 5.1 | h-n | 1.5 | a | 1.4 | a | 0 | a | 6.1 | a | 0 | a | 3.0 | a |
| C76-16 | 6.9 | c-g | 30.2 | b | 1.0 | a | 4.7 | i-n | 1.5 | a | 2.8 | a | 0 | a | 5.6 | a | 0 | a | 2.2 | a |
| SWR | 13.7 | b-g | . | | 0.0 | a | . | | 3.4 | a | . | | 0 | a | . | | 0 | a | . | |
| Sullivan | 4.4 | fg | . | | 0.0 | a | . | | 0.5 | a | . | | 0 | a | . | | 0 | a | . | |
| OLE | 16.7 | b-g | . | | 0.5 | a | . | | 1 | a | . | | 0 | a | . | | 0 | a | . | |
| GA09B | 2.0 | g | . | | 0.0 | a | . | | 0 | a | . | | 0 | a | . | | 0 | a | . | |
| CC763 | 25.5 | a-g | . | | 1.0 | a | . | | 1.5 | a | . | | 1 | a | . | | 0 | a | . | |
| CC631 | 17.2 | b-g | . | | 0.0 | a | . | | 1 | a | . | | 0.5 | a | . | | 0 | a | . | |
| CC610 | 11.8 | b-g | . | | 0.0 | a | . | | 1.5 | a | . | | 0 | a | . | | 0 | a | . | |
| CC552B | 26.5 | a-g | . | | 0.0 | a | . | | 2.5 | a | . | | 0.5 | a | . | | 0 | a | . | |
| CC552A | 26.5 | a-g | . | | 0.5 | a | . | | 0 | a | . | | 0 | a | . | | 0 | a | . | |

**Table 6.** *Cont.*

| Genotypes | TSWV (%) [†] | | SSR (%) | | SB (%) | | CBR (%) | | Thrips Damage (0–10) [‡] | |
|---|---|---|---|---|---|---|---|---|---|---|
| | 2017 | 2018 | 2017 | 2018 | 2017 | 2018 | 2017 | 2018 | 2017 | 2018 |
| CC516B | 25.0 a-g | . | 0.5 a | . | 0 a | . | 0 a | . | 0 a | . |
| CC516A | 21.6 a-g | . | 0.0 a | . | 3.4 a | . | 0 a | . | 0 a | . |
| CC433A | 16.2 b-g | . | 0.0 a | . | 3.4 a | . | 0 a | . | 0 a | . |
| CC132 | 25.5 a-g | . | 0.0 a | . | 2.9 a | . | 0 a | . | 0 a | . |
| CC115B | 15.7 b-g | . | 0.0 a | . | 0 a | . | 0 a | . | 0 a | . |
| CC050 | 37.3 ab | . | 0.0 a | . | 1 a | . | 1 a | . | 0 a | . |
| Mean | 20.5 | 40.4 | 0.6 | 9.42 | 2 | 6.98 | 0.4 | 6.48 | 0 | 3.1 |
| *p*-value | <0.0001 | 0.041 | 0.785 | <0.001 | 0.092 | 0.109 | 0.391 | 0.479 | - | 0.495 |

[†] Disease percentage was calculated as a fraction of the number of diseased plants observed, to the number of plants in each plot. [‡] Thrips damage was rated using a scale from 0 to 10, 0 being a plant not damaged by thrips, and 10 being all leaves damaged.

**Table 7.** The 2017 and 2018 stand count, plant height, lateral growth, Normalized Difference Vegetation Index (NDVI), canopy temperature depression (CTD), leaf wilting, tomato spotted wilt (TSW), southern stem rot (SSR), sclerotinia blight (SB), cylindrocladium black rot (CBR), thrips damage, pod yield, and post-harvest sprouting by varieties of the U.S. mini-core peanut collection. The plant height, lateral growth, and NDVI, are measurements taken at maximum vegetative growth (6 weeks after planting (WAP)). Leaf wilting and canopy temperature depression (CTD) are the average of two dates (10 and 12 WAP in 2017 and 5 to 7 WAP in 2018), with the highest values corresponding to sudden droughts. Values for TSW, SSR, SB, and CBR, are averages over both measurement dates (10 and 12 WAP). The values followed by the same letters are not significantly different using Fisher's protected LSD at α = 0.1.

| Varieties | Stand Count (Plants/Plot) | | Plant Height (cm) | | Lateral Growth (cm) | | NDVI | | CTD (°C) | | Leaf Wilting (0–5) | | TSW (%) | | SSR (%) | | SB (%) | | CBR (%) | | Thrips Damage (0–10) | | Pod Yield (kg ha$^{-1}$) | | Sprouting (#/Plot) | |
|---|---|---|---|---|---|---|---|---|---|---|---|---|---|---|---|---|---|---|---|---|---|---|---|---|---|---|
| | 2017 | 2018 | 2017 | 2018 | 2017 | 2018 | 2017 | 2018 | 2017 | 2018 | 2017 | 2018 | 2017 | 2018 | 2017 | 2018 | 2017 | 2018 | 2017 | 2018 | 2017 | 2018 | 2017 | 2018 | 2017 | 2018 |
| *Fastigiata* | 42 a | 45 a | 29 ab | 28 a | 48 a | 63 a | 0.88 a | 0.73 a | −0.5 a | 2.1 a | 2 bc | 3 ab | 22 a | 41 a | 0.4 a | 10.5 a | 1.9 a | 5.8 a | 0.3 a | 6.8 a | 0 a | 3.3 a | 5312 a | 2836 a | 2.4 a | 13.0 a |
| *Hypogaea* | 44 a | 43 a | 28 b | 27 a | 48 a | 63 a | 0.88 a | 0.73 a | −0.5 a | 2.4 a | 2 c | 2 b | 18 a | 40 a | 0.7 a | 9.4 a | 2.0 a | 6.7 a | 0.5 a | 6.4 a | 0 a | 3.1 a | 5569 a | 2921 a | 1.9 a | 6.9 b |
| *Peruvian* | 49 a | 38 a | 27 b | 26 a | 45 a | 65 a | 0.88 a | 0.71 a | −0.7 a | 2.5 a | 2 a | 3 a | 21 a | 44 a | 1.0 a | 10.5 a | 2.0 a | 7.2 a | 0.0 a | 7.6 a | 0 a | 3.3 a | 4658 a | 2500 a | 1.7 a | 1.8 b |
| *Vulgaris* | 46 a | 46 a | 32 a | 28 a | 50 a | 62 a | 0.87 a | 0.71 a | −0.5 a | 2.3 a | 2 ab | 3 ab | 23 a | 40 a | 0.5 a | 8.5 a | 2.0 a | 8.5 a | 0.3 a | 6.2 a | 0 a | 3.1 a | 5037 a | 2903 a | 2.2 a | 12.6 a |
| **Mean** | 45 | 43 | 29 | 27 | 48 | 63 | 0.88 | 0.72 | −0.5 | 2.3 | 2 | 3 | 21 | 41 | 1 | 9.7 | 2.0 | 7.1 | 0.3 | 6.8 | 0 | 3.2 | 5144 | 2790 | 2.0 | 8.6 |
| *p*-value | 0.665 | 0.785 | <0.0001 | 0.679 | 0.998 | 0.569 | 0.556 | 0.384 | 0.893 | 0.183 | 0.0003 | 0.009 | 0.266 | 0.938 | 0.652 | 0.486 | 0.997 | 0.082 | 0.266 | 0.823 | - | 0.572 | 0.420 | 0.802 | 0.647 | 0.005 |

### 3.4. Progression of Morphological and Physiological Traits among Varieties

On average, in all genotypes, plant height, lateral growth, and NDVI were significantly different across WAP; maximal growth rates shown by NDVI were reached at 9 WAP in 2017, and 6 WAP in 2018, with significant but small changes afterwards (Figure 2). Among varieties, *vulgaris* was significantly taller than the other varieties, whereas lateral growth was not statistically different among varieties (Figure 3). NDVI differed during the early stages of growth until 6 WAP, with *vulgaris* having the highest NDVI and *peruviana* the lowest, in both years. Because of NDVI saturation during the late season (12 WAP), all varieties showed no significant differences for NDVI at later growth stages (Figure 3). Sudden droughts were recorded at 10 and 12 WAP in 2017, and 5 and 7 WAP in 2018. During these times, CTD and wilting significantly increased for all varieties, in comparison with times of no drought stress within each year (Figure 4). Even though *peruviana* showed significantly lower CTD values in the absence of drought, indicating cooler canopies, there were no significant differences among botanical varieties for CTD during sudden droughts (Table 6). In 2017, wilting was low with no differences among varieties at 5 and 7 WAP, and visual scores ranged from 0 to 1. Towards the end of the season (after 10 WAP), wilting scores were higher, and varieties *peruviana* and *vulgaris* showed significantly more wilting than *hypogaea* and *fastigiata* (Table 6). In 2018, more wilting was observed at 5 and 7 WAP than at 10 and 12 WAP. As in 2017, in 2018, *peruviana* and *vulgaris* were significantly more wilted than *hypogaea* and *fastigiata* (Figure 4). Based on data from 2017, taller genotypes were more wilted and had more TSW pressure than smaller genotypes (Figure 5).

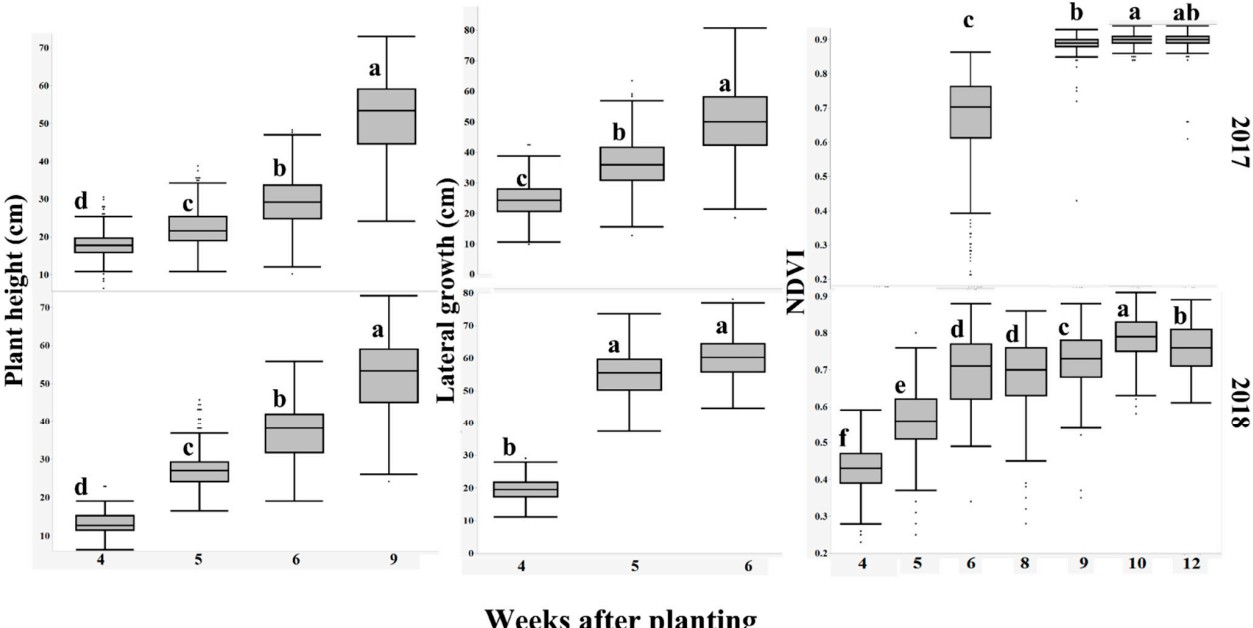

**Figure 2.** The *x*-axis shows progression of measured growth traits (plant height, lateral growth, and NDVI) for 104 mini-core peanut genotypes in 2017, and 88 in 2018, with weeks after planting on the *y*-axis. Each box and whisker plot represents the measured trait, including all genotypes on that day. Plots with the same letters are not significantly different across weeks after planting using LS means at $\alpha = 0.05$.

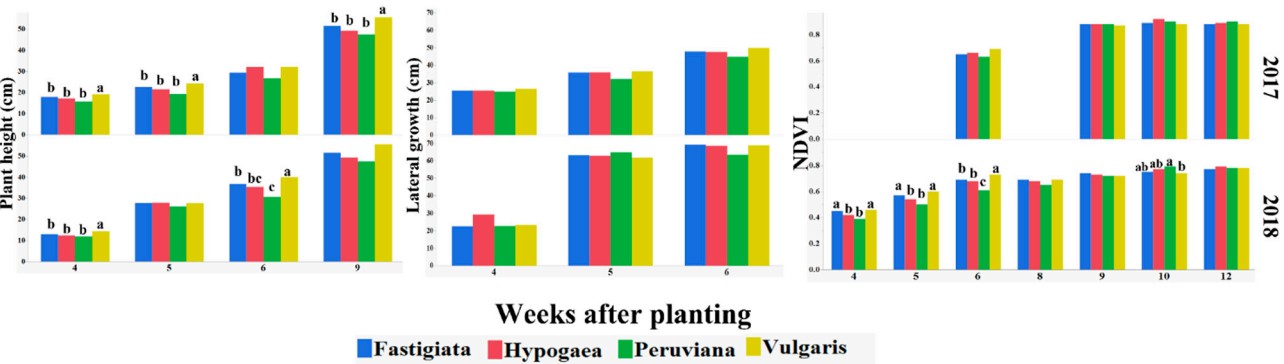

**Figure 3.** The *y*-axis shows progression of measured growth traits (plant height, lateral growth, and NDVI) for four peanut varieties (*fastigiata*, *hypogaea*, *peruviana*, and *vulgaris*) of the U.S. mini-core peanut collection in 2017 and 2018, with weeks after planting on the *x*-axis. The bars with no or similar letters are not significantly different within individual weeks after planting using LS means at $\alpha = 0.05$.

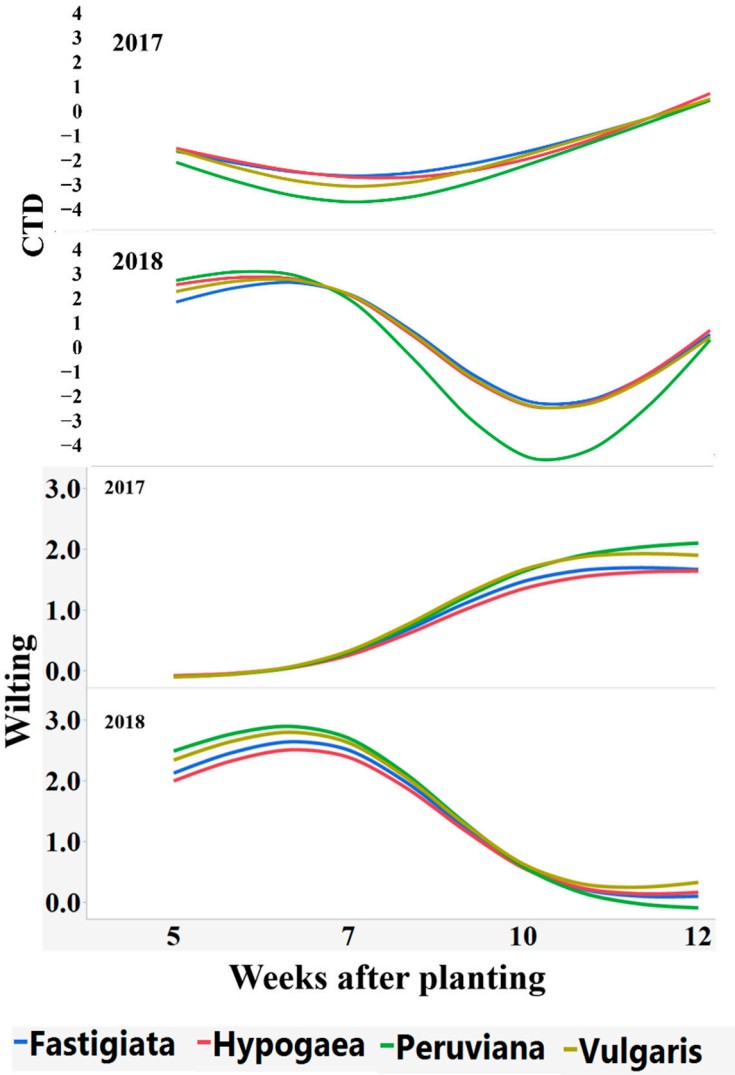

**Figure 4.** Progression of canopy temperature depression (CTD, canopy minus air temperature), and leaf wilting of peanut varieties over the growing seasons during 2017 and 2018.

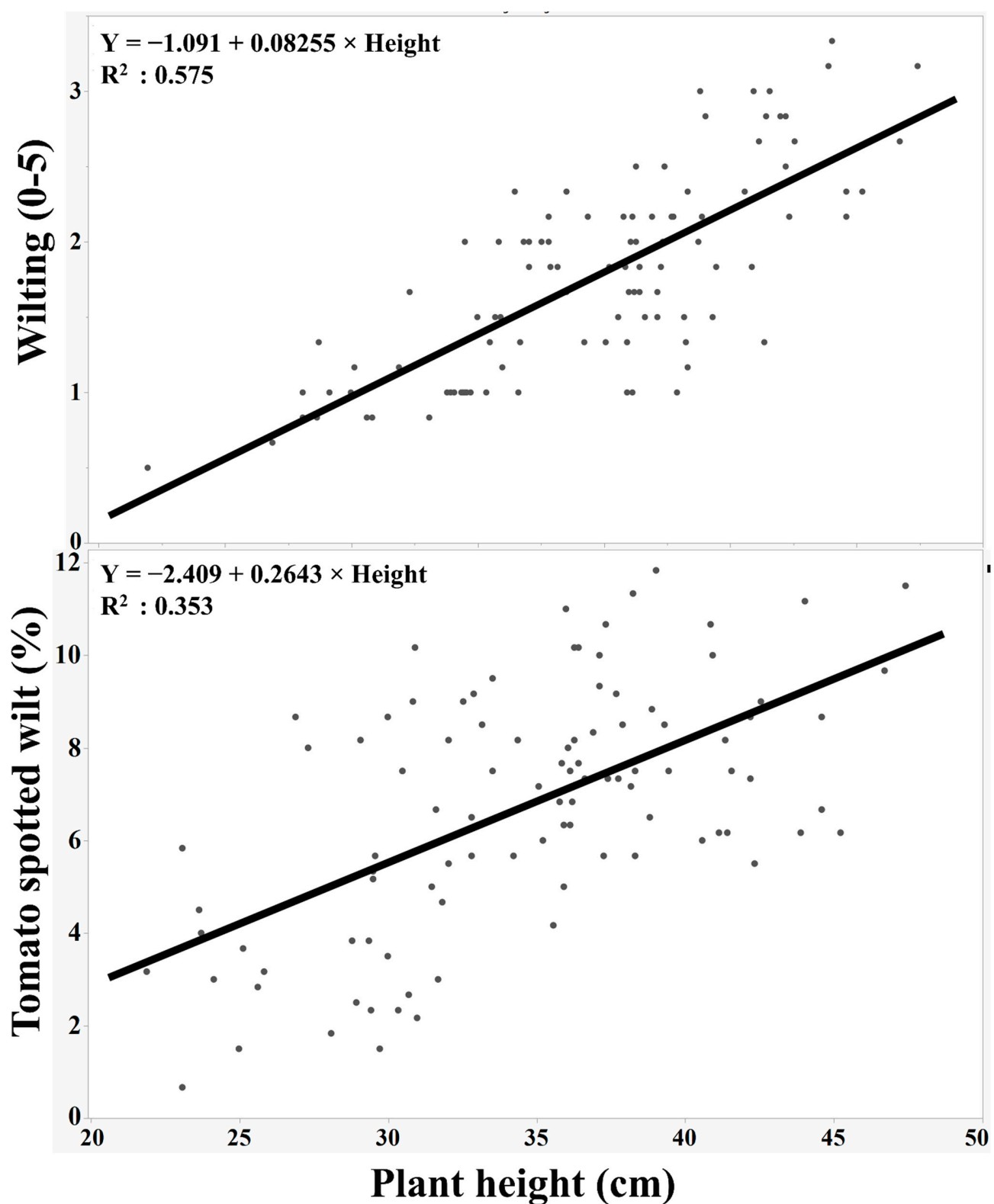

**Figure 5.** Correlation of wilting and tomato spotted wilt (TSW) with plant height for the U.S. peanut mini-core collection genotypes. Each data point for the upper graph is average plant height vs. maximum wilting at 10 weeks after planting (WAP), averaged across replications; each data point for the lower graph is average plant height vs. TSW at 12 WAP, averaged across replications in 2017.

### 3.5. Correlation of Ground Data with Aerially Derived Leaf Reflectance and Indices

There were significant correlations ($p < 0.0001$) among ground data with aerially derived VIs. In 2017, plant height and lateral growth of *hypogaea*, *fastigiata*, *vulgaris*, and *peruviana* were negatively correlated to red, green, and blue reflectance ($r = -0.32$ to $-0.80$); and positively correlated with aerial NDVI ($r = 0.24$ to $0.51$) and RGB color indices, including but not limited to lightness, a*, and u* ($r = -0.53$ to $-0.80$), green area (GA) ($r \geq 0.69$), and greener area (GGA) ($r \geq 0.59$) (Table 8). Leaf wilting, TSW, SSR, SB, CBR, and thrips damage, were significantly correlated ($p < 0.0001$) with red, green, and blue leaf reflectance, and RGB color space indices b*, v*, and CSI, among others. In 2017, peanut pod yield was correlated to several VIs, with regards to the variety (Table 8). The correlations were weaker in 2018 than 2017 for most traits, but with a similar trend of correlations (Table 9).

### 3.6. Heritability

Stand count was highly heritable ($H^2 = 0.87$), based on data in both years (Table 10). Plant height recorded greatest heritability ($H^2$ from 0.45 to 0.94) in mid-season; while ground NDVI and CTD were at the beginning of the season ($H^2 > 0.9$), after which $H^2$ gradually declined. Leaf wilting was mostly heritable ($H^2 = 0.65$) at 7 WAP, coinciding with high heritability of CTD ($H^2 = 0.79$) at that growth stage. All diseases had highest heritability ($H^2 > 0.5$) at 10–11 WAP, when most severe disease incidence was recorded. Pod yield had poor heritability. Averaging the measured traits over the growing season did not yield any significant improvement in heritability. Overall, VIs had higher heritability than red, green, and blue reflectance (Table 11). Though none of the VIs had high heritability values ($H^2 > 0.5$) consistently throughout the growing season, NIR, RGR, NDRGI, GNDVI, BNDVI, CIG, GLI, and mSR, performed better than the others at certain times during the growing season. When averaged over the growing season, red, green, blue, BGI, RGR, NGRDI, PPR, NCPI, and CVI, had heritability above 50%. Among color space indices, most had high heritability when estimated early in the season (up to 8 WAP). Only auI had consistently high heritability values over the growing season. When averaged over the different growth stages, a*, ab, uv, auI, NDabI, and NDuvI, had over 50% heritability.

**Table 8.** Heatmap correlation matrix of aerial reflectance, color space indices, and their derived vegetation indices, with physiological, morphological, and agronomic traits of peanuts in 2017.

| Indices | *Hypogaea* | | | | | | | | | | | | *Fastigiata* | | | | | | | | | | | | *Vulgaris* | | | | | | | | | | | |
|---|---|---|---|---|---|---|---|---|---|---|---|---|---|---|---|---|---|---|---|---|---|---|---|---|---|---|---|---|---|---|---|---|---|---|---|---|
| | Stand Count | Plant Height | Lateral Growth | Ground NDVI | CTD | Wilting | TSW | SSR | SB | CBR | Yield | Sprouting | Stand Count | Plant Height | Lateral Growth | Ground NDVI | CTD | Wilting | TSW | SSR | SB | CBR | Yield | Sprouting | Stand Count | Plant Height | Lateral Growth | Ground NDVI | CTD | Wilting | TSW | SSR | SB | CBR | Yield | Sprouting |
| Red | −0.72 | −0.29 | −0.76 | −0.74 | 0.41 | 0.59 | 0.48 | −0.12 | 0.27 | 0.06 | −0.57 | 0.35 | −0.69 | −0.62 | −0.60 | −0.71 | 0.45 | 0.45 | 0.45 | 0.25 | 0.34 | 0.23 | −0.40 | 0.14 | −0.75 | −0.64 | −0.74 | −0.81 | 0.36 | 0.30 | 0.24 | 0.17 | −0.04 | 0.04 | −0.58 | 0.18 |
| Green | −0.72 | −0.42 | −0.74 | −0.74 | 0.38 | 0.49 | 0.44 | −0.12 | 0.22 | 0.01 | −0.59 | 0.33 | −0.64 | −0.60 | −0.44 | −0.65 | 0.49 | 0.65 | 0.50 | 0.34 | 0.31 | 0.23 | −0.40 | 0.12 | −0.62 | −0.66 | −0.62 | −0.69 | 0.33 | 0.19 | 0.18 | 0.13 | −0.11 | 0.05 | −0.54 | 0.18 |
| Blue | −0.80 | −0.40 | −0.78 | −0.77 | 0.40 | 0.57 | 0.32 | −0.17 | 0.15 | 0.11 | −0.46 | 0.20 | −0.68 | −0.71 | −0.60 | −0.76 | 0.41 | 0.44 | 0.25 | 0.02 | 0.44 | 0.18 | −0.30 | −0.33 | −0.75 | −0.70 | −0.77 | −0.83 | 0.33 | 0.24 | 0.10 | 0.07 | −0.24 | 0.07 | −0.43 | −0.03 |
| NIR | −0.05 | −0.39 | 0.37 | 0.02 | 0.14 | 0.20 | 0.15 | 0.32 | 0.09 | 0.10 | 0.27 | 0.03 | −0.17 | −0.25 | 0.16 | −0.09 | 0.25 | 0.40 | 0.14 | 0.09 | 0.30 | 0.16 | 0.14 | 0.43 | −0.27 | −0.58 | −0.16 | −0.24 | 0.18 | −0.01 | 0.26 | −0.01 | 0.35 | 0.11 | 0.41 | −0.08 |
| NDVI | 0.35 | −0.07 | 0.69 | 0.46 | −0.16 | −0.15 | −0.15 | 0.28 | −0.07 | 0.03 | 0.54 | −0.23 | 0.36 | 0.26 | 0.52 | 0.44 | −0.14 | −0.19 | −0.32 | −0.18 | −0.03 | −0.08 | 0.57 | 0.19 | 0.35 | 0.04 | 0.45 | 0.43 | −0.13 | −0.08 | 0.11 | −0.07 | 0.26 | 0.04 | 0.55 | −0.14 |
| BGI | −0.63 | −0.05 | −0.27 | −0.26 | 0.12 | −0.13 | −0.45 | −0.01 | −0.25 | 0.16 | 0.03 | −0.05 | −0.07 | −0.21 | −0.31 | −0.21 | −0.18 | −0.47 | −0.38 | −0.45 | 0.13 | −0.08 | −0.39 | −0.19 | −0.05 | −0.28 | −0.23 | −0.04 | 0.08 | −0.17 | −0.12 | −0.29 | 0.04 | −0.20 | −0.12 | |
| RGR | −0.36 | 0.33 | 0.11 | 0.14 | −0.01 | −0.01 | 0.29 | −0.10 | 0.29 | 0.22 | −0.36 | 0.34 | 0.25 | 0.26 | −0.03 | 0.22 | −0.35 | −0.39 | −0.01 | −0.24 | 0.17 | 0.09 | −0.26 | 0.17 | 0.24 | 0.43 | 0.24 | 0.29 | −0.19 | 0.24 | 0.27 | 0.20 | 0.29 | −0.02 | −0.56 | 0.20 |
| NPPR | 0.56 | 0.06 | 0.04 | 0.04 | −0.06 | 0.05 | 0.13 | 0.03 | −0.01 | −0.23 | 0.07 | −0.19 | −0.08 | −0.01 | 0.19 | 0.01 | 0.27 | 0.40 | 0.29 | 0.44 | −0.16 | 0.02 | 0.03 | 0.24 | −0.05 | −0.24 | −0.02 | −0.06 | 0.12 | −0.15 | −0.04 | −0.02 | 0.04 | −0.02 | 0.31 | −0.04 |
| NGRDI | 0.35 | −0.33 | −0.12 | −0.14 | 0.02 | 0.01 | −0.29 | 0.09 | −0.29 | −0.22 | 0.33 | −0.35 | −0.25 | −0.26 | 0.04 | −0.21 | 0.34 | 0.38 | 0.01 | 0.23 | −0.17 | −0.10 | 0.25 | −0.16 | −0.24 | −0.43 | −0.24 | −0.28 | 0.18 | −0.23 | −0.28 | −0.19 | −0.29 | 0.02 | 0.55 | −0.21 |
| PPR | 0.61 | 0.05 | 0.26 | 0.25 | −0.12 | 0.13 | 0.44 | 0.01 | 0.24 | −0.16 | −0.08 | 0.04 | 0.08 | 0.22 | 0.31 | 0.22 | 0.17 | 0.46 | 0.38 | 0.45 | −0.13 | 0.08 | −0.06 | 0.41 | 0.20 | 0.05 | 0.28 | 0.23 | 0.03 | −0.08 | 0.17 | 0.12 | 0.29 | −0.04 | 0.16 | 0.09 |
| NCPI | 0.54 | 0.43 | 0.47 | 0.48 | −0.18 | 0.59 | 0.57 | −0.04 | 0.38 | −0.04 | −0.27 | 0.22 | 0.37 | 0.55 | 0.34 | 0.50 | −0.16 | 0.36 | 0.41 | 0.40 | −0.07 | 0.12 | −0.18 | 0.44 | 0.53 | 0.60 | 0.62 | 0.62 | −0.19 | 0.68 | 0.28 | 0.20 | 0.39 | −0.05 | −0.04 | 0.17 |
| SRI | 0.33 | −0.09 | 0.68 | 0.44 | −0.14 | −0.19 | −0.21 | 0.32 | −0.14 | 0.04 | 0.53 | −0.24 | 0.36 | 0.27 | 0.53 | 0.43 | −0.14 | −0.23 | −0.35 | −0.18 | −0.04 | −0.11 | 0.58 | 0.25 | 0.35 | 0.04 | 0.45 | 0.43 | −0.12 | −0.06 | 0.02 | −0.11 | 0.26 | 0.06 | 0.60 | −0.20 |
| GRVI | 0.38 | 0.02 | 0.68 | 0.46 | −0.14 | −0.12 | −0.17 | 0.30 | −0.10 | 0.07 | 0.50 | −0.15 | 0.44 | 0.36 | 0.48 | 0.49 | −0.27 | −0.17 | −0.35 | −0.24 | −0.01 | −0.08 | 0.50 | 0.26 | 0.40 | 0.26 | 0.49 | 0.49 | −0.19 | −0.02 | 0.06 | −0.08 | 0.30 | 0.06 | 0.54 | −0.15 |
| IO | 0.53 | 0.43 | 0.47 | 0.48 | −0.17 | 0.59 | 0.57 | −0.03 | 0.38 | −0.05 | −0.31 | 0.18 | 0.37 | 0.55 | 0.34 | 0.50 | −0.15 | 0.36 | 0.41 | 0.41 | −0.07 | 0.12 | −0.20 | 0.45 | 0.52 | 0.60 | 0.62 | 0.62 | −0.19 | 0.68 | 0.27 | 0.20 | 0.38 | −0.05 | −0.09 | 0.10 |
| GNDVI | 0.40 | 0.04 | 0.69 | 0.48 | −0.15 | −0.09 | −0.11 | 0.27 | −0.04 | 0.06 | 0.51 | −0.17 | 0.43 | 0.34 | 0.46 | 0.48 | −0.26 | −0.14 | −0.33 | −0.23 | 0.01 | −0.06 | 0.50 | 0.30 | 0.41 | 0.26 | 0.48 | 0.49 | −0.19 | −0.05 | 0.13 | −0.05 | 0.29 | 0.04 | 0.51 | −0.12 |
| BNDVI | 0.42 | 0.06 | 0.73 | 0.53 | −0.19 | −0.07 | −0.01 | 0.30 | 0.03 | 0.02 | 0.42 | −0.11 | 0.46 | 0.42 | 0.59 | 0.57 | −0.18 | 0.14 | −0.07 | 0.06 | −0.06 | 0.00 | 0.27 | 0.54 | 0.49 | 0.28 | 0.60 | 0.59 | −0.18 | −0.05 | 0.17 | −0.02 | 0.32 | 0.02 | 0.42 | −0.01 |
| CIG | 0.38 | 0.02 | 0.68 | 0.46 | −0.14 | −0.12 | −0.17 | 0.30 | −0.10 | 0.07 | 0.50 | −0.15 | 0.44 | 0.36 | 0.48 | 0.49 | −0.27 | 0.36 | −0.35 | −0.24 | −0.01 | −0.08 | 0.50 | 0.36 | 0.40 | 0.26 | 0.49 | 0.49 | −0.19 | −0.02 | 0.06 | −0.08 | 0.30 | 0.06 | 0.54 | −0.15 |
| CVI | 0.55 | 0.43 | 0.46 | 0.48 | −0.18 | 0.59 | 0.58 | −0.04 | 0.38 | −0.04 | −0.20 | 0.24 | 0.37 | 0.55 | 0.33 | 0.50 | −0.16 | 0.42 | 0.41 | 0.40 | −0.08 | 0.13 | −0.17 | 0.44 | 0.53 | 0.61 | 0.62 | 0.62 | −0.19 | 0.68 | 0.29 | 0.20 | 0.39 | −0.05 | 0.00 | 0.20 |
| GLI | 0.57 | −0.16 | 0.07 | 0.05 | −0.06 | 0.07 | 0.15 | 0.05 | 0.01 | −0.24 | 0.13 | −0.18 | −0.10 | −0.03 | 0.19 | 0.00 | 0.29 | 0.24 | 0.29 | 0.44 | −0.17 | 0.02 | 0.07 | 0.21 | −0.06 | −0.25 | −0.02 | −0.07 | 0.13 | −0.16 | −0.04 | −0.03 | 0.04 | −0.02 | 0.36 | −0.02 |
| GBNDVI | 0.40 | 0.04 | 0.71 | 0.50 | −0.16 | −0.09 | −0.09 | 0.29 | −0.03 | 0.05 | 0.48 | −0.15 | 0.45 | 0.38 | 0.52 | 0.53 | −0.24 | −0.04 | −0.25 | −0.14 | −0.02 | −0.04 | 0.44 | 0.27 | 0.45 | 0.27 | 0.54 | 0.54 | −0.19 | −0.05 | 0.13 | −0.05 | 0.31 | 0.03 | 0.49 | −0.09 |
| GRNDVI | 0.38 | −0.01 | 0.70 | 0.47 | −0.15 | −0.12 | −0.14 | 0.28 | −0.07 | 0.05 | 0.52 | −0.20 | 0.41 | 0.31 | 0.50 | 0.47 | −0.22 | −0.17 | −0.33 | −0.21 | −0.01 | −0.07 | 0.55 | 0.27 | 0.40 | 0.19 | 0.49 | 0.48 | −0.17 | −0.06 | 0.10 | −0.07 | 0.28 | 0.04 | 0.54 | −0.15 |
| RBNDVI | 0.25 | −0.17 | 0.53 | 0.28 | −0.07 | −0.13 | −0.15 | 0.22 | −0.02 | 0.02 | 0.58 | −0.25 | 0.12 | −0.09 | 0.21 | 0.11 | −0.08 | −0.40 | −0.44 | −0.37 | 0.05 | −0.08 | 0.51 | −0.06 | 0.18 | 0.05 | 0.19 | 0.01 | −0.05 | −0.14 | 0.19 | −0.03 | 0.18 | 0.05 | 0.41 | −0.06 |
| mSR | 0.37 | −0.01 | 0.70 | 0.49 | −0.16 | −0.15 | −0.14 | 0.33 | −0.09 | 0.03 | 0.46 | −0.17 | 0.42 | 0.39 | 0.58 | 0.53 | −0.17 | −0.05 | −0.22 | −0.06 | −0.06 | −0.07 | 0.41 | 0.52 | 0.45 | 0.21 | 0.57 | 0.54 | −0.15 | −0.04 | 0.06 | −0.08 | 0.32 | 0.05 | 0.51 | −0.13 |
| GARI | 0.38 | −0.01 | 0.71 | 0.49 | −0.17 | −0.13 | −0.10 | 0.30 | −0.05 | 0.03 | 0.49 | −0.19 | 0.41 | 0.34 | 0.56 | 0.51 | −0.17 | −0.06 | −0.23 | −0.09 | −0.05 | −0.05 | 0.47 | 0.40 | 0.43 | 0.16 | 0.54 | 0.52 | −0.15 | −0.06 | 0.12 | −0.05 | 0.29 | 0.03 | 0.51 | −0.10 |
| Intensity | −0.85 | −0.65 | −0.82 | −0.93 | 0.56 | 0.27 | 0.16 | −0.20 | −0.06 | 0.02 | −0.56 | 0.32 | −0.83 | −0.86 | −0.78 | −0.92 | 0.45 | 0.48 | 0.32 | 0.19 | 0.09 | 0.11 | −0.45 | 0.05 | −0.82 | −0.87 | −0.86 | −0.93 | 0.37 | 0.25 | −0.16 | −0.01 | −0.16 | 0.01 | −0.48 | 0.12 |
| Hue | 0.79 | 0.64 | 0.64 | 0.83 | −0.65 | −0.40 | −0.22 | 0.23 | 0.00 | −0.05 | 0.54 | −0.36 | 0.70 | 0.69 | 0.70 | 0.82 | −0.28 | −0.62 | −0.22 | −0.15 | 0.20 | −0.06 | 0.55 | −0.33 | 0.69 | 0.46 | 0.57 | 0.74 | −0.40 | −0.27 | 0.10 | −0.04 | −0.07 | 0.02 | 0.63 | −0.27 |
| Saturation | 0.83 | 0.52 | 0.76 | 0.69 | −0.26 | 0.12 | 0.35 | 0.22 | 0.42 | −0.05 | 0.09 | 0.12 | 0.55 | 0.72 | 0.56 | 0.59 | −0.38 | 0.44 | 0.37 | 0.35 | 0.13 | 0.14 | −0.09 | 0.43 | 0.55 | 0.72 | 0.67 | 0.67 | −0.19 | −0.35 | 0.33 | 0.14 | 0.29 | −0.04 | 0.18 | 0.14 |
| Lightness | −0.82 | −0.63 | −0.80 | −0.92 | 0.56 | 0.31 | 0.24 | −0.18 | 0.01 | 0.01 | −0.59 | 0.34 | −0.84 | −0.84 | −0.77 | −0.92 | 0.45 | 0.57 | 0.43 | 0.30 | 0.15 | 0.15 | −0.51 | 0.13 | −0.82 | −0.87 | −0.85 | −0.92 | 0.38 | 0.24 | −0.10 | 0.02 | −0.15 | 0.01 | −0.50 | 0.17 |
| a* | −0.72 | −0.69 | −0.77 | −0.88 | 0.56 | −0.04 | −0.36 | −0.23 | −0.39 | 0.09 | −0.22 | 0.05 | −0.78 | −0.91 | −0.83 | −0.89 | 0.37 | −0.36 | −0.46 | −0.40 | −0.31 | −0.19 | −0.09 | −0.06 | −0.80 | −0.73 | −0.85 | −0.92 | 0.35 | 0.36 | −0.41 | −0.17 | −0.06 | −0.02 | −0.43 | −0.02 |
| b* | 0.60 | 0.46 | 0.69 | 0.57 | −0.17 | 0.46 | 0.61 | 0.06 | 0.46 | −0.06 | −0.31 | 0.29 | 0.41 | 0.61 | 0.41 | 0.43 | −0.29 | 0.64 | 0.53 | 0.44 | 0.21 | 0.20 | −0.23 | 0.30 | 0.48 | 0.60 | 0.64 | 0.58 | −0.15 | −0.17 | 0.48 | 0.18 | 0.18 | 0.01 | −0.12 | 0.27 |
| u* | −0.69 | −0.66 | −0.67 | −0.84 | 0.60 | 0.05 | −0.24 | −0.24 | −0.32 | 0.09 | −0.31 | 0.16 | −0.75 | −0.82 | −0.80 | −0.88 | 0.33 | −0.20 | −0.41 | −0.37 | −0.34 | −0.18 | −0.21 | 0.06 | −0.74 | −0.61 | −0.73 | −0.83 | 0.35 | 0.33 | −0.37 | −0.13 | −0.02 | −0.03 | −0.49 | 0.08 |
| v* | 0.41 | 0.42 | 0.62 | 0.50 | −0.13 | 0.47 | 0.56 | −0.03 | 0.34 | −0.05 | −0.46 | 0.32 | 0.33 | 0.55 | 0.32 | 0.34 | −0.23 | 0.66 | 0.54 | 0.43 | 0.22 | 0.20 | −0.28 | 0.24 | 0.44 | 0.51 | 0.58 | 0.53 | −0.13 | 0.03 | 0.36 | 0.24 | 0.02 | 0.03 | −0.33 | 0.28 |
| GA | 0.89 | 0.70 | 0.81 | 0.91 | −0.60 | 0.04 | 0.08 | 0.22 | 0.25 | −0.01 | 0.43 | −0.09 | 0.82 | 0.91 | 0.83 | 0.92 | −0.46 | 0.29 | 0.28 | 0.07 | 0.47 | 0.05 | 0.32 | −0.17 | 0.86 | 0.79 | 0.87 | 0.97 | −0.42 | −0.20 | 0.33 | 0.10 | 0.18 | 0.01 | 0.46 | 0.06 |
| GGA | 0.85 | 0.62 | 0.80 | 0.90 | −0.61 | −0.25 | −0.11 | 0.24 | 0.08 | −0.07 | 0.55 | −0.34 | 0.78 | 0.86 | 0.82 | 0.89 | −0.36 | −0.49 | −0.12 | 0.00 | 0.15 | 0.02 | 0.52 | −0.28 | 0.75 | 0.63 | 0.78 | 0.86 | −0.33 | −0.36 | 0.14 | 0.00 | 0.00 | −0.02 | 0.58 | −0.25 |
| CSI | −0.67 | −0.42 | −0.66 | −0.81 | 0.63 | 0.44 | 0.25 | −0.20 | 0.07 | 0.13 | −0.55 | 0.38 | −0.55 | −0.43 | −0.50 | −0.64 | 0.14 | 0.64 | 0.29 | 0.04 | 0.05 | 0.00 | −0.30 | 0.27 | −0.18 | 0.00 | −0.15 | −0.22 | 0.07 | 0.37 | 0.04 | 0.08 | 0.17 | 0.02 | −0.56 | 0.28 |
| ab | −0.71 | −0.67 | −0.80 | −0.84 | 0.46 | −0.28 | −0.54 | −0.17 | −0.46 | 0.09 | 0.06 | −0.16 | −0.72 | −0.91 | −0.79 | −0.82 | 0.38 | −0.53 | −0.51 | −0.46 | −0.25 | −0.19 | 0.10 | −0.26 | −0.76 | −0.79 | −0.90 | −0.90 | 0.30 | 0.30 | −0.48 | −0.26 | −0.11 | −0.01 | −0.16 | −0.20 |
| uv | −0.63 | −0.66 | −0.69 | −0.84 | 0.57 | −0.22 | −0.47 | −0.18 | −0.41 | 0.10 | −0.07 | −0.07 | −0.77 | −0.87 | −0.82 | −0.89 | 0.36 | −0.47 | −0.50 | −0.46 | −0.30 | −0.20 | −0.01 | −0.16 | −0.75 | −0.66 | −0.79 | −0.86 | 0.34 | 0.27 | −0.48 | −0.24 | −0.03 | −0.04 | −0.29 | −0.13 |
| abI | −0.72 | −0.61 | −0.58 | −0.78 | 0.61 | 0.43 | 0.25 | −0.23 | 0.03 | 0.05 | −0.56 | 0.36 | −0.67 | −0.68 | −0.70 | −0.79 | 0.25 | 0.62 | 0.21 | 0.14 | −0.20 | 0.06 | −0.56 | 0.33 | −0.64 | −0.44 | −0.56 | −0.70 | 0.34 | 0.29 | −0.04 | 0.06 | 0.10 | −0.02 | −0.66 | 0.30 |
| uvI | −0.73 | −0.64 | −0.61 | −0.83 | 0.62 | 0.36 | 0.14 | −0.25 | −0.07 | 0.05 | −0.52 | 0.35 | −0.72 | −0.75 | −0.75 | −0.85 | 0.29 | 0.49 | 0.05 | 0.00 | −0.25 | −0.01 | −0.50 | 0.28 | −0.71 | −0.54 | −0.66 | −0.79 | 0.36 | 0.33 | −0.14 | 0.01 | 0.01 | −0.01 | −0.62 | 0.26 |
| auI | −0.27 | 0.40 | 0.56 | 0.55 | −0.34 | 0.18 | 0.04 | −0.01 | 0.02 | 0.10 | 0.11 | 0.04 | 0.19 | 0.26 | 0.20 | 0.17 | −0.30 | 0.13 | −0.23 | −0.19 | −0.38 | −0.14 | −0.30 | 0.28 | 0.18 | 0.16 | 0.07 | 0.23 | −0.08 | 0.18 | −0.35 | −0.08 | −0.07 | −0.05 | 0.03 | 0.08 |
| bvI | 0.77 | 0.48 | 0.73 | 0.69 | −0.26 | −0.28 | −0.13 | 0.23 | 0.07 | −0.01 | 0.49 | −0.14 | 0.58 | 0.70 | 0.56 | 0.61 | −0.40 | −0.28 | −0.19 | −0.06 | −0.08 | −0.07 | 0.42 | 0.42 | 0.52 | 0.72 | 0.68 | 0.63 | −0.18 | −0.32 | 0.10 | 0.00 | 0.23 | −0.05 | 0.39 | 0.04 |
| NDabI | 0.69 | 0.48 | 0.55 | 0.65 | −0.44 | −0.53 | −0.36 | 0.25 | −0.12 | −0.06 | 0.68 | −0.38 | 0.54 | 0.66 | 0.66 | 0.66 | −0.13 | −0.63 | −0.20 | −0.15 | 0.20 | −0.09 | 0.54 | −0.34 | 0.49 | 0.29 | 0.48 | 0.55 | −0.17 | −0.31 | −0.12 | −0.09 | −0.16 | −0.01 | 0.70 | −0.31 |
| NDuvI | 0.71 | 0.59 | 0.63 | 0.76 | −0.54 | −0.45 | −0.20 | 0.26 | 0.01 | −0.06 | 0.61 | −0.38 | 0.65 | 0.76 | 0.75 | 0.77 | −0.23 | −0.51 | −0.05 | 0.00 | 0.25 | −0.02 | 0.53 | −0.28 | 0.62 | 0.47 | 0.63 | 0.70 | −0.25 | −0.36 | 0.04 | −0.04 | −0.06 | −0.01 | 0.68 | −0.28 |
| NDLab | 0.71 | 0.60 | 0.55 | 0.77 | −0.62 | −0.41 | −0.25 | 0.22 | −0.03 | −0.04 | 0.53 | −0.36 | 0.68 | 0.65 | 0.68 | 0.80 | −0.26 | −0.65 | −0.27 | −0.19 | 0.16 | −0.08 | 0.55 | −0.33 | 0.64 | 0.44 | 0.54 | 0.69 | −0.37 | −0.27 | 0.05 | −0.06 | −0.09 | 0.02 | 0.65 | −0.29 |
| NDLuv | 0.75 | 0.65 | 0.63 | 0.85 | −0.64 | −0.32 | −0.13 | 0.24 | 0.08 | −0.04 | 0.45 | −0.33 | 0.76 | 0.72 | 0.73 | 0.88 | −0.33 | −0.53 | −0.11 | −0.04 | 0.22 | −0.01 | 0.44 | −0.27 | 0.74 | 0.56 | 0.64 | 0.81 | −0.43 | −0.30 | 0.16 | 0.00 | 0.01 | 0.02 | 0.60 | −0.25 |
| GI | 0.68 | 0.42 | 0.66 | 0.80 | −0.62 | −0.44 | −0.26 | 0.20 | −0.08 | −0.12 | 0.55 | −0.38 | 0.55 | 0.43 | 0.50 | 0.63 | −0.14 | −0.64 | −0.29 | −0.04 | −0.06 | 0.00 | 0.30 | −0.27 | 0.17 | 0.00 | 0.14 | 0.20 | −0.06 | −0.38 | −0.09 | −0.10 | −0.20 | −0.03 | 0.57 | −0.30 |
| GPI | 0.78 | 0.61 | 0.78 | 0.84 | −0.53 | −0.18 | −0.06 | 0.25 | 0.13 | −0.05 | 0.54 | −0.31 | 0.73 | 0.90 | 0.85 | 0.83 | −0.35 | −0.36 | 0.00 | 0.02 | 0.27 | 0.03 | 0.52 | −0.29 | 0.71 | 0.69 | 0.85 | 0.85 | −0.25 | −0.34 | 0.19 | 0.02 | 0.06 | −0.02 | 0.56 | −0.21 |
| NDGI | −0.66 | −0.43 | −0.66 | −0.79 | 0.62 | 0.42 | 0.24 | −0.20 | 0.06 | 0.12 | −0.52 | 0.36 | −0.57 | −0.43 | −0.50 | −0.66 | 0.17 | 0.64 | 0.29 | 0.03 | 0.06 | −0.01 | −0.22 | 0.25 | −0.17 | −0.01 | −0.15 | −0.21 | 0.07 | 0.37 | 0.05 | 0.10 | 0.19 | 0.02 | −0.53 | 0.28 |

**Table 9.** Heatmap correlation matrix of aerial reflectance, color space indices, and their derived vegetation indices, with physiological, morphological, and agronomic traits of peanuts in 2018.

| Indices | Hypogaea | | | | | | | | | | | | Fastigiata | | | | | | | | | | | | Vulgaris | | | | | | | | | | | |
|---|---|---|---|---|---|---|---|---|---|---|---|---|---|---|---|---|---|---|---|---|---|---|---|---|---|---|---|---|---|---|---|---|---|---|---|---|
| | Stand Count | Plant Height | Lateral Growth | Ground NDVI | CTD | Wilting | TSW | SSR | SB | CBR | Yield | Sprouting | Stand Count | Plant Height | Lateral Growth | Ground NDVI | CTD | Wilting | TSW | SSR | SB | CBR | Yield | Sprouting | Stand Count | Plant Height | Lateral Growth | Ground NDVI | CTD | Wilting | TSW | SSR | SB | CBR | Yield | Sprouting |
| Red | −0.40 | −0.50 | −0.26 | −0.25 | 0.34 | −0.31 | 0.12 | 0.27 | 0.34 | 0.18 | −0.31 | 0.36 | 0.04 | 0.07 | 0.06 | −0.32 | −0.23 | −0.27 | −0.15 | 0.19 | 0.13 | 0.16 | −0.56 | −0.38 | 0.13 | −0.10 | −0.33 | −0.27 | 0.18 | −0.31 | 0.05 | −0.05 | 0.10 | −0.36 | −0.46 | 0.36 |
| Green | −0.57 | −0.60 | −0.27 | −0.40 | 0.37 | −0.46 | −0.01 | 0.25 | 0.17 | 0.13 | −0.23 | 0.35 | −0.07 | −0.05 | 0.03 | −0.50 | −0.11 | −0.40 | −0.09 | 0.29 | 0.18 | 0.05 | −0.52 | −0.42 | 0.26 | −0.26 | −0.33 | −0.51 | 0.18 | −0.33 | 0.06 | −0.16 | 0.01 | −0.34 | −0.47 | 0.34 |
| Blue | −0.55 | −0.62 | −0.24 | −0.37 | 0.28 | −0.34 | 0.03 | 0.00 | −0.09 | 0.15 | −0.26 | −0.17 | −0.36 | −0.08 | 0.11 | −0.43 | −0.10 | 0.12 | 0.02 | −0.02 | −0.09 | 0.36 | −0.28 | −0.49 | 0.19 | −0.39 | −0.48 | −0.52 | 0.39 | −0.45 | −0.02 | 0.13 | 0.28 | −0.13 | −0.39 | 0.44 |
| NIR | 0.35 | 0.31 | 0.38 | 0.57 | −0.39 | −0.02 | 0.14 | 0.04 | 0.05 | 0.03 | 0.02 | −0.01 | 0.01 | −0.07 | −0.17 | 0.18 | −0.02 | 0.12 | 0.18 | 0.25 | 0.33 | 0.08 | −0.01 | 0.00 | 0.28 | 0.25 | 0.28 | 0.19 | −0.08 | 0.04 | 0.28 | 0.18 | −0.17 | 0.14 | −0.03 | 0.11 |
| NDVI | 0.48 | 0.51 | 0.43 | 0.59 | −0.49 | 0.14 | −0.04 | −0.19 | −0.26 | −0.15 | 0.23 | −0.29 | −0.05 | −0.08 | −0.17 | 0.27 | 0.09 | 0.20 | 0.18 | −0.08 | −0.01 | −0.13 | 0.34 | 0.09 | 0.14 | 0.23 | 0.40 | 0.27 | −0.13 | 0.18 | 0.04 | 0.13 | −0.15 | 0.37 | 0.33 | −0.24 |
| BGI | −0.47 | −0.56 | −0.19 | −0.29 | 0.15 | −0.20 | 0.06 | −0.11 | −0.13 | 0.10 | −0.22 | −0.30 | −0.46 | −0.13 | 0.16 | −0.27 | −0.03 | −0.08 | 0.09 | −0.12 | −0.10 | 0.40 | −0.02 | −0.45 | 0.05 | −0.32 | −0.39 | −0.41 | 0.46 | −0.49 | −0.05 | 0.16 | 0.26 | 0.02 | −0.24 | 0.33 |
| RGR | 0.44 | 0.33 | 0.11 | 0.38 | −0.13 | 0.38 | 0.22 | 0.25 | 0.45 | 0.20 | −0.45 | 0.31 | 0.14 | 0.23 | 0.05 | 0.41 | −0.21 | 0.32 | −0.14 | 0.12 | 0.15 | 0.31 | −0.53 | −0.19 | −0.20 | 0.24 | 0.03 | 0.39 | 0.08 | 0.00 | 0.06 | 0.05 | 0.16 | −0.27 | −0.36 | 0.32 |
| NPPR | 0.18 | 0.34 | 0.11 | 0.05 | −0.05 | −0.05 | −0.22 | −0.12 | −0.24 | −0.22 | 0.32 | −0.02 | 0.41 | −0.02 | −0.16 | 0.00 | 0.14 | −0.09 | 0.03 | −0.05 | −0.11 | −0.49 | 0.34 | 0.41 | 0.10 | 0.20 | 0.43 | 0.09 | −0.37 | 0.42 | 0.01 | −0.07 | −0.28 | 0.16 | 0.36 | −0.31 |
| NGRDI | −0.44 | −0.32 | −0.11 | −0.38 | 0.13 | −0.38 | −0.22 | −0.24 | −0.44 | −0.20 | 0.43 | −0.31 | −0.13 | −0.23 | −0.05 | −0.41 | 0.21 | −0.32 | 0.13 | −0.14 | −0.19 | −0.32 | 0.52 | 0.21 | 0.20 | −0.23 | −0.03 | −0.39 | −0.08 | −0.01 | −0.07 | −0.01 | −0.16 | 0.27 | 0.36 | −0.30 |
| PPR | 0.46 | 0.56 | 0.19 | 0.29 | −0.14 | 0.20 | −0.06 | 0.11 | 0.14 | −0.11 | 0.20 | 0.32 | 0.46 | 0.14 | −0.16 | 0.27 | 0.02 | 0.08 | −0.08 | 0.12 | 0.10 | −0.40 | 0.06 | −0.04 | −0.04 | 0.32 | 0.40 | 0.41 | −0.46 | 0.49 | 0.06 | −0.17 | −0.25 | −0.03 | 0.23 | −0.31 |
| NCPI | 0.55 | 0.59 | 0.20 | 0.39 | −0.17 | 0.32 | 0.05 | 0.20 | 0.32 | −0.01 | −0.03 | 0.41 | 0.41 | 0.24 | −0.12 | 0.46 | −0.08 | 0.23 | −0.14 | 0.18 | 0.19 | −0.22 | −0.23 | 0.44 | −0.11 | 0.33 | 0.30 | 0.51 | −0.50 | 0.39 | 0.10 | −0.17 | −0.18 | −0.14 | 0.07 | −0.24 |
| SRI | 0.46 | 0.48 | 0.44 | 0.57 | −0.47 | 0.14 | −0.06 | −0.25 | −0.26 | −0.14 | 0.23 | −0.31 | 0.03 | −0.05 | −0.15 | 0.28 | 0.07 | 0.23 | 0.12 | −0.19 | −0.19 | −0.23 | 0.24 | 0.02 | 0.13 | 0.24 | 0.40 | 0.29 | −0.16 | 0.19 | 0.00 | 0.16 | −0.15 | 0.32 | 0.32 | −0.23 |
| GRVI | 0.53 | 0.52 | 0.43 | 0.62 | −0.46 | 0.22 | 0.09 | −0.18 | −0.09 | −0.08 | 0.08 | −0.26 | 0.06 | 0.00 | −0.12 | 0.35 | 0.01 | 0.28 | 0.14 | −0.11 | −0.01 | −0.03 | 0.25 | −0.07 | 0.03 | 0.33 | 0.39 | 0.42 | −0.15 | 0.21 | 0.11 | 0.22 | −0.10 | 0.37 | 0.25 | −0.21 |
| IO | 0.55 | 0.59 | 0.20 | 0.38 | −0.17 | 0.33 | 0.07 | 0.19 | 0.35 | −0.07 | −0.03 | 0.45 | 0.42 | 0.25 | −0.11 | 0.45 | −0.11 | 0.22 | −0.12 | 0.17 | 0.17 | −0.23 | −0.13 | 0.40 | −0.11 | 0.32 | 0.31 | 0.51 | −0.50 | 0.39 | 0.07 | −0.20 | −0.15 | −0.15 | 0.11 | −0.23 |
| GNDVI | 0.55 | 0.55 | 0.42 | 0.64 | −0.48 | 0.23 | 0.10 | −0.13 | −0.07 | −0.08 | 0.07 | −0.24 | −0.01 | −0.02 | −0.15 | 0.34 | 0.04 | 0.26 | 0.18 | −0.06 | 0.09 | 0.00 | 0.22 | −0.05 | 0.04 | 0.33 | 0.39 | 0.40 | −0.12 | 0.19 | 0.12 | 0.25 | −0.11 | 0.39 | 0.24 | −0.20 |
| BNDVI | 0.57 | 0.61 | 0.36 | 0.57 | −0.41 | 0.23 | 0.03 | 0.04 | 0.11 | −0.13 | 0.18 | 0.17 | 0.18 | 0.02 | −0.18 | 0.36 | 0.05 | 0.22 | 0.06 | 0.09 | 0.17 | −0.31 | 0.06 | 0.07 | 0.00 | 0.43 | 0.52 | 0.46 | −0.30 | 0.33 | 0.10 | −0.08 | −0.32 | 0.15 | 0.36 | −0.29 |
| CIG | 0.53 | 0.52 | 0.43 | 0.62 | −0.46 | 0.22 | 0.09 | −0.18 | −0.09 | −0.08 | 0.08 | −0.26 | 0.06 | 0.00 | −0.12 | 0.35 | 0.01 | 0.28 | 0.14 | −0.11 | −0.01 | −0.03 | 0.25 | −0.07 | 0.03 | 0.33 | 0.39 | 0.42 | −0.15 | 0.21 | 0.11 | 0.22 | −0.10 | 0.37 | 0.25 | −0.21 |
| CVI | 0.56 | 0.59 | 0.19 | 0.39 | −0.17 | 0.31 | 0.04 | 0.20 | 0.31 | 0.01 | −0.01 | 0.39 | 0.40 | 0.23 | −0.13 | 0.46 | −0.06 | 0.24 | −0.15 | 0.18 | 0.20 | −0.20 | −0.30 | 0.46 | −0.12 | 0.33 | 0.29 | 0.52 | −0.51 | 0.38 | 0.11 | −0.15 | −0.19 | −0.13 | 0.05 | −0.24 |
| GLI | 0.18 | 0.35 | 0.11 | 0.05 | −0.06 | −0.05 | −0.21 | −0.11 | −0.24 | −0.22 | 0.42 | −0.02 | 0.40 | −0.02 | −0.15 | 0.00 | 0.14 | −0.10 | 0.03 | −0.02 | −0.06 | −0.47 | 0.34 | 0.39 | 0.09 | 0.20 | 0.43 | 0.09 | −0.37 | 0.43 | 0.01 | −0.12 | −0.28 | 0.15 | 0.37 | −0.34 |
| GBNDVI | 0.56 | 0.57 | 0.41 | 0.61 | −0.45 | 0.23 | 0.08 | −0.09 | −0.02 | −0.12 | 0.14 | −0.11 | 0.12 | 0.01 | −0.16 | 0.35 | 0.03 | 0.25 | 0.13 | −0.01 | 0.11 | −0.16 | 0.19 | −0.01 | 0.02 | 0.40 | 0.48 | 0.44 | −0.20 | 0.26 | 0.15 | 0.15 | −0.26 | 0.36 | 0.38 | −0.31 |
| GRNDVI | 0.52 | 0.53 | 0.43 | 0.62 | −0.49 | 0.19 | 0.02 | −0.18 | −0.18 | −0.12 | 0.16 | −0.28 | −0.01 | −0.04 | −0.15 | 0.32 | 0.05 | 0.24 | 0.17 | −0.09 | 0.00 | −0.09 | 0.29 | 0.01 | 0.08 | 0.29 | 0.40 | 0.35 | −0.13 | 0.19 | 0.07 | 0.18 | −0.13 | 0.30 | 0.30 | −0.23 |
| RBNDVI | −0.19 | −0.20 | 0.24 | 0.13 | −0.30 | −0.17 | 0.02 | −0.14 | −0.23 | −0.11 | 0.11 | −0.33 | −0.18 | −0.26 | −0.03 | −0.18 | 0.03 | −0.01 | 0.24 | −0.03 | 0.15 | 0.05 | 0.44 | 0.12 | 0.16 | −0.28 | −0.20 | −0.19 | 0.20 | −0.18 | 0.04 | 0.23 | −0.01 | 0.35 | 0.18 | −0.11 |
| mSR | 0.53 | 0.56 | 0.41 | 0.57 | −0.42 | 0.21 | −0.06 | −0.17 | −0.15 | −0.18 | 0.28 | −0.14 | 0.23 | 0.02 | −0.15 | 0.35 | 0.03 | 0.25 | 0.05 | −0.11 | −0.09 | −0.34 | 0.32 | 0.06 | 0.03 | 0.41 | 0.52 | 0.44 | −0.27 | 0.31 | 0.09 | 0.09 | −0.27 | 0.29 | 0.39 | −0.29 |
| GARI | 0.52 | 0.57 | 0.41 | 0.59 | −0.45 | 0.20 | −0.03 | −0.14 | −0.17 | −0.17 | 0.26 | −0.17 | 0.10 | −0.02 | −0.17 | 0.32 | 0.06 | 0.22 | 0.14 | −0.04 | 0.03 | −0.24 | 0.27 | 0.09 | 0.06 | 0.39 | 0.52 | 0.38 | −0.23 | 0.27 | 0.08 | 0.08 | −0.25 | 0.35 | 0.40 | −0.30 |
| Intensity | −0.39 | −0.37 | −0.38 | −0.38 | 0.30 | −0.34 | 0.07 | 0.24 | 0.23 | 0.19 | −0.26 | 0.28 | 0.20 | 0.14 | −0.14 | −0.03 | 0.09 | −0.11 | −0.09 | 0.18 | 0.10 | 0.21 | −0.46 | −0.44 | −0.25 | 0.11 | −0.13 | −0.39 | 0.24 | −0.35 | 0.05 | −0.04 | 0.15 | −0.35 | −0.47 | 0.41 |
| Hue | 0.50 | 0.48 | 0.40 | 0.54 | −0.32 | 0.32 | −0.18 | −0.27 | −0.47 | −0.18 | 0.51 | −0.39 | 0.52 | 0.27 | −0.04 | 0.22 | −0.13 | 0.13 | 0.19 | −0.14 | −0.16 | −0.15 | 0.29 | 0.08 | 0.23 | 0.34 | 0.53 | −0.24 | 0.17 | −0.09 | −0.06 | −0.03 | 0.23 | 0.41 | 0.06 | −0.35 |
| Saturation | 0.45 | 0.38 | 0.32 | 0.45 | −0.40 | 0.33 | −0.01 | 0.16 | 0.22 | −0.07 | 0.09 | 0.36 | −0.32 | 0.14 | 0.26 | −0.30 | −0.45 | −0.36 | −0.11 | 0.14 | 0.14 | −0.34 | 0.10 | 0.49 | −0.17 | 0.19 | 0.19 | 0.34 | −0.23 | 0.37 | 0.08 | −0.15 | −0.23 | −0.07 | −0.02 | −0.32 |
| Lightness | −0.35 | −0.34 | −0.37 | −0.34 | 0.28 | −0.33 | 0.02 | 0.26 | 0.21 | 0.14 | −0.25 | 0.35 | 0.19 | 0.20 | −0.14 | −0.02 | 0.05 | −0.14 | −0.09 | 0.27 | 0.18 | 0.09 | −0.51 | −0.41 | −0.27 | 0.15 | −0.04 | −0.35 | 0.23 | −0.34 | 0.07 | −0.13 | 0.03 | −0.35 | −0.48 | 0.35 |
| a* | −0.48 | −0.46 | −0.36 | −0.52 | 0.31 | −0.27 | 0.27 | 0.15 | 0.43 | 0.18 | −0.48 | 0.14 | −0.49 | −0.28 | 0.03 | −0.18 | 0.17 | −0.07 | −0.15 | −0.11 | −0.06 | 0.34 | −0.14 | −0.07 | −0.06 | −0.31 | −0.35 | −0.51 | 0.25 | −0.15 | −0.01 | 0.21 | 0.24 | −0.14 | −0.15 | 0.27 |
| b* | 0.43 | 0.32 | 0.13 | 0.49 | −0.40 | 0.19 | 0.00 | 0.24 | 0.25 | 0.04 | −0.15 | 0.42 | −0.19 | 0.32 | 0.11 | −0.21 | −0.38 | −0.42 | −0.13 | 0.28 | 0.23 | −0.22 | −0.41 | 0.09 | −0.05 | −0.30 | 0.29 | 0.18 | 0.26 | −0.16 | 0.21 | 0.07 | −0.18 | −0.17 | −0.13 | −0.12 |
| u* | −0.46 | −0.46 | −0.37 | −0.50 | 0.28 | −0.27 | 0.25 | 0.21 | 0.47 | 0.17 | −0.50 | 0.26 | −0.53 | −0.20 | 0.04 | −0.22 | 0.09 | −0.15 | −0.20 | −0.06 | −0.04 | 0.22 | −0.24 | −0.05 | −0.24 | −0.14 | −0.49 | 0.00 | 0.19 | 0.19 | 0.00 | 0.07 | −0.19 | −0.14 | −0.25 | 0.28 |
| v* | 0.29 | 0.21 | 0.00 | 0.36 | −0.25 | 0.04 | −0.02 | 0.25 | 0.20 | 0.07 | −0.17 | 0.41 | −0.09 | 0.38 | 0.03 | −0.13 | −0.32 | −0.36 | −0.11 | 0.32 | 0.25 | −0.15 | −0.57 | −0.13 | −0.32 | 0.30 | 0.17 | 0.22 | −0.10 | 0.09 | 0.07 | −0.19 | −0.14 | −0.25 | −0.23 | −0.02 |
| GA | 0.54 | 0.50 | 0.43 | 0.56 | −0.36 | 0.37 | −0.11 | −0.31 | −0.28 | −0.26 | 0.50 | 0.02 | 0.59 | 0.14 | −0.02 | 0.34 | −0.01 | 0.23 | 0.18 | 0.15 | 0.10 | −0.09 | 0.42 | 0.61 | 0.54 | −0.24 | 0.24 | 0.02 | −0.40 | −0.13 | 0.14 | 0.02 | −0.09 | 0.37 | 0.42 | −0.44 |
| GGA | 0.50 | 0.47 | 0.45 | 0.57 | −0.41 | 0.26 | −0.19 | −0.27 | −0.49 | −0.15 | 0.41 | −0.43 | 0.57 | 0.15 | −0.12 | 0.32 | 0.08 | 0.18 | 0.13 | −0.15 | −0.16 | −0.19 | 0.32 | 0.23 | 0.38 | 0.17 | 0.38 | 0.42 | −0.31 | 0.18 | −0.01 | 0.02 | −0.09 | 0.37 | 0.31 | −0.16 |
| CSI | −0.34 | −0.34 | −0.26 | −0.41 | 0.37 | 0.05 | 0.17 | 0.26 | 0.50 | 0.12 | −0.42 | 0.45 | −0.38 | −0.01 | 0.23 | −0.18 | −0.24 | −0.01 | −0.12 | 0.18 | 0.18 | 0.17 | −0.33 | −0.20 | −0.32 | 0.08 | −0.18 | 0.03 | 0.27 | 0.06 | 0.02 | −0.03 | 0.09 | −0.35 | −0.25 | 0.13 |
| ab | −0.48 | −0.46 | −0.33 | −0.52 | 0.32 | −0.26 | 0.19 | −0.07 | 0.09 | 0.09 | −0.25 | −0.24 | −0.36 | −0.31 | 0.01 | −0.13 | 0.22 | 0.01 | −0.05 | −0.28 | −0.26 | 0.35 | 0.08 | −0.09 | 0.06 | −0.34 | −0.36 | −0.49 | 0.24 | −0.16 | −0.07 | 0.26 | 0.25 | 0.04 | −0.01 | 0.21 |
| uv | −0.45 | −0.46 | −0.37 | −0.48 | 0.27 | −0.27 | 0.26 | 0.12 | 0.38 | 0.13 | −0.47 | 0.09 | −0.49 | −0.16 | 0.04 | −0.23 | 0.07 | −0.16 | −0.20 | −0.20 | −0.21 | 0.21 | −0.16 | −0.02 | −0.23 | −0.05 | −0.24 | −0.47 | 0.25 | −0.11 | −0.06 | 0.27 | 0.24 | −0.15 | −0.19 | 0.22 |
| abI | −0.47 | −0.46 | −0.38 | −0.51 | 0.29 | −0.28 | 0.18 | 0.27 | 0.46 | 0.18 | −0.49 | 0.39 | −0.54 | −0.23 | 0.05 | −0.23 | 0.10 | −0.15 | −0.19 | 0.15 | 0.16 | 0.14 | −0.31 | −0.03 | −0.26 | −0.21 | −0.23 | −0.50 | 0.25 | −0.13 | 0.08 | 0.04 | 0.02 | −0.25 | −0.41 | 0.30 |
| uvI | −0.47 | −0.46 | −0.37 | −0.51 | 0.29 | −0.28 | 0.20 | 0.27 | 0.47 | 0.20 | −0.50 | 0.36 | −0.55 | −0.23 | 0.04 | −0.21 | 0.11 | −0.14 | −0.17 | 0.11 | 0.12 | 0.21 | −0.31 | −0.09 | −0.24 | −0.23 | −0.26 | −0.50 | 0.26 | −0.14 | 0.06 | 0.10 | 0.09 | −0.25 | −0.39 | 0.34 |
| auI | 0.07 | 0.05 | 0.17 | 0.16 | −0.04 | 0.19 | 0.06 | 0.15 | 0.31 | 0.03 | −0.07 | 0.12 | 0.03 | 0.02 | −0.09 | 0.08 | 0.08 | −0.11 | −0.23 | −0.11 | −0.21 | 0.01 | 0.18 | 0.09 | 0.09 | 0.13 | −0.25 | 0.12 | −0.02 | 0.14 | 0.01 | −0.57 | −0.35 | 0.09 | 0.06 | −0.17 |
| bvI | 0.25 | 0.20 | 0.26 | 0.21 | −0.26 | 0.29 | 0.06 | 0.07 | 0.24 | −0.08 | 0.15 | 0.21 | −0.40 | −0.23 | 0.30 | −0.32 | −0.22 | −0.16 | −0.12 | −0.09 | −0.04 | −0.34 | 0.38 | 0.56 | 0.00 | −0.04 | 0.08 | 0.16 | −0.25 | 0.39 | 0.04 | −0.02 | −0.24 | 0.06 | 0.46 | −0.46 |
| NDabI | 0.49 | 0.48 | 0.40 | 0.53 | −0.31 | 0.31 | −0.15 | −0.19 | −0.37 | −0.16 | 0.43 | −0.28 | 0.52 | 0.26 | −0.06 | 0.25 | −0.09 | 0.17 | 0.20 | −0.02 | −0.25 | −0.15 | 0.21 | 0.14 | 0.28 | 0.21 | 0.20 | 0.52 | −0.26 | 0.13 | −0.13 | 0.06 | −0.11 | 0.16 | 0.41 | −0.28 |
| NDuvI | 0.49 | 0.47 | 0.40 | 0.53 | −0.31 | 0.31 | −0.20 | −0.23 | −0.44 | −0.19 | 0.44 | −0.32 | 0.54 | 0.26 | −0.05 | 0.23 | −0.11 | 0.15 | 0.15 | −0.18 | −0.26 | −0.25 | 0.44 | 0.18 | 0.26 | 0.23 | 0.25 | 0.51 | −0.27 | 0.14 | −0.11 | 0.05 | −0.10 | 0.23 | 0.38 | −0.30 |
| NDLab | 0.46 | 0.46 | 0.36 | 0.50 | −0.27 | 0.26 | −0.18 | −0.28 | −0.47 | −0.17 | 0.50 | −0.40 | 0.54 | 0.20 | −0.05 | 0.22 | −0.09 | 0.16 | 0.19 | −0.12 | −0.12 | −0.12 | 0.29 | −0.01 | 0.26 | 0.18 | 0.22 | 0.49 | −0.25 | 0.12 | −0.07 | −0.08 | −0.02 | 0.25 | 0.42 | −0.30 |
| NDLuv | 0.45 | 0.45 | 0.35 | 0.49 | −0.27 | 0.25 | −0.19 | −0.29 | −0.47 | −0.19 | 0.52 | −0.37 | 0.56 | 0.18 | −0.04 | 0.19 | −0.10 | 0.13 | 0.18 | −0.07 | −0.06 | −0.18 | 0.23 | 0.05 | 0.23 | 0.21 | 0.27 | 0.49 | −0.26 | 0.13 | −0.05 | −0.17 | −0.10 | 0.23 | 0.42 | −0.36 |
| GI | 0.34 | 0.34 | 0.28 | 0.41 | −0.37 | −0.04 | −0.17 | −0.26 | −0.49 | −0.12 | 0.42 | −0.45 | 0.38 | 0.03 | −0.23 | 0.18 | 0.23 | 0.02 | 0.13 | −0.18 | −0.18 | −0.18 | 0.34 | 0.19 | 0.32 | −0.08 | 0.18 | −0.04 | −0.27 | −0.06 | −0.02 | 0.04 | −0.08 | 0.35 | 0.26 | −0.13 |
| GPI | 0.51 | 0.48 | 0.48 | 0.57 | −0.40 | 0.34 | −0.19 | −0.27 | −0.49 | −0.17 | 0.38 | −0.40 | 0.60 | 0.22 | −0.08 | 0.33 | 0.00 | 0.22 | 0.14 | −0.13 | −0.16 | −0.20 | 0.33 | 0.26 | 0.35 | 0.27 | 0.42 | 0.47 | −0.30 | 0.21 | 0.01 | 0.01 | −0.09 | 0.39 | 0.35 | −0.17 |
| NDGI | −0.36 | −0.36 | −0.27 | −0.41 | 0.38 | 0.03 | 0.17 | 0.25 | 0.48 | 0.09 | −0.45 | 0.44 | −0.36 | 0.00 | 0.23 | −0.17 | −0.23 | −0.01 | −0.18 | 0.15 | 0.11 | 0.15 | −0.41 | −0.11 | −0.26 | 0.07 | −0.22 | 0.05 | 0.27 | 0.07 | 0.00 | −0.03 | 0.12 | −0.32 | −0.23 | 0.13 |

**Table 10.** Heritability of physiological, morphological, disease, sprouting, and yield measurements, of peanuts over different growth stages. Heritability values range from 0 to 1; the closer the values to 1 the higher the heritability.

| Traits | Weeks after Planting | | | | | | | | | Average [†] |
|---|---|---|---|---|---|---|---|---|---|---|
| | 4 | 5 | 6 | 7 | 9 | 10 | 11 | 12 | 16 | |
| | Vegetative Phase | Beginning Bloom | Beginning Peg | Beginning Pod | Full Pod | Beginning Seed | Seed Development | Full Seed | Harvest | |
| Stand count | 0.87 | . | . | . | . | . | . | . | . | 0.87 |
| Thrips | 0.01 | . | . | . | . | . | . | . | . | 0.01 |
| Plant height | 0.18 | 0.22 | 0.45 | . | 0.94 | . | . | . | . | 0.37 |
| Lateral growth | 0.32 | 0.03 | 0.07 | . | . | . | . | . | . | 0.07 |
| NDVI | 0.95 | 0.91 | 0.25 | 0.80 | 0.04 | 0.06 | . | 0.08 | . | 0.02 |
| CTD | . | 0.97 | 0.03 | 0.79 | 0.11 | 0.33 | . | 0.26 | . | 0.07 |
| Wilting | . | . | . | 0.65 | . | 0.05 | . | 0.16 | . | 0.32 |
| TSW | . | . | . | . | . | 0.21 | 0.52 | 0.26 | . | 0.15 |
| SSR | . | . | . | . | . | 0.58 | 0.33 | 0.15 | . | 0.10 |
| SB | . | . | . | . | . | 0.24 | 0.66 | 0.52 | . | 0.20 |
| CBR | . | . | . | . | . | 0.38 | 0.56 | 0.27 | . | 0.09 |
| Sprouting | . | . | . | . | . | . | . | . | 0.26 | 0.26 |
| Yield | . | . | . | . | . | . | . | . | 0.14 | 0.14 |

[†] Average values have been calculated by averaging the actual measurements over all weeks after planting.

**Table 11.** Heritability of aerially derived vegetation indices over different growth stages. Heritability values range from 0 to 1; the closer the values to 1 the higher the heritability.

| Indices | Weeks after Planting | | | | | | Avg. [†] | Indices | Weeks after Planting | | | | | | Avg. |
|---|---|---|---|---|---|---|---|---|---|---|---|---|---|---|---|
| | 4 | 6 | 8 | 10 | 12 | 14 | | | 4 | 6 | 8 | 10 | 12 | 14 | |
| | Vegetative Phase | Beginning Peg | Pod Development | Beginning Seed | Full Seed | Pod Maturity | | | Vegetative Phase | Beginning Peg | Pod Development | Beginning Seed | Full Seed | Pod Maturity | |
| Red | 0.09 | 0.01 | 0.07 | 0.33 | 0.17 | 0.1 | 0.52 | Intensity | 0.31 | 0.33 | 0.25 | 0.49 | 0.23 | 0.13 | 0.42 |
| Green | 0.36 | 0.01 | 0.1 | 0.25 | 0.11 | 0.41 | 0.61 | Hue | 0.39 | 0.48 | 0.56 | 0.02 | 0.13 | 0.03 | 0.29 |
| Blue | 0.67 | 0.03 | 0.08 | 0.11 | 0.17 | 0.1 | 0.53 | Saturation | 0.3 | 0.38 | 0.24 | 0.05 | 0.05 | 0.21 | 0.49 |
| NIR | 0.22 | 0.7 | 0.07 | 0.47 | 0.21 | 0.48 | 0.45 | Lightness | 0.27 | 0.29 | 0.24 | 0.41 | 0.12 | 0.26 | 0.37 |
| NDVI | 0.25 | 0.52 | 0.05 | 0.23 | 0.13 | 0.22 | 0.42 | a* | 0.38 | 0.51 | 0.59 | 0.05 | 0.16 | 0.02 | 0.5 |
| BGI | 0.33 | 0.21 | 0.22 | 0.24 | 0.05 | 0.08 | 0.56 | b* | 0.17 | 0.24 | 0.3 | 0.29 | 0.06 | 0.58 | 0.34 |
| RGR | 0.06 | 0.72 | 0.87 | 0.02 | 0.45 | 0.02 | 0.54 | u* | 0.25 | 0.4 | 0.63 | 0.02 | 0.49 | 0.02 | 0.33 |
| NPPR | 0.63 | 0.26 | 0.53 | 0.08 | 0.1 | 0.03 | 0.39 | v* | 0.15 | 0.21 | 0.37 | 0.56 | 0.07 | 0.69 | 0.32 |
| NGRDI | 0.06 | 0.72 | 0.86 | 0.02 | 0.49 | 0.02 | 0.6 | GA | 0.4 | 0.36 | 0.51 | 0.9 | 0.19 | 0.04 | 0.24 |
| PPR | 0.33 | 0.21 | 0.22 | 0.24 | 0.06 | 0.09 | 0.5 | GGA | 0.12 | 0.44 | 0.39 | 0.53 | 0.07 | 0.02 | 0.05 |
| NCPI | 0.09 | 0.29 | 0.13 | 0.03 | 0.06 | 0.62 | 0.62 | CSI | 0.06 | 0.16 | 0.39 | 0.42 | 0.07 | 0.02 | 0.04 |
| SRI | 0.19 | 0.52 | 0.04 | 0.21 | 0.13 | 0.23 | 0.4 | ab | 0.44 | 0.54 | 0.46 | 0.27 | 0.06 | 0.05 | 0.52 |
| GRVI | 0.35 | 0.61 | 0.04 | 0.61 | 0.09 | 0.26 | 0.39 | uv | 0.2 | 0.36 | 0.63 | 0.07 | 0.14 | 0.02 | 0.52 |
| IO | 0.09 | 0.29 | 0.12 | 0.03 | 0.08 | 0.54 | 0.39 | abI | 0.3 | 0.43 | 0.52 | 0.02 | 0.13 | 0.03 | 0.28 |
| GNDVI | 0.43 | 0.61 | 0.05 | 0.67 | 0.09 | 0.25 | 0.35 | uvI | 0.31 | 0.45 | 0.6 | 0.02 | 0.19 | 0.03 | 0.25 |
| BNDVI | 0.64 | 0.63 | 0.04 | 0.37 | 0.25 | 0.16 | 0.38 | auI | 0.78 | 0.82 | 0.92 | 0.22 | 0.66 | 0.93 | 0.8 |
| CIG | 0.35 | 0.61 | 0.04 | 0.61 | 0.09 | 0.26 | 0.39 | bvI | 0.48 | 0.3 | 0.2 | 0.05 | 0.14 | 0.32 | 0.49 |
| CVI | 0.09 | 0.3 | 0.13 | 0.03 | 0.06 | 0.66 | 0.74 | NDabI | 0.33 | 0.42 | 0.43 | 0.87 | 0.44 | 0.06 | 0.87 |
| GLI | 0.63 | 0.26 | 0.52 | 0.08 | 0.1 | 0.03 | 0.46 | NDuvI | 0.32 | 0.44 | 0.54 | 0.03 | 0.3 | 0.03 | 0.5 |
| GBNDVI | 0.49 | 0.61 | 0.04 | 0.57 | 0.18 | 0.2 | 0.39 | NDLab | 0.27 | 0.41 | 0.54 | 0.02 | 0.11 | 0.04 | 0.18 |
| GRNDVI | 0.3 | 0.57 | 0.04 | 0.59 | 0.1 | 0.24 | 0.38 | NDLuv | 0.28 | 0.45 | 0.65 | 0.03 | 0.17 | 0.03 | 0.2 |
| RBNDVI | 0.53 | 0.55 | 0.15 | 0.09 | 0.1 | 0.41 | 0.34 | GI | 0.06 | 0.15 | 0.39 | 0.42 | 0.08 | 0.02 | 0.04 |
| mSR | 0.41 | 0.59 | 0.04 | 0.63 | 0.28 | 0.16 | 0.35 | GPI | 0.18 | 0.46 | 0.43 | 0.61 | 0.06 | 0.02 | 0.05 |
| GARI | 0.35 | 0.57 | 0.04 | 0.54 | 0.23 | 0.17 | 0.38 | NDGI | 0.06 | 0.17 | 0.39 | 0.44 | 0.1 | 0.02 | 0.03 |

[†] Average values have been calculated by averaging the actual measurements over all weeks after planting.

## 4. Discussion

The information compiled in Table 1 could be a useful tool for further studies; updates by other authors can be made when more data become available. In Table 1, the market types described by [41,43] mostly coincided, but there were a few exceptions. For example, accession PI 292950 was described as a Runner by one author and mixed type by the other. Similarly, PI 403813 was classified as Spanish in one paper and Valencia in another. It was most confusing when sorting the accessions by market and botanical types. It is expected that Virginia and Runner market types have morphological types and belong to subspecies *hypogaea*, Valencia market type to variety *fastigiata*, and Spanish market type to variety *vulgaris*. However, the mini-core collection has unique phenotypes with market types that do not match botanical varieties, or subspecies (Stalker 2017). For example, PI 268868 is a Virginia market type belonging to the *hypogaea* variety, but its pod shape resembles *fastigiata*. Even more interesting, PI 290566 is described as a Runner belonging to variety *fastigiata* and having a *fastigiata* pod shape. In this table, we added kernel color information, i.e., hue, lightness, a*, and b* color properties, derived from pictures available on the GRIN database.

In this study, trait differences among years may be explained, in part, by differences in weather patterns during the growing seasons (Figure 1). For example, more $GDD_{13}$ were accumulated before 6 WAP (26 June) in 2018, as compared to 2017, followed by an increased mid- to end-season precipitation (Figure 1). Under these conditions, plants grew faster in the early 2018 season, compared to 2017. In 2018, warm temperature to mid-season was accompanied by heavy rainfall in subsequent weeks (8–10 WAP), causing a humid environment (average RH was 88% during 8–10 WAP), and increased disease pressure ($p < 0.0001$), compared to 2017. The average disease incidence in 2018 for TSW, SSR, SB, and CBR, were 13.7, 3.2, 2.4, and 2.2, respectively, higher than 6.9, 0.19, 0.68, and 0.13, respectively, in 2017.

Repeatedly measured ANOVA showed significant interactions of WAP with genotype and variety for plant height, during the rapid growth phase from 4 to 6 WAP in both years (Figure 2). Unlike in 2017, lateral growth seemed to plateau after 6 WAP in 2018, when weather conditions favored excessive vegetative growth early in the season. In both years, NDVI increased rapidly and reached close to the maximum values by 7 WAP. This coincided with the beginning of the pegging growth stage, when lateral branches from two adjacent rows are close to touching. At this point, agronomists recommend application of growth regulators to restrain abundant biomass growth and maintain row direction visibility at digging; otherwise, pods could be cut into the ground and yield substantially reduced [93]. Therefore, NDVI plateauing from several flight missions can be used as a marker for the best time to control biomass accumulation with growth regulator applications. Our results confirmed that variety *vulgaris* is morphologically taller than other varieties, and had the highest NDVI early in the season [48,49,94] (Figure 3).

Leaf wilting varied by year. Compared to 2018, leaves in 2017 started to wilt later in the season (10 WAP) after 12–14 days (17–29 July 2017) of insignificant precipitation (Figure 4). Varieties *vulgaris* and *fastigiata* were more wilted than *hypogaea*; *peruviana* had the highest wilting values. In 2018, wilting was highest at 5 to 7 WAP because of increased temperatures at this time, reflected by more $GDD_{13}$ accumulation than in 2017; *peruviana* and *vulgaris* were more wilted than *hypogaea* and *fastigiata*. In both years, *peruviana* had the coolest canopies among all varieties during sudden droughts (5 and 7 WAP in 2017; 10 and 12 WAP in 2018), which could be the result of increased transpiration, i.e. lower CTD values, and increased water use efficiency [95]. Warmer temperatures and less rainfall in 2018 also resulted in higher thrips pressure [96], causing significantly more damage in 2018 (average value 3.12) as compared to almost zero in 2017. Among the genotypes, CC760 was the tallest and had one of the highest wilting scores (>3) in both years (Table 5); CC760 also had one of the highest incidences of TSW in 2017. In 2017, taller genotypes were more prone to wilting ($R^2 = 0.58$) and high TSWV incidence ($R^2 = 0.35$) than shorter genotypes (Figure 5). The positive relationship between plant height and leaf wilting may be related to longer internodes, more open canopies, and increased exposure to radiation and wind,

for tall genotypes favoring moisture loss through latent heat flux associated with boundary layer thickness [97,98]. Longer internodes leading to open canopies could also be more favorable to thrips infestation, and thrips-vectored TSWV incidence and severity [99–101].

Cultivars Wynne, Walton, and C76-16, were top yielders in both years, which is expected as they were the high yielding checks selected for our study. However, a few accessions were comparable with these cultivars for yield production. For example, in 2017, CC650, CC246, and CC223, produced comparable yield with the check genotypes, above 7000 kg ha$^{-1}$. In 2018, CC789, CC068, and CC477, produced over 3800 kg ha$^{-1}$. The average yield in Virginia was 5100 kg ha$^{-1}$ in 2017 and 4700 kg ha$^{-1}$ in 2018 (USDA-NASS, 2017 and 2018), which suggests that new sources for yield improvement exist within the mini-core collection. Previous studies have also shown CC068 to be comparatively resistant to SB and SSR [40,47]. Post-harvest sprouting was lower in 2017 than in 2018, with CC038 sprouting the most. Post-harvest sprouting is caused by weakened pegs allowing pods to detach from the vines during digging [102]. Though the peanuts were harvested around 16 WAP in both years, disease pressure in 2018 may have reduced leaf photosynthesis and assimilate partitioning to pegs during pod development, possibly resulting in weakened pegs and more pod loss [103]. In 2018, post-harvest sprouting was significantly correlated ($p < 0.0001$) to all three fungal diseases, SSR, SB, and CBR (data not shown).

Among varieties, *vulgaris* was the tallest and had the most lateral growth in 2017, but in 2018 there were no differences. In 2018, increased GDD$_{13}$ accumulation before 6 WAP may have reduced the differences among varieties. Variety *hypogaea* had the highest pod yield in 2017, whereas in 2018, yield differences were minimized by heavy disease pressure. In both years, post-harvest sprouting was more severe for *vulgaris* and *fastigiata*, in comparison with *hypogaea* and *peruviana*, with the earlier maturing having higher sprouting than the later.

Aerially collected VIs were significantly correlated with all morphological and agronomic characteristics measured in this study, and was similar for each variety. In 2018, correlations were weaker than in 2017, but this could have been caused by faster growth early in the 2018 season. Fewer differences between entries and more disease were observed in 2018, compared with 2017. Regardless, some VIs continued to show significant correlations across years with the plant characteristics, in particular within variety *hypogaea*, which seemed to be less affected by sudden droughts and had less disease than the other varieties (Table 8 and 9). Of these, several VIs also showed improved H$^2$ over yield and other peanut traits, suggesting possible use for breeding selection for improved peanut cultivars.

Aerial imagery has shown potential to be a faster and relatively cheap option for crop phenotyping [19,104,105]. It can be tool for varietal selection of crops by remote trait estimation, and use of spectral reflectance and its derivatives, as a trait itself. However, in light of recent technological advancements, we have observed UAVs and associated sensors becoming outdated within the first couple of years. This is a challenge for low budget research programs [106]. Further, little progress has been made in UAV autonomy and in-season decision making using aerial sensors [107]. In-season decision making has been hindered by a lack of high-end processing computers and autonomy in image processing; the former being a budget issue, and the latter being an issue with the lack of technology. Use of aerial imagery data for machine learning, deep learning, computer vision, internet-of-things, and crop modelling approaches, requires significant technical expertise in the field of computer science and calls for interdisciplinary research [107–109]. Although our study offers a methodology for faster phenotyping, further research is required to make this faster and more autonomous.

### 5. Conclusions

In 2017 and 2018, in Suffolk, VA, this study evaluated up to 93 U.S. peanut mini-core germplasm collection accessions for morphological, physiological and agronomic attributes, viral and fungal diseases, pod yield, and post-harvest sprouting. This study also evaluated 24 VIs extracted from blue, green, red, and NIR reflectance; 11 VIs from color space indices (CIE-Lab and CIE-Luv), and 13 VIs from combinations of reflectance and color space indices extracted from aerially collected plot images. Genotypes CC548, CC535, CC249, and CC233, were among the least wilted under intermittent drought conditions and produced high yields each year. CC650 had a high yield in 2017, and CC068 produced the highest yield under severe disease pressure in 2018. Aerial VIs were associated with the physiological and agronomical characteristics for all botanical varieties, but the strength of the association depended on year (less in 2018 than in 2017) and the trait (crop stand, height, and branching > yield > disease and post-harvest sprouting). Broad sense heritability ($H^2$) varied depending on the trait and the growth stage when data were collected. Certain VIs, such as the normalized difference CIE-Lab (NDLab) and CIE-Luv (NDLuv), were significantly correlated with physiologic and agronomic characteristics in both years; NDLab and NDLuv were significantly correlated with pod yield. While $H^2$ for pod yield was low, $H^2$ for NDLab and NDLuv was higher than 0.5 when these VIs were assessed during the pod development stage. These results indicate that UAV-based sensors have potential for measuring physiologic and agronomic characteristics for peanut breeding and precision agriculture applications.

**Author Contributions:** M.B., M.D.B., R.S.B., K.D.C., N.W., P.P., J.M. and J.C. conceptualized the project. M.D.B., R.S.B., K.D.C., M.W., J.C. and C.-J.S. multiplied and provided peanut seeds for the study. J.O. and S.S. prepared the flight plan and flew the UAV for aerial images. J.O. and A.-B.C. helped develop protocols and routines for image processing and analysis. S.S. mainly accomplished the hypothesis and objective development, with advice and comments from M.B., D.S.M. and W.E.T., S.S. collected and analyzed the data and images, derived the spectral and color indices, and wrote the manuscript. M.D.B., R.S.B. and M.B. provided thorough review of the data analysis and the manuscript. All authors have read and agreed to the published version of the manuscript.

**Funding:** This study was funded by USDA NIFA-CARE and NIFA-AFRI grant (grant no.-2017-67013-26193) and the Virginia Crop Improvement Association (VCIA).

**Institutional Review Board Statement:** Not applicable.

**Informed Consent Statement:** Not applicable.

**Data Availability Statement:** The datasets analyzed during the current study are not publicly available as they are being used to write other manuscripts. The datasets will be made available from the corresponding author on request by reviewers or editors. Information regarding U.S. minicore peanut germplasm is publicly available within the Germplasm Resource Information Network (GRIN) plant germplasm database (https://npgsweb.ars-grin.gov/gringlobal/search, accessed on 20 July 2022).

**Acknowledgments:** The authors would like to thank the sponsors, USDA-NIFA and VCIA, and lab technicians Doug Redd, Frank Bryant, and Collin Hoy, for their help in tillage operations, establishment, management, and field data collection of the peanut plots. Funding for publication was provided by the Virginia Tech' Open Access Subvention Fund. Mention of trade names or commercial products in this publication is solely for the purpose of providing specific information, and does not imply recommendation or endorsement by the U.S. Department of Agriculture. USDA is an equal opportunity provider and employer.

**Conflicts of Interest:** The authors hereby declare that they have NO affiliations with or involvement in any organization or entity with any financial interest (such as honoraria; educational grants; participation in speakers' bureaus; membership; employment; consultancies; stock ownership; or other equity interest; expert testimony; or patent-licensing arrangements), or non-financial interest (such as personal or professional relationships; affiliations; knowledge; or beliefs) in the subject matter or materials discussed in this manuscript.

**Abbreviations**

Weeks after planting, WAP; normalized difference vegetation index, NDVI; canopy temperature depression, CTD; tomato spotted wilt virus, TSWV; southern stem rot, SSR; sclerotinia blight, SB; cylindrocladium black rot, CBR; Germplasm Resource Information Network, GRIN; vegetation indices, VIs; red-green-blue, RGB; near-infrared, NIR.

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
