# Peer review of "Evaluation of the U.S. Peanut Germplasm Mini-Core Collection in the Virginia-Carolina Region Using Traditional and New High-Throughput Methods"

_agronomy, doi:10.3390/agronomy12081945_

Round 1

Reviewer 1 Report

General: This paper is nicely drafted article about evaluation of peanut germplasm using HTP. It will help in decision making processes specially related to screening cultivar for stress tolerance

Abstract:

Line 36: Please write 1-2 lines about future implications of this work.

Introduction: Nicely written introduction. However, recent literature are missing 

Methodology:

Please highlight more about statistical analysis of imageries. In varietal comparison, tukey test can be performed.

 Discussion

I would suggest the authors to have a more supported discussion with references considering the main points: The limitations of method and considerations when to apply the studied methodology and then the potential next steps or further investigation to address these limitations. 

References: Please double check the style of references (MDPI) and missing one

Author Response

General: This paper is nicely drafted article about evaluation of peanut germplasm using HTP. It will help in decision making processes specially related to screening cultivar for stress tolerance

Abstract:

Line 36: Please write 1-2 lines about future implications of this work.

We thank the reviewer for this comment, future implications of this study has been included at the end of abstract “Further, this study indicates that UAV-based sensors have potential for measuring physiologic and agronomic characteristics measured for peanut breeding, variable rate input application, real time decision making, and precision agriculture applications.”

Introduction: Nicely written introduction. However, recent literature are missing 

We appreciate and agree with the reviewer’s comment, recent literature relevant to effect of drought stress and previous studies using HTP have been added. New literature added are as followes:

Araya, A.; Jha, P.; Zambreski, Z.; Faye, A.; Ciampitti, I.; Min, D.; Gowda, P.; Singh, U.; and Prasad, P. Evaluating Crop Management Options for Sorghum, Pearl Millet and Peanut to Minimize Risk under the Projected Midcentury Climate Scenario for Different Locations in Senegal. Climate Risk Management, 2022, 36 100436.

Araya, A.; Prasad, P.; Ciampitti, I.; and Jha, P. Using Crop Simulation Model to Evaluate Influence of Water Management Practices and Multiple Cropping Systems on Crop Yields: A Case Study for Ethiopian Highlands. Field Crops Research, 2021, 260 108004.

Jha, P. K.; Kumar, S. N.; and Ines, A. V. Responses of Soybean to Water Stress and Supplemental Irrigation in Upper Indo-Gangetic Plain: Field Experiment and Modeling Approach. Field Crops Research, 2018, 219 76-86.

Yadav, M. R.; Choudhary, M.; Singh, J.; Lal, M. K.; Jha, P. K.; Udawat, P.; Gupta, N. K.; Rajput, V. D.; Garg, N. K.; and Maheshwari, C. Impacts, Tolerance, Adaptation, and Mitigation of Heat Stress on Wheat under Changing Climates. International Journal of Molecular Sciences, 2022, 23, no. 5 2838.

Methodology:

Please highlight more about statistical analysis of imageries. In varietal comparison, tukey test can be performed.

Based on the reviewer’s comments, details of statistical analysis of imageries have been added (lines 265 – 269, and Line 273 – 274). It reads “All image derived VIs and color space indices were correlated to ground based traits. Pearson’s correlation was performed separately for image and ground traits of each botanical variety. Since only two genotypes of variety peruviana were included, it was pooled along with vulgaris owing to their morphological similarities (Stalker, 2017). PROC CORR was further used to create Pearson’s correlation matrix heatmap.”; and “H2 was calculated for all ground based and aerially derived traits.”

For varietal comparison of our study, Tukey test was not suitable because we had 104 genotypes to compare, and Tukey’s post hoc test is unprotected and is prone to Type 1 error.

 Discussion

I would suggest the authors to have a more supported discussion with references considering the main points: The limitations of method and considerations when to apply the studied methodology and then the potential next steps or further investigation to address these limitations. 

That is a great suggestion, we have added a paragraph with limitations of our methodology and suggestions for further research (Line 547 - 559). It reads “Aerial imagery has shown potential to be a faster and relatively cheaper option for crop phenotyping [19, 104, 105]. It can be tool for varietal selection of crops by remote trait estimation and use of spectral reflectance and its derivatives as trait itself. However, in the event of recent technological advancements we have observed UAVs and associated sensors getting outdated by newer and better ones within first couple of years. This is a challenge for low budget research programs [106]. Further, little progress has been made in UAV autonomy and in season decision making using aerial sensors [107]. In season decision making has been hindered by lack of high-end processing computers and autonomy in image processing. The former being a budget issue and later being an issue with lack of technology. Use of aerial imagery data for machine learning, deep learning, computer vision, internet-of-things, and crop modelling approaches requires significant technical expertise in the field of computer science and calls for interdisciplinary research [106, 108, 109]. Though, our study offers a methodology for faster phenotyping, further research is required to make it faster and more autonomous.”.

References: Please double check the style of references (MDPI) and missing one

We thank the reviewer for bringing it to our cognizance, we have rectified the reference style to match that of MDPI.

Reviewer 2 Report

Peanut is a widely cultivated crop species and a significant economic crop, while the plants are constantly exposed to various unfavourable conditions in environments. This study evaluated 93 U.S. peanut germplasm which provide powerful tools for breeding.

1. In line 31, the mini-core collection has been identified in previous studies,  with morphological, physiological and agronomic characteristics, whether the phenotyping data consisted with the previous reports.

2. In conclusion part, this study evaluated up to 93 U.S. peanut mini-core germplasm collection accessions for morphological, physiological, and agronomic attributes, viral and fungal diseases, pod yield, and post-harvest sprouting, in 2017 and 2018 in Suffolk, VA. Why were 93 mini-core accessions planted in 2017, 81 accessions were planted in 2018.

3. In line 467, CC068 produced highest yield under severe disease pressure in 2018, which indicated several diseases had occurred. However, which kinds of diseases occurred and whether CC068 showed resistance in your survey.

Author Response

Peanut is a widely cultivated crop species and a significant economic crop, while the plants are constantly exposed to various unfavourable conditions in environments. This study evaluated 93 U.S. peanut germplasm which provide powerful tools for breeding.

  1. In line 31, the mini-core collection has been identified in previous studies, with morphological, physiological and agronomic characteristics, whether the phenotyping data consisted with the previous reports.

We thank the reviewer for this comment, US peanut minicore has not been evaluated for morphological, physiological and agronomic characteristics in any previous study. We have added the information in line 99- 101, and it reads: “To our knowledge, the U.S. peanut minicore has neither been evaluated in the VC region, nor been evaluated for morphological, physiological and agronomic characteristics anywhere else.”

  1. In conclusion part, this study evaluated up to 93 U.S. peanut mini-core germplasm collection accessions for morphological, physiological, and agronomic attributes, viral and fungal diseases, pod yield, and post-harvest sprouting, in 2017 and 2018 in Suffolk, VA. Why were 93 mini-core accessions planted in 2017, 81 accessions were planted in 2018.

We thank the reviewer for this comment, We planted 93 accessions in 2017 and 81 in 2018 because we did not have enough seeds in 2018 to plant all accessions. We have mentioned that in line 138-139, “Based on seed availability, 93 mini core accessions and 11 check cultivars were planted in 2017, and 81 accessions and 7 check cultivars were planted in 2018.”

  1. In line 467, CC068 produced highest yield under severe disease pressure in 2018, which indicated several diseases had occurred. However, which kinds of diseases occurred and whether CC068 showed resistance in your survey.

We thank the reviewer for this comment. The diseases that occurred in 2018 were southern stem rot (SSR), Sclerotinia blight (SB), Cylindrocladium black rot (CBR) and tomato spotted wilt virus (TSWV), we have mentioned that in line 483-485, “The average disease incidence in 2018 for TSW, SSR, SB and CBR were 13.7, 3.2, 2.4, and 2.2 respectively, higher than 6.9, 0.19, 0.68, and 0.13, respectively, in 2017.” Also, CC068 has shown resistance to SB and SSR in previous controlled environment studies. We have added that to line “Previous studies have also shown CC068 to be comparatively resistant to SB and SSR [40, 47].”